# Measuring the state and temporal evolution of glaciers in Alaska and Yukon using SAR-derived 3D time series of glacier surface flow

Sergey Samsonov[1], Kristy Tiampo[2], and Ryan Cassotto[2]

[1]Canada Centre for Mapping and Earth Observation, Natural Resources Canada, 560 Rochester Street, Ottawa, ON K1S5K2 Canada
[2]Earth Science & Observation Center, Cooperative Institute for Research in Environmental Sciences, University of Colorado, Boulder, CO 80309 USA

**Correspondence:** Sergey Samsonov (sergey.samsonov@canada.ca)

**Abstract.** Climate change has reduced global ice mass over the last two decades as enhanced warming has accelerated surface melt and runoff rates. Additionally, glaciers have undergone dynamic processes in response to a warming climate that impacts the surface geometry and mass distribution of glacial ice. However, until recently no single technique could consistently measure the evolution of surface flow for an entire glaciated region in three dimensions with high temporal and spatial resolution. We have improved upon earlier methods to measure the evolution of surface flow by developing a technique for mapping, in unprecedented detail, the temporal evolution of glaciers in southeastern Alaska during 2016-2020. We observe seasonal and interannual variations in flow velocities at Seward and Malaspina glaciers as well as culminating phases of surging at Klutlan, Walsh and Kluane glaciers. On a broader scale, this technique can be used for reconstructing the response of worldwide glaciers to the warming climate using archived SAR data and for near real-time monitoring of these glaciers using rapid revisit SAR data from satellites, such as Sentinel-1 (6 or 12 days revisit period) and the forthcoming NISAR mission (12 days revisit period).

## 1 Introduction

Glacier dynamics, specifically the direction and intensity of glacier flow, adjust in response to the warming climate, leading to changes in seasonal flooding and droughts, landscapes and habitats, and ultimately sea-level variations. Surface flow is a key variable for determining glacier mass balance (Shepherd et al., 2020), ice thickness (Morlighem et al., 2011; Werder et al., 2019) and surface mass balance (Bisset et al., 2020). Here we present a technique that can be used for measuring the temporal evolution of surface flow for an entire glaciated region in three dimensions (3D) with high temporal and spatial resolution.

Modern techniques and platforms used for monitoring glacier flow include Synthetic Aperture Radar (SAR) (Goldstein et al., 1993; Mohr et al., 1998; Rignot, 2002; Joughin, 2002), Global Navigation Satellite System (GNSS) (van de Wal et al., 2008; Bartholomew et al., 2010), optical imagery (Berthier et al., 2005; Herman et al., 2011; Dehecq et al., 2015; Fahnestock et al., 2016), and uncrewed aerial vehicles (Immerzeel et al., 2014). Among these, SAR is the only active side-looking sensor with global coverage at high temporal and spatial resolutions that can operate in any weather conditions, day or night. SAR techniques comprise displacement measurements with sub-meter to meter-scale precision using speckle offset tracking

(SPO) (Strozzi et al., 2002) and split-beam interferometry (or multi-aperture interferometry, MAI) (Bechor and Zebker, 2006; Gourmelen et al., 2011), and centimeter-scale differential interferometry (DInSAR) (Massonnet and Feigl, 1995; Rosen et al., 2000). SPO applies image correlation algorithms to radar data to measure displacements in the satellite range and azimuth directions using two SAR images. Since its early inception SAR has been used in glacier monitoring for estimating flow veloc-ities, surface flux, tidal variations, grounding line behaviour and subglacial lake activity (Goldstein et al., 1993; Joughin et al., 1995, 1998; Rignot, 1998; Shepherd et al., 2001; Gray et al., 2005; Palmer et al., 2010; Minchew et al., 2017). In this study we use the SPO technique to produce deformation maps in range and azimuth coordinates that do not require phase unwrapping.

The SAR-derived displacements for a single epoch can be transformed into 3D (north, east, vertical) displacements by either combining multiple data sets or assuming various model constraints (Mohr et al., 1998; Wright et al., 2004; Gourmelen et al., 2007; Kumar et al., 2011; Hu et al., 2014). However, the 3D displacement time series cannot be easily computed due to limitations inherent in the data acquisition strategy. Specifically, SAR data on ascending and descending orbits are usually acquired at different days, often with different incidence angles, and varying temporal and spatial resolutions and wavelengths. The Multidimensional Small Baseline Subset (MSBAS) methodology (Samsonov and d'Oreye, 2012, 2017; Samsonov, 2019; Samsonov et al., 2020) has been developed especially for computing multidimensional displacement time series from SAR data acquired with different acquisition parameters.

Historically, three components of mean glacier velocity were computed from DInSAR and/or range offsets by introducing a surface-parallel flow (SPF) constraint. This approach was used for 3D mapping of Greenlandic (Joughin et al., 1998; Mohr et al., 1998) and Himalayan (Kumar et al., 2011) glaciers and validated by independent GPS (Kumar et al., 2011) measurements. In our previous work (Samsonov, 2019), we adopted the SPF method for computing the 3D flow displacement time series of the Barnes Ice Cap using ascending and descending DInSAR data combined using MSBAS technique. However, the SPF constraint ignores submergence and emergence velocities and other vertical motion. In some studies, ascending and descending DInSAR (Gray, 2011) or range and azimuth offsets (Wang et al., 2019) were used to compute 3D glacier velocities for a few isolated epochs. Recently, Guo et al. (2020) developed a technique based on MSBAS that computes 3D flow velocity time series from ascending and descending range and azimuth offsets and used it for studying Hispar Glacier in Central Karakoram. Here, we present our independently developed version of this algorithm, which offers several distinct advantages over Guo et al. (2020). First, our technique does not use weights determined by the pixel spacing. Second, our open source software provides additional functionalities, such as zeroth, first (implemented in Guo et al. (2020)) and second-order Tikhonov regularizations. Third, the user can choose to compute 1D, 2D, constrained 3D, unconstrained 3D (presented in this manuscript), and 4D (Samsonov et al., 2021) velocity and/or displacement time series. Finally, the software is also parallelized (OpenMP, MPI), making it suitable for running on personal workstations and high-performance computers.

In contrast to MSBAS-based techniques, Minchew et al. (2017) and Milillo et al. (2017), took a different approach and inferred time-dependent 3D flow velocity by assuming a form for the temporal basis functions based on prior knowledge of the study area. The need for prior knowledge means that this method is not general and so its application is limited to areas where the assumed basis functions should be valid. But the advantage of the Minchew et al. (2017) approach is interpretability of the results, a straightforward connection of the results to the physics of the systems being observed, and robust quantification

of uncertainties. A recent improvement to Minchew et al. (2017) is Riel et al. (2021) that adopts some of the methods of
Riel et al. (2014, 2018) and applies them to remote sensing observations of glaciers. From a methodological perspective, this
generalizes the approach of Minchew et al. (2017) and allows for a generic set of temporal basis functions, from which a
sparsity-inducing optimization is used to identify the simplest set of basis functions that describe the data. The advantage
there is also in the interpretability of the results and robust uncertainty quantification, which provides the ability to decompose
the observed signal into short and long-term variations, and features the ability to constrain transients, secular, and periodic
signals. However, this method still requires a priori knowledge to provide confidence in the resulting basis functions. The
technique we present here is complementary because it does not rely on basis functions and provides flexibility at the expense
of interpretability of the results, whereas the Minchew et al. (2017) and Riel et al. (2021) techniques sacrifice flexibility in the
method for enhanced interpretability of the results.

Here we focus on dynamic changes along six land-terminating glaciers in southeastern Alaska during 20 October 2016 -
21 January 2021: Agassiz, Seward, Malaspina, Klutlan, Kluane and Walsh Glaciers (Figure 4). This technique can be used to
analyze 3D flow velocities of glacier surfaces over large regional scales using nearly three decades years of archived SAR data
and for near real-time monitoring of these glaciers using rapid revisit SAR data.

## 2    Model

The inversion technique described below utilizes ascending and descending range and azimuth speckle offset products computed from SAR data using speckle offset tracking algorithm implemented in GAMMA software (Wegmuller and Werner, 1997). We chose to use speckle offsets because their computation does not require phase unwrapping, which is not possible due to large flow velocities in our study area.

The 3D displacement time series are computed by inverting a set of linear equations, first solving for the north, east and vertical flow velocities $V_{n,e,v}$ for each acquisition epoch (Fialko et al., 2001; Bechor and Zebker, 2006) and then for cumulative 3D flow displacements $D_{n,e,v}$

$$\begin{pmatrix} A \\ \lambda L \end{pmatrix} \begin{pmatrix} V_n \\ V_e \\ V_v \end{pmatrix} = \begin{pmatrix} RO^{asc} \\ AO^{asc} \\ RO^{dsc} \\ AO^{dsc} \\ 0 \end{pmatrix} \tag{1a}$$

$$D_{n,e,v}^{i+1} = D_{n,e,v}^i + V_{n,e,v}^i \Delta t_i. \tag{1b}$$

Equation (1a) has a straightforward application: time interval multiplied by velocity is equal to displacement. Here, in a matrix form, $RO$ are the range and $AO$ are the azimuth offsets computed from SAR data; $L$ is the Tikhonov regularization matrix multiplied by the scalar regularization parameter, $\lambda$, and $A$ is the transform matrix constructed from the time intervals

between consecutive SAR acquisitions and the range ($s_\rho$) and azimuth ($s_\alpha$) directional cosines with north, east, and vertical components

$$s_\rho = \{s_{n\rho}, s_{e\rho}, s_{v\rho}\} = \{\sin(\phi)\sin(\theta), -\cos(\phi)\sin(\theta), \cos(\theta)\}$$

$$s_\alpha = \{s_{n\alpha}, s_{e\alpha}, s_{v\alpha}\} = \{\cos(\phi), \sin(\phi), 0\}, \tag{2}$$

where $\phi$ is the azimuth and $\theta$ is the incidence angles. The azimuth angle is the satellite heading, measured from the north; it

discerns ascending vs descending orbits. The incidence angle is the angle between the nadir and the look direction from the satellite; it is one of the acquisition parameters of the side-looking SAR sensor.

The need for regularization arises because SAR images from different tracks are acquired at different times, which results in more unknowns than equations, producing a rank-deficient, underdetermined problem. When solving a set of linear equations in general there can be three possible scenarios: the number of equations can be less, equal, or greater than the number of

unknowns. In the equal case, the matrix is square and no regularization is required (but can still be applied). In the greater case, the least square solution is found using SVD, this scenario is common in 1D MSBAS, where usually there are more interferograms than SLCs. In the lesser case, as always in 2D and 3D MSBAS, the solution is found using either the truncated-SVD or the zeroth-order Tikhonov regularization. The higher-order regularizations must be applied if the objective is to fill the temporal gaps due to missing data, which results in smoothing and the interpolation of missing values in the temporal domain.

We observe that the first and second-order regularizations work equally well in this case, probably because of slowly changing velocities.

In the $M \times N$ transform matrix $A$ with $M$ rows and $N$ columns, $N$ is equal to the number of available distinct SLC images (with the boundary correction - defined below) minus one then multiplied by three (i.e. $N = 3(\sum_{k=1}^{K} N_{slc}^k - 1)$, where $K$ is the total number of ascending and descending sets and $N_{slc}^k$ is the number of SLC images in $k$ set). $M$ is equal to the total number

of range and azimuth offset maps computed from those SLC images (i.e. $M = \sum_{k=1}^{K} (N_\alpha^k + N_\rho^k)$, where $K$ is the total number of ascending and descending sets, $N_\alpha^k$ is the number of computed azimuth offset maps and $N_\rho^k$ is the number of computed range offset maps in $k$ set).

The regularization matrix $L$ has the same number of columns $N$ as the transform matrix $A$ but its number of rows depends on the regularization order. It is equal to $N$ for the zeroth order, $N - 3$ for the first order, and $N - 6$ for the second order.

The structure of $A$ can be deduced from a simplified example shown in Figure 1 and described below. In this example, it is assumed that the ascending set consists of three SAR images acquired on $t_0$, $t_2$, and $t_4$ and the descending set consists of four SAR images acquired on $t_{-1}$, $t_1$, $t_3$, and $t_5$. Two ascending range $RO^{asc} = \{\rho_{0-2}^{asc}, \rho_{2-4}^{asc}\}$ and azimuth $AO^{asc} = \{\alpha_{0-2}^{asc}, \alpha_{2-4}^{asc}\}$ offset products are computed from three ascending SAR images, and three descending range $RO^{dsc} = \{\rho_{-1-1}^{dsc}, \rho_{1-3}^{dsc}, \rho_{3-5}^{dsc}\}$ and azimuth $AO^{dsc} = \{\alpha_{-1-1}^{dsc}, \alpha_{1-3}^{dsc}, \alpha_{3-5}^{dsc}\}$ offset products are computed from four descending SAR images (therefore, $M =$

$2 + 2 + 3 + 3 = 10$). A boundary correction (shown as blue arrows in Figure 1) is applied to the first and last descending offset products $\rho_{-1-1}^{dsc}$, $\alpha_{-1-1}^{dsc}$, $\rho_{3-5}^{dsc}$, and $\alpha_{3-5}^{dsc}$ by multiplying by $(t_1 - t_0)/(t_1 - t_{-1})$ and $(t_4 - t_3)/(t_5 - t_3)$ in order to adjust the temporal coverage to match the ascending offset products. The boundary-corrected descending offsets therefore become $\rho_{0-1}^{dsc}$, $\alpha_{0-1}^{dsc}$, $\rho_{3-4}^{dsc}$, and $\alpha_{3-4}^{dsc}$. Note that the boundary correction reduces the number of SLC images by two; after

correction, $t_{-1}$ effectively becomes $t_0$ and $t_5$ effectively becomes $t_4$ (i.e. reducing the total number of SLC images to 5 and thus $N = 3(5-1) = 12$). The first-order regularization matrix $L$ in this case has twelve columns and nine rows.

Assuming that $\Delta t_i = t_{i+1} - t_i$ in this simplified example, equation (1a) becomes

$$
\begin{pmatrix}
s_{n\rho}^{asc}\Delta t_0 & s_{e\rho}^{asc}\Delta t_0 & s_{v\rho}^{asc}\Delta t_0 & s_{n\rho}^{asc}\Delta t_1 & s_{e\rho}^{asc}\Delta t_1 & s_{v\rho}^{asc}\Delta t_1 & 0 & 0 & 0 & 0 & 0 & 0 \\
0 & 0 & 0 & 0 & 0 & 0 & s_{n\rho}^{asc}\Delta t_2 & s_{e\rho}^{asc}\Delta t_2 & s_{v\rho}^{asc}\Delta t_2 & s_{n\rho}^{asc}\Delta t_3 & s_{e\rho}^{asc}\Delta t_3 & s_{v\rho}^{asc}\Delta t_3 \\
s_{n\alpha}^{asc}\Delta t_0 & s_{e\alpha}^{asc}\Delta t_0 & 0 & s_{n\alpha}^{asc}\Delta t_1 & s_{e\alpha}^{asc}\Delta t_1 & 0 & 0 & 0 & 0 & 0 & 0 & 0 \\
0 & 0 & 0 & 0 & 0 & 0 & s_{n\alpha}^{asc}\Delta t_2 & s_{e\alpha}^{asc}\Delta t_2 & 0 & s_{n\alpha}^{asc}\Delta t_3 & s_{e\alpha}^{asc}\Delta t_3 & 0 \\
s_{n\rho}^{dsc}\Delta t_0 & s_{e\rho}^{dsc}\Delta t_0 & s_{v\rho}^{dsc}\Delta t_0 & 0 & 0 & 0 & 0 & 0 & 0 & 0 & 0 & 0 \\
0 & 0 & 0 & s_{n\rho}^{dsc}\Delta t_1 & s_{e\rho}^{dsc}\Delta t_1 & s_{v\rho}^{dsc}\Delta t_1 & s_{n\rho}^{dsc}\Delta t_2 & s_{e\rho}^{dsc}\Delta t_2 & s_{v\rho}^{dsc}\Delta t_2 & 0 & 0 & 0 \\
0 & 0 & 0 & 0 & 0 & 0 & 0 & 0 & 0 & s_{n\rho}^{dsc}\Delta t_3 & s_{e\rho}^{dsc}\Delta t_3 & s_{v\rho}^{dsc}\Delta t_3 \\
s_{n\alpha}^{dsc}\Delta t_0 & s_{e\alpha}^{dsc}\Delta t_0 & 0 & 0 & 0 & 0 & 0 & 0 & 0 & 0 & 0 & 0 \\
0 & 0 & 0 & s_{n\alpha}^{dsc}\Delta t_1 & s_{e\alpha}^{dsc}\Delta t_1 & 0 & s_{n\alpha}^{dsc}\Delta t_2 & s_{e\alpha}^{dsc}\Delta t_2 & 0 & 0 & 0 & 0 \\
0 & 0 & 0 & 0 & 0 & 0 & 0 & 0 & 0 & s_{n\alpha}^{dsc}\Delta t_3 & s_{e\alpha}^{dsc}\Delta t_3 & 0 \\
\lambda & 0 & 0 & -\lambda & 0 & 0 & 0 & 0 & 0 & 0 & 0 & 0 \\
0 & \lambda & 0 & 0 & -\lambda & 0 & 0 & 0 & 0 & 0 & 0 & 0 \\
0 & 0 & \lambda & 0 & 0 & -\lambda & 0 & 0 & 0 & 0 & 0 & 0 \\
0 & 0 & 0 & \lambda & 0 & 0 & -\lambda & 0 & 0 & 0 & 0 & 0 \\
0 & 0 & 0 & 0 & \lambda & 0 & 0 & -\lambda & 0 & 0 & 0 & 0 \\
0 & 0 & 0 & 0 & 0 & \lambda & 0 & 0 & -\lambda & 0 & 0 & 0 \\
0 & 0 & 0 & 0 & 0 & 0 & \lambda & 0 & 0 & -\lambda & 0 & 0 \\
0 & 0 & 0 & 0 & 0 & 0 & 0 & \lambda & 0 & 0 & -\lambda & 0 \\
0 & 0 & 0 & 0 & 0 & 0 & 0 & 0 & \lambda & 0 & 0 & -\lambda
\end{pmatrix}
\begin{pmatrix}
V_n^0 \\ V_e^0 \\ V_v^0 \\ V_n^1 \\ V_e^1 \\ V_v^1 \\ V_n^2 \\ V_e^2 \\ V_v^2 \\ V_n^3 \\ V_e^3 \\ V_v^3
\end{pmatrix}
=
\begin{pmatrix}
\rho_{0-2}^{asc} \\ \rho_{2-4}^{asc} \\ \alpha_{0-2}^{asc} \\ \alpha_{2-4}^{asc} \\ \rho_{0-1}^{dsc} \\ \rho_{1-3}^{dsc} \\ \rho_{3-4}^{dsc} \\ \alpha_{0-1}^{dsc} \\ \alpha_{1-3}^{dsc} \\ \alpha_{3-4}^{dsc} \\ 0 \\ 0 \\ 0 \\ 0 \\ 0 \\ 0 \\ 0 \\ 0 \\ 0
\end{pmatrix}. \tag{3}
$$

The 3D flow displacement time series are then computed as in the equation (1b) as $D_{n,e,v}^{i+1} = D_{n,e,v}^{i} + V_{n,e,v}^{i}\Delta t_i$, for $i = \{0,1,2,3\}$, assuming that the initial displacements $D_{n,e,v}^0$ are equal to zero. Note, that in this notation $D_{n,e,v}^0$ are the 3D displacements at the time $t_0$, while $V_{n,e,v}^0$ and $\Delta t_0$ are the 3D velocities and the time interval at the time epoch $t_0 - t_1$, thus effectively available at the time $t_1$. For simplicity of presentation, a linear trend is computed by applying linear regression to the derived values, calculated over the entire record, to illustrate the 3D displacement time series and three linear rate maps used for visualizing the results. Note, that in the case of non-steady-state flow the linear rates, which effectively are mean linear rates, can significantly differ from the instantaneous flow velocities. Linear rates can potentially be computed over a time interval of any duration (for example, one month or one year).

Tikhonov regularizations of various orders can be applied during the inversion, resulting in temporal smoothing. The zeroth-order regularization effectively is the constant displacement constraint. The first-order regularization effectively is the constant velocity constraint, and the second-order regularization effectively is the constant acceleration constraint. The first and second-order regularizations both produce good, virtually indistinguishable, results. The example above, equation (3), uses first-order regularization. Zeroth- and second-order regularizations are explicitly shown in Samsonov and d'Oreye (2017) for the 2D case. The magnitude of smoothing is controlled by the regularization parameter $\lambda$ that can be selected, for example, using the L-curve method (Hansen and O'Leary, 1993; Samsonov and d'Oreye, 2017). The value of $\lambda$ equal to 0.1 was used in our case.

MSBAS methodology has been developed for computing multidimensional time series by combining multiple DInSAR data acquired at different times and in various observational geometries. The 2D (east and vertical) method was described in

(Samsonov and d'Oreye, 2012, 2017) and the surface-parallel-flow constrained 3D (north, east, vertical) method in (Samsonov, 2019; Samsonov et al., 2020). The unconstrained 3D method (i.e. without surface-parallel-flow constrain) presented here uses both range and azimuth measurements for computing 3D displacements. This work is now possible due to the improved availability over large areas of high-quality, high-resolution, temporally dense ascending and descending SAR data and the increase in computational power that allows computing a large amount of range and azimuth offset maps and inverting large

matrices. Since this method does not make any assumptions about the direction of motion, it provides the optimal solution applicable to any surface motion (e.g. glacier flow, tectonic and anthropogenic deformation, etc). The typical size of the transform matrix exceeds 100s and often 1000s of columns and rows for each pixel. It is $446 \times 666$ (or $1109 \times 666$ including regularization terms; in the following, for simplicity, matrix $L$ is assumed to be a part of matrix $A$) in our case. Thus, the total number of azimuth and range offset maps $M$ equals 446, and the number of unknowns $N$ equals 666, which corresponds to

223 SLC images after applying the boundary correction. The additional $N - 3 = 663$ rows represent the first-order Tikhonov regularization terms. The singular value decomposition (SVD) algorithm from the Linear Algebra PACKage (LAPACK) library called from C++ code is used for inverting this matrix for each pixel. Processing is parallelized using Open Multi-Processing (OpenMP) implementation of multi-threading. Depending on the number of cores in the processing unit and the number of pixels, this process can take from several hours to several days. Processing time in our case, on a 44-core workstation is

approximately 24 hours. The Message Passing Interface (MPI) version of the software has also been developed. The processing time in an MPI version is reduced proportionally to the number of nodes.

## 2.1  Synthetic tests

The effectiveness of the proposed technique was demonstrated with synthetic tests using the actual transform matrix $A$, in detail described in the next section. First, we reconstructed deformation components using the harmonic input signal in only

one of the components, described in the respectful legends in Figure 2(a-c). Then we added 10% of Gaussian noise and repeated the computations (Figure 2(d-f)). Second, we reconstructed deformation components using the complex partially *uncorrelated* input signal: harmonic (with the different period in all components) and linear input signals in the horizontal components and the harmonic signals in the vertical component. Three runs were performed with 0%, 10% and 30% Gaussian noise added (Figure 3(a-c)). Finally, we reconstructed deformation components using the complex *correlated* input signal: harmonic (with

the same period in all components) and linear input signals in all three components. Three runs were performed, again with 0%, 10% and 30% Gaussian noise added (Figure 3(d-f)).

Without added noise the reconstructed output signal is practically identical to the input signal; with added noise, the reconstructed signal still resembles the input signal very well. For a quantitative assessment, we computed correlation and covariance matrices between three vectors comprising east, north, and vertical components of velocity at each observation epochs. Six cor-

relation and covariance matrices are presented in Table 2 for each of six tests shown in Figure 3. Both matrices provide valuable information about the quality of reconstruction.

In the covariance matrices, diagonal elements are variances of north, east, and vertical components of the velocity. They reflect variability due to a true signal and noise. Potentially an input model can be subtracted to compute variances due to

noise, however, it is not a goal of this test. Instead, we are interested in covariance (i.e. non-diagonal) terms of the covariance matrix. They are expected to be small (comparable) in comparison to diagonal terms in the case of the uncorrelated (correlated) signal. In the correlation matrices, it is expected that non-diagonal terms should be small (close to one) in the case of uncorrelated (correlated) signal. Indeed, this pattern is clearly observed, suggesting that cross-feed between different components is negligibly small in both cases of uncorrelated (Table 2 a-c) and correlated (Table 2 d-f) signals.

Overall, these tests signify that the ascending/descending geometry is sufficient for a full reconstruction of 3D motion. This also can be inferred theoretically, by computing a rank of the transform matrix in the case of one ascending and one descending pair, acquired at the same time, which would be equal to three.

## 3   Study area and Data

Southeast Alaska has experienced significant ice mass loss and retreat over the last 50 years (Arendt et al., 2009; Arendt, 2011). Of the 27,000 glaciers that occupy the region, the majority (99.8%) are land terminating (RGIConsortium, 2017). Consequently, monitoring the mass balance and ice dynamic variations of Alaska's land-terminating glaciers is paramount for the future of its landscape and resultant contributions to sea level rise (Larsen et al., 2015). Unlike the plethora of ice velocity data products available for Greenland and Antarctica, regional studies of Alaskan glacier surface velocities pale in comparison. The first regional map of Alaskan glacier flow velocities was released in 2013 using ALOS PALSAR data (Burgess et al., 2013). Soon after, feature tracking of Landsat optical data began to regularly map regional surface velocities (Fahnestock et al., 2015; Gardner et al., 2018, 2019). Recent studies demonstrate the importance of characterizing the temporal evolution of glacier surface flow for understanding changes in ice dynamics in Alaska (Waechter et al., 2015; Altena et al., 2019). However, all regional studies of Alaskan glacier flow have so far been limited to two dimensions; thus, ignoring an important vertical component of flow, which links glacier surface elevation change and its mass balance. Here, we introduce a technique to generate a dense record of regional Alaskan glacier surface flow in three dimensions.

We focus on studying the dynamic changes along six land-terminating glaciers in southeastern Alaska during 20 October 2016 - 21 January 2021: Agassiz, Seward, Malaspina, Klutlan, Kluane and Walsh Glaciers (Figure 4). The Malaspina Glacier is the world's largest piedmont glacier covering approximately 2200 km$^2$ on the flat coastal foreland (Sharp, 1958; Muskett et al., 2003; Sauber et al., 2005) and is partially fed by Seward Glacier, a surge-type glacier that originates in the upper reaches of Mt. Logan (Sharp, 1951; Ford et al., 2003). A mass budget deficit in the Malaspina-Seward complex has long been recognized (Sharp, 1951). Agassiz Glacier is another surge-type that flows in an adjacent sinuous valley northwest of the Malaspina-Seward complex (Muskett et al., 2003; Sauber et al., 2005). The Klutlan Glacier is an 82-km long surge-type valley glacier located at elevations between 1300 and 2100 m; it has surged repeatedly over the last few hundred years (Wright, 1980; Driscoll, 1980). A surge at Kluane Glacier in the eastern St. Elias Mountains during 2017-2018 was previously reported in Main et al. (2019). Walsh Glacier is a 90-km long surge-type valley glacier located at a higher elevation of about 1500-3000. It is fed by two major branches, one from the north and one from the east and converges with the Logan Glacier downstream (Fu and Zhou, 2020).

In this study, we used 218 ascending (track 123) and 232 descending (track 116) Sentinel-1 Interferometric Wide (IW) single-look complex (SLC) images with 2.3 m (range) × 14.9 m (azimuth) spatial resolution from the NASA Distributed Active Archive Center (DAAC) operated by the Alaska Satellite Facility (ASF) (Table 1). Two ascending and two descending frames along the azimuth directions were concatenated for each, resulting in 109 and 116 swaths, respectfully. Ascending and descending sets were processed individually using GAMMA software (Wegmuller and Werner, 1997) that produced range and azimuth offsets for consecutive pairs (Figure 5). To compute offsets, we used a 64×16 pixels sampling interval (or approximately 200×200 m) and a square 128×128 pixels (or approximately 400×1600 m) correlation window. Such a large window was required to obtain a distinct, statistically significant peak of the 2D cross-correlation function; its square shape produced similar precision in range and azimuth directions in radar coordinates, and azimuth precision four times lower than range precision in geocoded products. Note, that the correlation window is not uniform, with larger weights given to the pixel in the center of the window. While the estimation of spatial resolution resulting from a non-uniform weighting of the pixel is beyond the scope of this study, the initial tests suggest that the spatial resolution is significantly better than the window size, which is also confirmed by the developers of GAMMA software. Offsets were spatially filtered using a Gaussian filter with a 1.3 km (6-sigma) filter-width, geocoded using TerraSAR-x 90 m DEM and resampled to a common grid with a ground spacing of 200 m. Using Gaussian weights for filtering proved to be particularly beneficial as the filter produced satisfactory results for small and large glaciers. Filter-width was chosen experimentally for our study but maybe sub-optimal in other regions.

## 4 Results

The magnitude of the mean 3D linear flow velocities plotted for the entire region using a logarithmic scale is shown in Figure 6. This figure contains a massive amount of data. Thus, an in-depth analysis was further performed for four small areas of interest (AOI1, AOI2, AOI3 and AOI4 in Figure 4) shown in detail in Figures 7-10. The flow lines in Figure 4 were computed using the Open Global Glacier Model (OGGM) software (Maussion et al., 2019) and the central flow lines were chosen for in-depth analysis. Note, that these flow lines are approximated to the actual glacier flow pattern. They are, however, computed in a consistent and repeatable way. In addition, time series sampled from 5× 5-pixel regions along Malaspina/Seward Glaciers (P1-P4), the Klutlan Glacier (P5-P6), Walsh Glacier (P7-P8) and Kluane Glacier (P9) are provided in Figures 11 and 12.

For each AOR, the SAR intensity images show the six glaciers in detail: Agassiz, Malaspina and Seward (AG, MG and SG, Figure 7(a)), Klutlan (KtG, Figure 8(a)), Walsh (WG, Figure 9(a)) and Kluane (KnG, Figure 10(a)) Glaciers. For five of these glaciers (excluding the Agassiz Glacier) velocities are sampled along flow lines with 20 km markers shown. Mean flow velocities are shown in Figures 7(b), 8(b), 9(b) and and 10(b); horizontal flow velocities are shown as vectors and vertical flow velocities are colour-coded, with red representing downward motion. For aesthetic purposes, horizontal flow vectors are resampled to a coarser resolution. The fastest horizontal flow velocity exceeds 1000 m/year and the fastest vertical flow velocity exceeds 200 m/year. Overall, Seward Glacier experiences the fastest motion and Malaspina Glacier experiences the slowest motion (Figure 7(b)); vertical flow is predominately downward along both glaciers. In contrast, vertical flow along the Klutlan (Figure 8(b)), Walsh (Figure 9(b)) and Kluane (Figure 10(b)) Glaciers changes direction a number of times.

The direction and magnitude of the mean linear flow velocities sampled along central flow lines from Malaspina and Seward, Klutlan, Walsh and Kluane Glaciers are shown in Figures 7(c), 8(c), 9(c) and 10(c) as vectors with tails that start at the surface elevation of each glacier. Animations of these flow velocities as time series along these profiles are also provided as supplemental materials. Note that the vertical axis (surface elevation) and horizontal axis (distance along profile) are scaled differently, producing significant but equal angular distortion in the flow velocities and topographic slopes. The mean linear flow velocities provide insight into the direction and magnitude of mean velocities calculated over a specific interval (e.g. the Sentinel-1 record); however, these values can vary over time. This is evident in the temporal evolution of the horizontal velocity magnitude and vertical velocity sampled along these profiles for the Seward and Malaspina Glaciers (Figures 7(d)-7(e)), the Klutlan Glacier (Figures 8(d)-8(e)), the Walsh Glacier (Figures 9(d)-9(e)) and the Kluane Glacier (Figures 10(d)-10(e)). Flow along the lower reaches of the Malaspina Glacier varies seasonally; although, the seasonal acceleration was delayed in 2020 and was higher in magnitude. Seasonal flow along the Klutlan, Walsh and Kluane Glaciers is far less pronounced; however, each shows an episodic shift in a flow that occurred around mid-2018, mid-2017, and mid-2018, respectively.

Examples of 3D flow displacement and velocity time series for the $5\times$ 5-pixel regions P1-P9 are shown in Figures 11-12. Similar time series can be easily produced for any coloured pixel in Figures 6; the locations selected were chosen to demonstrate diverse ice dynamic observations possible with the MSBAS-3D method. Regions P1 and P4 are located on the lower lobes of the Malaspina Glacier, at an elevation of about 200 m above sea level. The displacement time series show that flow is predominately west-southwest at P1 and northeast at P4. An abrupt change in a flow regime occurred at P1 at the end of June 2020. Since then, the flow velocity at P1 has remained elevated in comparison to the values observed in prior years. This many-fold velocity increase can also be observed in Figures 7d-7e along the later part of the profile. Horizontal and vertical flow velocities at these regions are only a few m/year, with a seasonal signal evident at P4 in the vertical component. Such seasonal signals are observed at most low-elevation glaciers. Regions P2 and P3 are located at elevations of about 1000 and 700 m. At these locations, horizontal flow dominates flow displacement, while vertical flow displacement is minimal. The southwest direction of flow is persistent at both locations. Flow velocities along the main branch of the Agassiz Glacier, not shown, are very similar to the flow velocities along the Seward Glacier but of a lesser magnitude.

Regions P5 and P6 are located on Klutlan Glacier at elevations of about 1900 and 1500 m, respectively. The overall vertical flow is slightly downward in these regions but horizontal and vertical components both show significant variability over time. Regions P7 and P8 are located on Walsh Glacier at an elevation of about 1700 and 2000 m, respectively. At P7, northwest upward displacement is observed until July 2017 when a gradual reduction occurred. Region P9 is located on Kluane Glacier at an elevation of about 1700 m. Here, southeast and upward displacement is observed during 2018 when a gradual reduction occurred. Error bars throughout Figures 11-12 show measurement variability within the $5\times$ 5-pixel region, rather than precision, though both quantities are likely related.

## 270  5   Discussion

The technique presented in this study is a viable solution for computing 3D flow displacement time series from ascending and descending range and azimuth SAR measurements. Synthetic tests (Figures 2-3) suggest that the precision of the inversion largely depends on the precision of input data and is not limited by the Sentinel-1 suboptimal acquisition geometry (i.e. nonorthogonal orbits). Range offsets can be substituted or complemented with DInSAR measurements since both measures

the same quantity; similarly, azimuth offsets can be substituted or complemented with Multiple Aperture Interferometry (MAI) (Bechor and Zebker, 2006) measurements. For high resolution SAR data, the precision of the SPO technique approaches that of DInSAR. In addition to glaciers, this technique can be used for studying other geophysical processes (e.g. landslides, sea/river/lake ice drift) if their motion exceeds the sensitivity of SPO and/or MAI techniques.

The reported precision of the individual offset maps computed using the SPO technique is 1/10-1/30 of the SAR pixel size

Strozzi et al. (2002). An average precision of our speckle offset product computed over a typical interval of 12 days (i.e. Sentinel-1 repeat period) is about 1 m (or 30 m/year) in range and 4 m (or 120 m/year) in azimuth. Our precision is lower than reported in (Strozzi et al., 2002) because we intentionally interpret the motion outside of glaciers (e.g. irregular snowdrift, landslides) as noise. However, for computing the mean linear velocity the length of the time series is more important than the precision of individual measurements. Standard deviations of the mean linear velocities averaged over the entire region are 0.7,

0.3, and 0.2 m/year (while the maximum values are 21, 18 and 7 m/year, these higher values would be due to seasonal variations and changes in surge activity) for northward, eastward and vertical components, respectively. This is somewhat analogous to the precision of GNSS-derived deformation rates, which largely depend on the length of time series rather than the precision of individual GNSS measurement. The best approach for estimating the absolute measurement accuracy, of course, is by comparing these remote-sensing measurements with ground-based measurements (Gudmundsson and Bauder, 1999), which

unfortunately are not available for this region and this period. SAR measures glacier motion at a certain depth rather than at the surface. Previous studies for this region suggest that the C-band SAR penetrates about four meters into the glacier's firn layer in dry conditions (Rignot et al., 2001). Standard deviation and coefficient of determination for each component of velocity and each pixel are provided in the supplement.

One of the practical computational challenges of the SPO technique is the selection of pixels, which offsets are computed

with high confidence. After multiple tests, we determined that the SNR function works very well for this purpose but only when the search window is large. However, such a large window applied to the medium resolution SAR data limits the spatial resolution of the results. It is possible to use high-resolution SAR data and the $128\times28$ pixels search window to overcome this limitation and achieve a high spatial resolution of results; however, such SAR data is not yet readily available on a global scale. The utilization of high-resolution SAR data also allows using a spatial filter with large, in terms of pixels, window size.

We compared the magnitude of mean linear horizontal flow velocities along the four profiles with the results presented in Gardner et al. (2019). There, surface velocities are derived from Landsat 4, 5, 7, and 8 imagery over the period from 1985 to 2018 using the auto-RIFT feature tacking processing chain described in Gardner et al. (2018). We used the horizontal velocities computed during 2017, 2018, and the entire 2018-present periods; these results are shown in the supplement. The velocities

computed over the 2017 and 2018 periods (Figures S7-S14) are in reasonable agreement. When we compare entire datasets (Figures S15-S18), they still show some agreement. Statistical parameters, such as correlation coefficients and RMSEs are provided in the figure captions. We observe that in areas experiencing nearly constant flow velocity, for example at Seward and Malaspina Glaciers, both datasets show close results with a correlation of 0.93 and RMSE of 269 m/year. At Klutlan, Walsh and Kluane Glaciers, SAR-derived velocities are affected by the surges, which are not reflected in Gardner et al. (2019) resulting in a lower correlation (0.43-0.80) and a larger RMSE (80-266 m/year) in comparison to the average velocity at those glaciers. Furthermore, the Landsat record will be temporally biased towards cloud-free images and periods when sufficient sunlight is available to obtain optical imagery; thus eliminating a significant portion of late fall and early winter scenes. One final discrepancy can be attributed to the differences in processing parameters, such as correlation window and filter shape and strength. Significant filtering is required in our processing because, for time series analysis, every single range and azimuth offset map must be defined at every pixel, which can be only achieved by using a large correlation window followed by strong filtering. Preservation of spatial coverage in every single range and azimuth offset map forces us to select pixels with a moderate signal-to-noise ratio (SNR), which would not have been selected if we wanted to compute only the mean velocity. Our software can potentially handle missing values in data by interpolating in the time domain (using first- and second-order regularizations), but here we have chosen not to introduce interpolation bias and instead lowered SNR. Overall, in addition to providing the three components of flow velocity at a higher and more consistent temporal resolution, our study demonstrates that deviations from the mean flow velocity can be very significant.

SAR-derived time series are often compared with the GNSS-derived time series and both techniques are considered conceptually similar; however, there is an important difference between the two, particularly when applied to glacier flow. SAR measures displacement within fixed geophysical locations (Eulerian representation), while GNSS measurements track receivers that are mounted to non-stationary geophysical surfaces (Lagrangian representation). Lagrangian displacement time series describes a trajectory of an object in space as a function of time, whereas the Eulerian displacement time series describes the cumulative length (of ice, rock, etc.) that flows through a fixed geophysical location. While the distinction between these reference frames is well established in the glaciological community, it is less commonly known in the solid-earth geophysical community (Samsonov and Tiampo, 2006; Samsonov et al., 2007; Gourmelen et al., 2010; Shen and Liu, 2020). For the latter, GNSS and SAR measurements can be considered nearly identical only when flow velocity at a fixed geophysical location (i.e. SAR pixel) is equal to the flow velocity at the GNSS site; that is when GNSS-derived displacements are contained within a single SAR pixel. This occurs when the flow velocity is very small and the material is rigid, as in tectonic deformation studies. For rapidly deforming glacier surfaces, differentiation is far more critical. Although the time-series in Figure 11 resemble GNSS-derived displacements, it is important to remember that these Eulerian measurements represent the cumulative displacement at any one pixel over time. Hence to emphasize this difference, we use the *flow displacement* terminology.

The overall direction of vertical flow is down along almost the entire length of the Seward and Malaspina Glaciers (Figure 7). The downward flow is expected in the upper reaches of accumulation zones because of firn compaction, and in areas with steeply dipping surfaces due to sloping bed topography; however, downward flow, with the slope steeper than surface topography in the lower ablation zone is more concerning. In general, the accumulation of snow and ice in high elevations

produces a net mass gain that replenishes ice lost through ablation processes along the lower glacier. In a steady-state, these processes balance each other and lead to submergent flow in the accumulation zone and emergent flow in the ablation zone; thus, ice mass lost through melt in the ablation zone is replenished by ice that emerges from the depths of ice columns to the glacier surface to maintain a consistent surface elevation (Hooke, 2019). The predominately downward flow of ice observed throughout the Malaspina Glacier's massive lobe (Figure 7b,c,e, 12d) indicates that ablation rates have exceeded emergence velocities during our four year study period, implying that the glacier is still adjusting to climatic warming. Indeed, the Seward, Malaspina, and Agassiz Glaciers are not in a steady-state (Muskett et al., 2003; Larsen et al., 2015). Seasonal variability is observed along the Seward and Malaspina Glaciers (Figures 7d,e, 12b,c,d). The fastest horizontal motion occurs during late spring-early summer and the slowest in late summer-early fall, consistent with other glaciers in the region (Abe and Furuya, 2015; Vijay and Braun, 2017; Enderlin et al., 2018). The fastest vertical motion is observed in the middle of summer. An animation provided with the supplementary data clearly shows seasonality in flow rates over the entire complex. Seasonal variability at P1 is obstructed by a many-fold increase in velocity observed in the second half of 2020 that lasts until the end of this study.

Velocities along Klutlan Glacier vary in more complex ways with multiple zones of upward and downward flow observed (Figure 8). This surge-type glacier (RGIConsortium, 2017) has a 30 (Meier and Post, 1969) to 60-year surge cycle (Wright, 1980; Driscoll, 1980). Altena et al. (2019) used optical satellite data to show its most recent surge initiated in 2014 and continued through 2017. The surge started mid-glacier and had two propagating fronts: a dominant surge front that propagated downglacier and a secondary subdued front that propagated upglacier. Our SAR-based record shows that surge activity terminated in mid-2018. The time series at P5 and P6 (Figures 11e,f, 12e,f) show complex flow dynamics in both the horizontal and vertical components.

The Walsh Glacier is another surge-type glacier with recent surge activity. Using optical Landsat data, Fu and Zhou (2020) showed the latest surge initiated before 2015 (Figure 9). Our SAR-based observations show residual surge activity continued into 2017 and abruptly ended in mid-2017 (Figure 9(d)). The time series at P8 (Figures 11h, 12h) show regular increases in flow velocity during summer, while at P7 these seasonal increases are less pronounced (Figures 11g, 12g). The surge at P7 during 2017 is a dominant signal.

A surge of the Kluane Glacier has previously been detected using RADARSAT-2 SAR measurements (Main et al., 2019). It occurred during 2018 in a secondary valley of the glacier (Figure 10). The entire surge cycle is captured by our time series (Figures 11i, 12i). Such a complex flow pattern can only be derived from side-looking SAR measurements that capture horizontal and vertical components of motion.

These six, in-depth analyzed glaciers were selected from the regional results shown in Figure 6. Other glaciers in this region may have also experienced surges or other interesting behaviours. The entire data set, which includes instantaneous velocities and cumulative displacements for each pixel, and the processing software are provided with this manuscript.

## 6 Conclusions

We presented a flow displacement technique to observe variations in glacier surface flow in 3D using ascending and descending SAR scenes. The 3D flow displacement (and/or velocity) time series computed allowed us to map in unprecedented detail the state and the temporal evolution of six glaciers in Southeast Alaska during 20 October 2016 - 21 January 2021. On a broader scale, this technique can be used for reconstructing the historic response of the worldwide glaciers to the warming climate using over 30 years of available satellite SAR records. The horizontal components can be resolved to study flow variations over time, and if integrated along a profile that is perpendicular to flow, ice flux. The vertical component can be used to assess changes in vertical ice flux or changes in surface slope over time, which is useful for studying glacier surge dynamics or variations in driving stress as a glacier dynamically adjusts to a changing climate. The software is freely available to the research community.

*Code and data availability.* The data and processing software used in this study can be downloaded from Mendeley Data http://dx.doi.org/... (will be added after the paper is accepted)

*Video supplement.* The animations of flow velocities for studied glaciers (files movie_malaspina.gif, movie_klutlan.gif, movie_walsh.gif, movie_kluane.gif) are provided. Comparisons between the magnitude of mean linear horizontal flow velocities along the four profiles with the results presented in Gardner et al. (2019) are also provided.

*Author contributions.* Sergey Samsonov - Conceptualization; Data curation; Formal analysis; Investigation; Methodology; Project administration; Resources; Software; Visualization; Writing - original draft; Writing - review & editing. Kristy Tiampo - Investigation; Formal analysis; Methodology; Writing - review & editing. Ryan Cassotto - Investigation; Formal analysis; Methodology; Writing - review & editing.

*Competing interests.* No competing interests are present.

*Acknowledgements.* We thank the European Space Agency for acquiring and the National Aeronautics and Space Administration (NASA) and ASF for distributing Sentinel-1 SAR data. Figures were plotted with GMT and Gnuplot software. Work of Sergey Samsonov was supported by the Canadian Space Agency through the Data Utilization and Application Plan (DUAP) program. The work of Kristy Tiampo was supported by CIRES, University of Colorado Boulder. The work of Ryan Cassotto was supported by NASA Grant No. 80NSSC17K0017.

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

**Table 1.** Sentinel-1 SAR data used in this study, where $\theta$ is incidence and $\phi$ is azimuth angles.

|  | Span | $\theta^\circ$ | $\phi^\circ$ | Number of SLC swaths |
|---|---|---|---|---|
| Sentinel-1 track 123 (asc) | 20160816-20210128 | 39 | 342 | 109 |
| Sentinel-1 track 116 (dsc) | 20161020-20210121 | 39 | 198 | 116 |
| Total (after boundary correction) | 20161020-20210121 |  |  | 223 |

**Table 2.** Correlation (first matrix in cell) and covariance (second matrix in cell) matrices of north, east, and vertical components of velocity for six synthetic tests shown in Fig 3. Indices a-f correspond to subfigures of Fig 3. Columns are in order - north, east, vertical. Units of covariance matrix terms are $(m/year)^2$

| a | | | | | |
|---|---|---|---|---|---|
| $\begin{pmatrix} 1.000 & -0.087 & -0.083 \\ -0.087 & 1.000 & -0.040 \\ -0.083 & -0.040 & 1.000 \end{pmatrix}$ | | | $\begin{pmatrix} 0.470 & -0.023 & -0.052 \\ -0.023 & 0.143 & -0.014 \\ -0.052 & -0.014 & 0.846 \end{pmatrix}$ | | |

| b | | | | | |
|---|---|---|---|---|---|
| $\begin{pmatrix} 1.000 & -0.084 & -0.081 \\ -0.084 & 1.000 & -0.080 \\ -0.081 & -0.080 & 1.000 \end{pmatrix}$ | | | $\begin{pmatrix} 0.465 & -0.022 & -0.051 \\ -0.022 & 0.151 & -0.029 \\ -0.051 & -0.029 & 0.847 \end{pmatrix}$ | | |

| c | | | | | |
|---|---|---|---|---|---|
| $\begin{pmatrix} 1.000 & -0.069 & -0.073 \\ -0.069 & 1.000 & -0.140 \\ -0.073 & -0.140 & 1.000 \end{pmatrix}$ | | | $\begin{pmatrix} 0.486 & -0.021 & -0.047 \\ -0.021 & 0.201 & -0.059 \\ -0.047 & -0.059 & 0.877 \end{pmatrix}$ | | |

| d | | | | | |
|---|---|---|---|---|---|
| $\begin{pmatrix} 1.000 & 0.999 & -0.997 \\ 0.999 & 1.000 & -0.995 \\ -0.997 & -0.995 & 1.000 \end{pmatrix}$ | | | $\begin{pmatrix} 0.581 & 0.287 & -0.145 \\ 0.287 & 0.142 & -0.072 \\ -0.145 & -0.072 & 0.036 \end{pmatrix}$ | | |

| e | | | | | |
|---|---|---|---|---|---|
| $\begin{pmatrix} 1.000 & 0.975 & -0.937 \\ 0.975 & 1.000 & -0.923 \\ -0.937 & -0.923 & 1.000 \end{pmatrix}$ | | | $\begin{pmatrix} 0.594 & 0.290 & -0.151 \\ 0.290 & 0.149 & -0.075 \\ -0.151 & -0.075 & 0.044 \end{pmatrix}$ | | |

| f | | | | | |
|---|---|---|---|---|---|
| $\begin{pmatrix} 1.000 & 0.824 & -0.681 \\ 0.824 & 1.000 & -0.618 \\ -0.681 & -0.618 & 1.000 \end{pmatrix}$ | | | $\begin{pmatrix} 0.653 & 0.297 & -0.162 \\ 0.297 & 0.199 & -0.081 \\ -0.162 & -0.081 & 0.087 \end{pmatrix}$ | | |

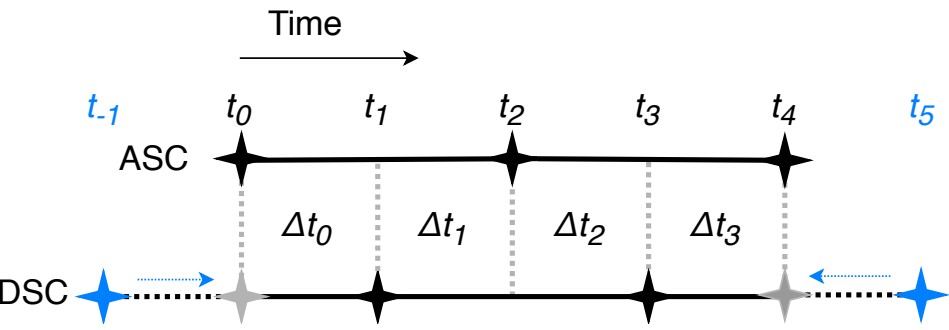

**Figure 1.** Schematics of simplified case described by (3). Ascending and descending SAR acquisitions at time $t_i$ are marked with black stars. Horizontal solid lines represent range and azimuth offset maps. Vertical dashed lines divide temporal scale in time intervals $\Delta t_i = t_{i+1} - t_i$ between consecutive acquisitions. Time of first and last descending acquisitions (marked with blue stars) are adjusted to match first and last time of ascending acquisitions (marked with gray stars).

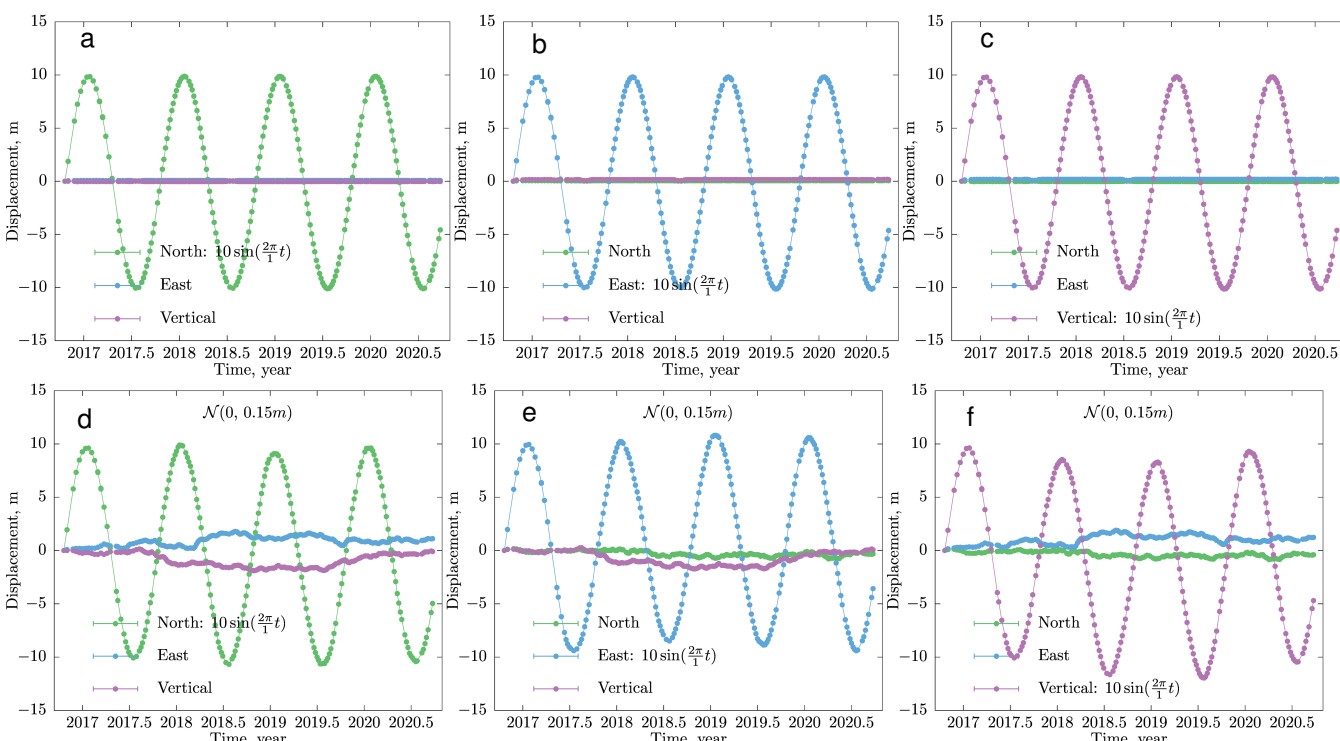

**Figure 2.** Results of numerical simulations demonstrating ability of this technique to reconstruct input signal in one of components. Equations of input signals are shown in corresponding subfigure legends where $t$ is time. Harmonic input signals are assumed. Gaussian noise with mean value of zero and standard deviations of 0.15 m (which is approximately 10% of signal) is added to subfigures in second row.

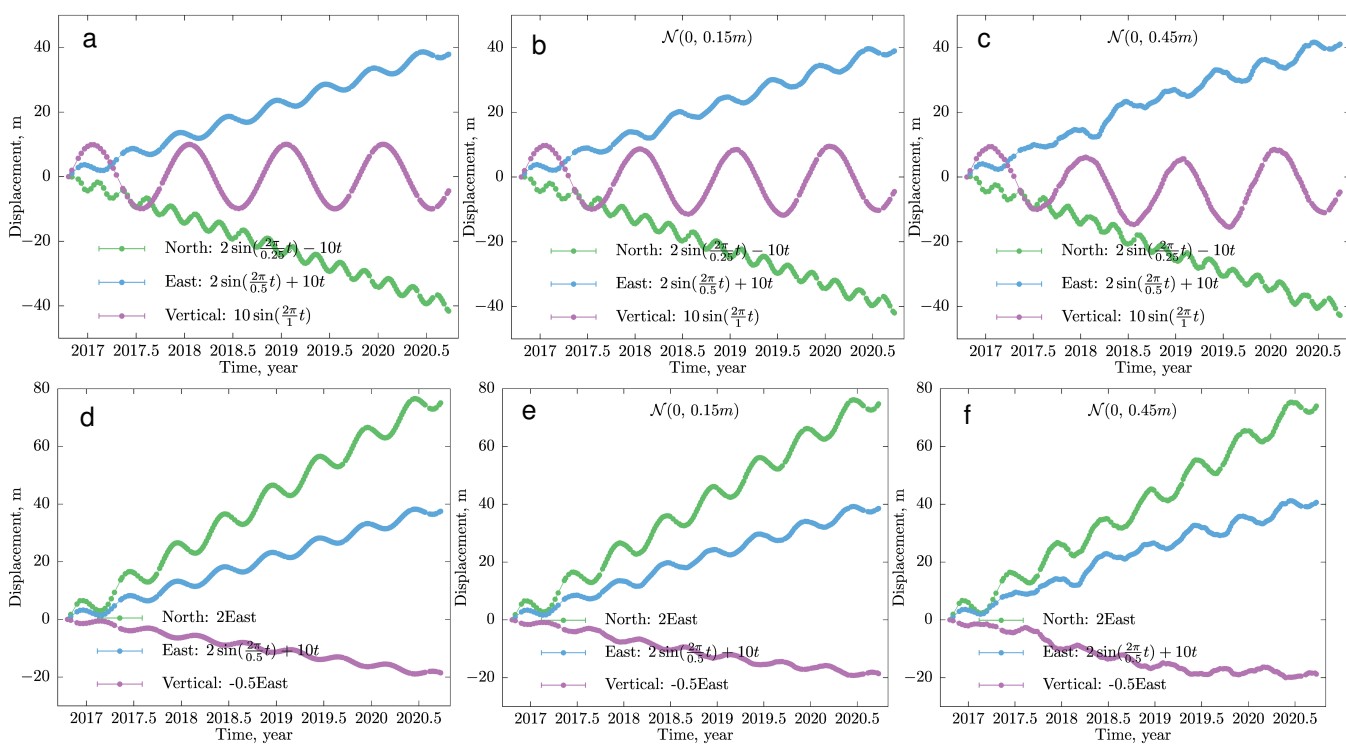

**Figure 3.** Results of numerical simulations demonstrating ability of this technique to reconstruct complex uncorrelated and correlated input signals in all three components. Equations of input signals are shown in corresponding subfigure legends where $t$ is time. Harmonic and linear input signals are assumed. Gaussian noise with mean value of zero and standard deviations in range 0.15-0.45 m (which is approximately 10-30% of signal) is added to subfigures in second and third columns.

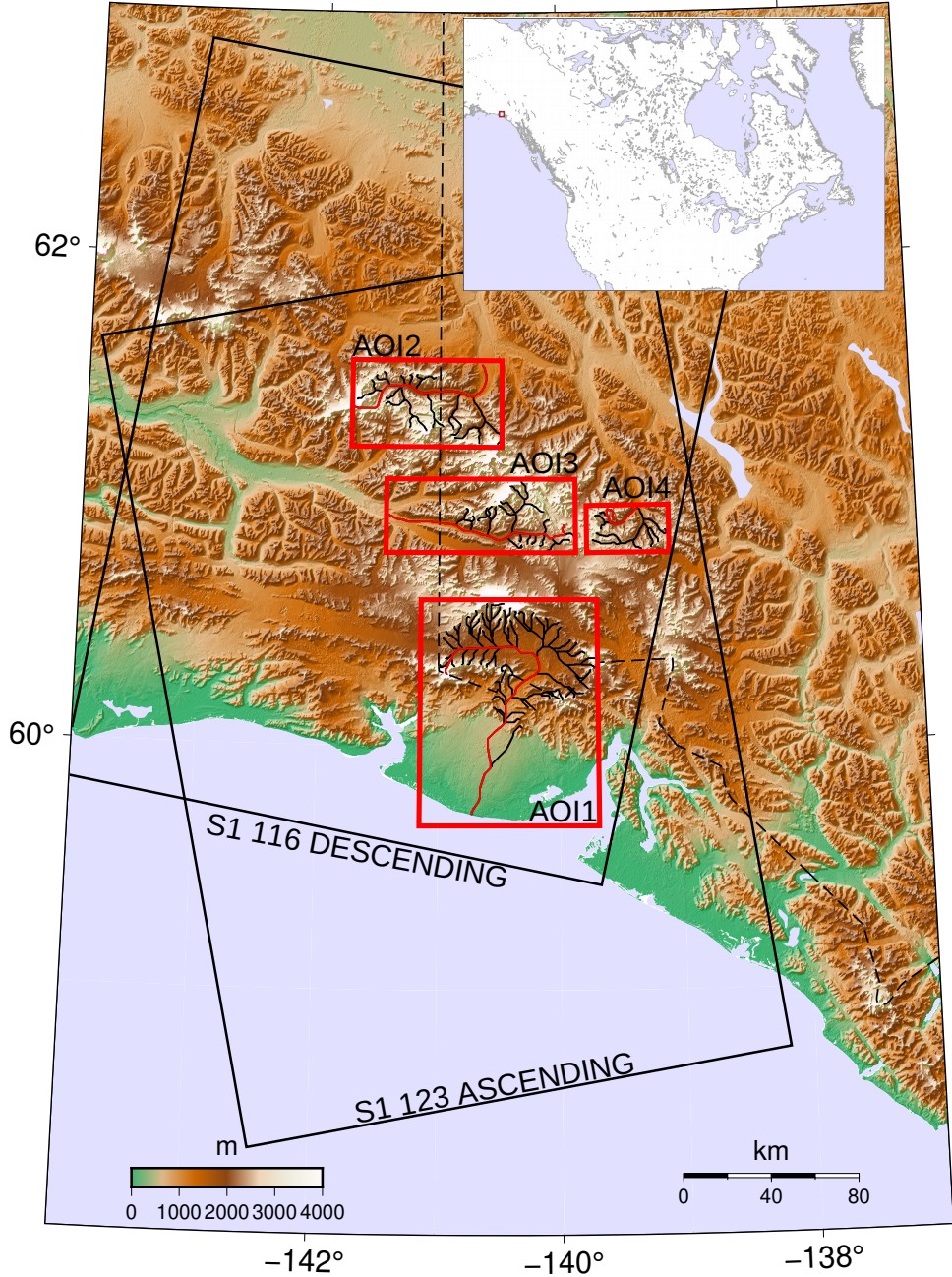

**Figure 4.** Outlines of four areas of interest (AOIs) in southeastern Alaska are shown in red. AOI1 covers Agassiz (AG), Malaspina (MG) and Seward (SG) Glaciers. AOI2 covers Klutlan Glacier (KG). AOI3 covers Walsh Glacier (WG). AOI4 covers Kluane Glacier. Flow lines in black and red were computed using Open Global Glacier Model (OGGM) software (Maussion et al., 2019). Outlines of ascending (track 123) and descending (track 116) Sentinel-1 swaths are shown in black. Background is 30 m Advanced Spaceborne Thermal Emission and Reflection Radiometer (ASTER) digital elevation model (Abrams et al., 2020). Canada-USA border is shown as dashed black line.

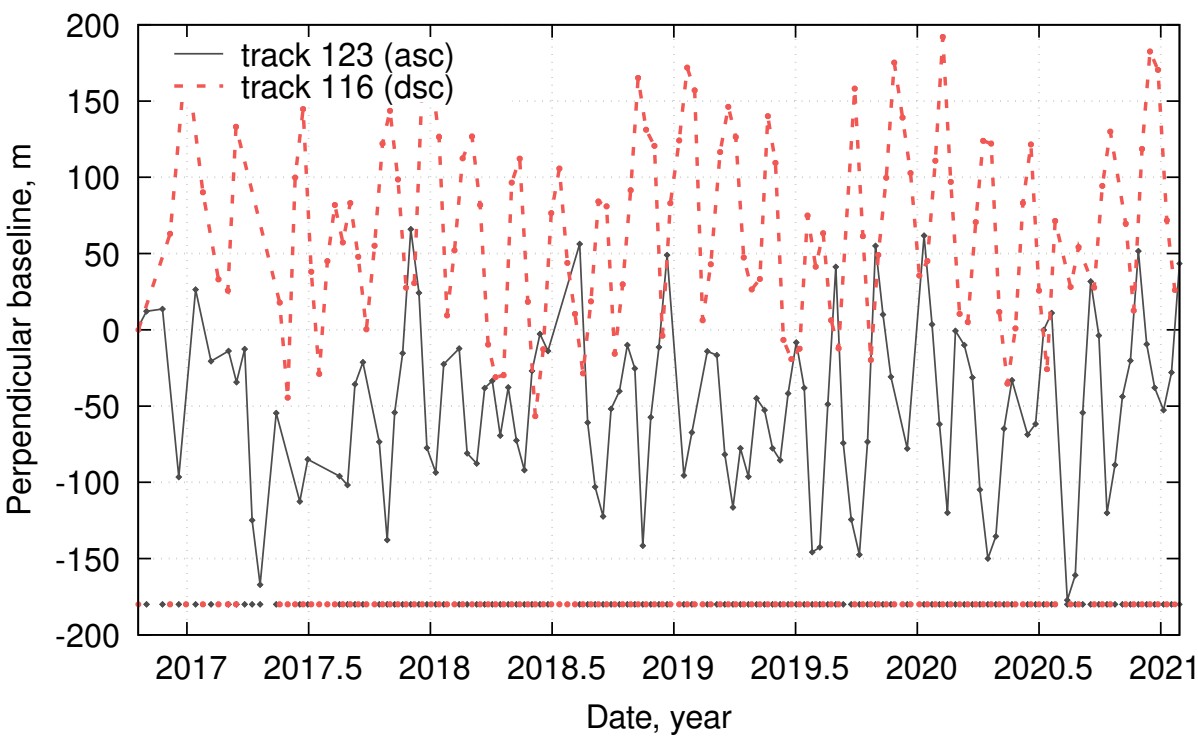

**Figure 5.** Spatial and temporal baselines of Sentinel-1 pairs used in this study. Mean temporal resolution, i.e. mean temporal spacing between consecutive SAR acquisition regardless of orbit direction, computed as duration divided by number of SAR images (4.25 years*365/223) is about 7 days. Note that offset between ascending and descending sets depends on selection of reference images, which is arbitrary.

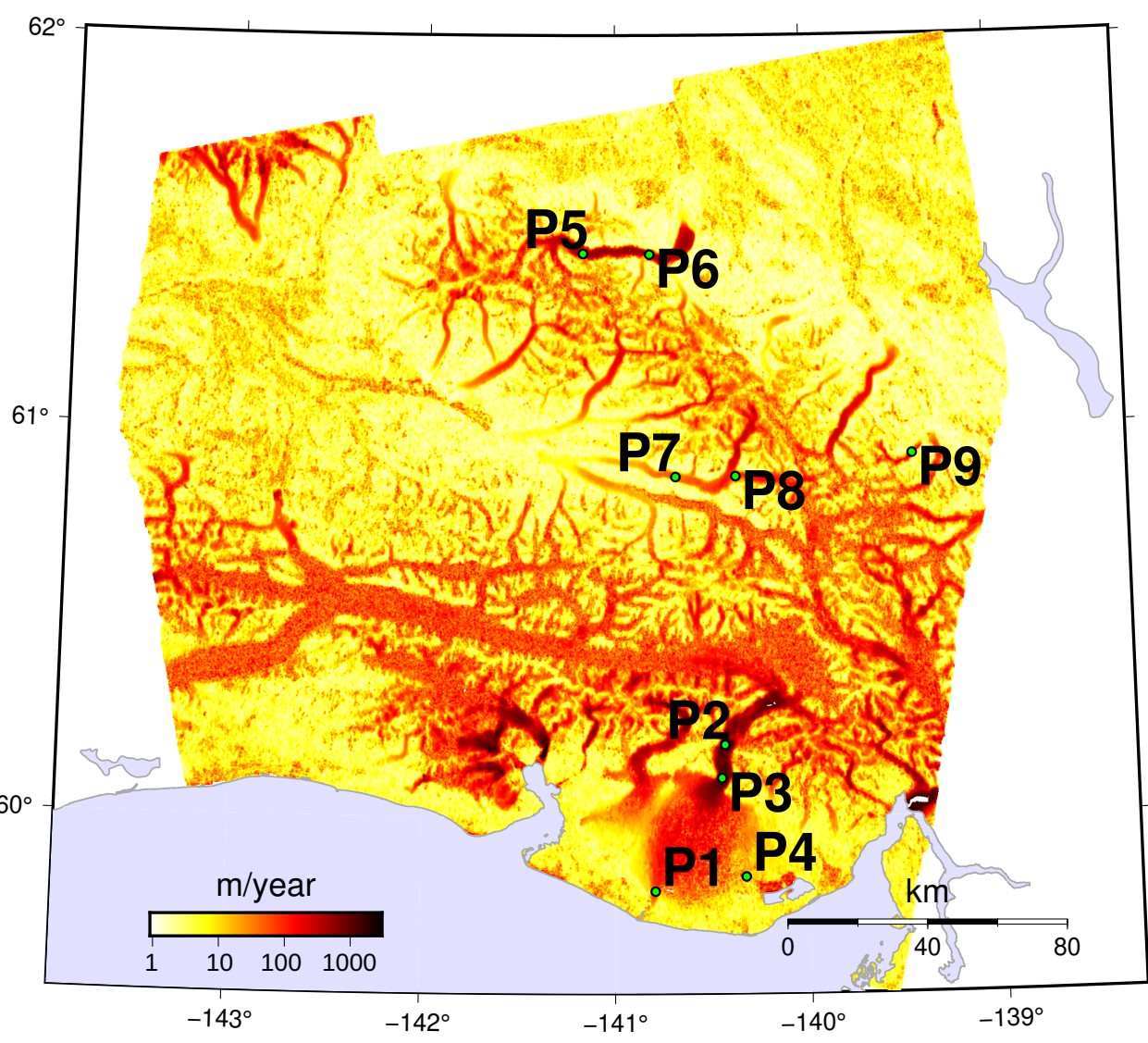

**Figure 6.** Magnitude of mean 3D flow velocities plotted using logarithmic scale. For regions P1-P4 at Malaspina/Seward Glaciers, P5-P6 at Klutlan Glacier, P7-P8 at Walsh Glacier and P9 and Kluane Glacier time series are provided in Figures 11-12.

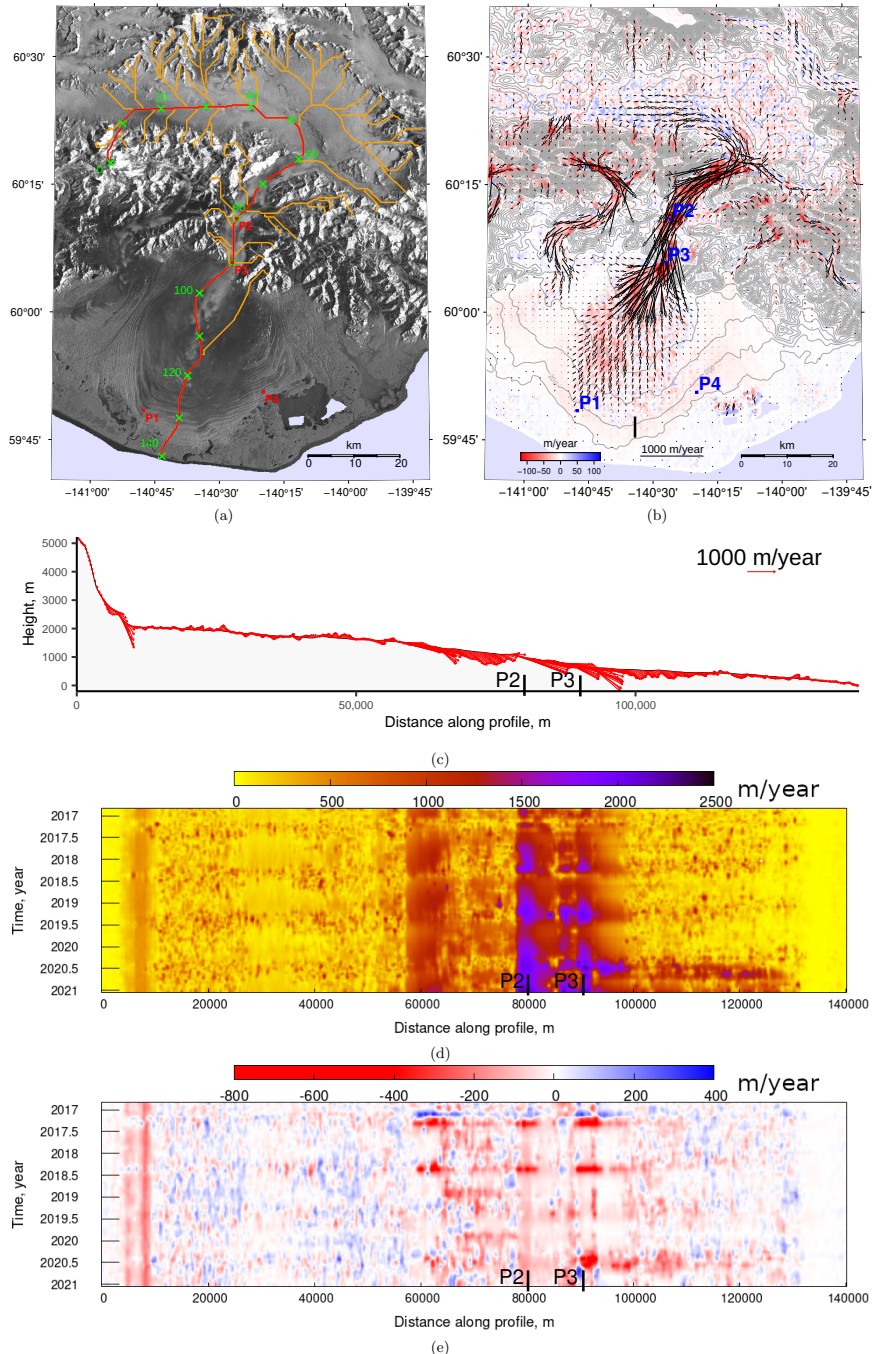

**Figure 7.** (a) Sentinel-1 SAR intensity image acquired on 20191222 (in YYYYMMDD format) over AOI1 that covers Agassiz (AG), Malaspina (MG) and Seward (SG) Glaciers. Flowlines are in orange and red. Markers in green show distance in kilometres along selected in red flowline. (b) Time-averaged 3D flow velocities: horizontal velocity is shown as (coarse-resolution) vector map and vertical velocity is colour-coded. Surface topographic contour lines derived from TerraSAR-x 90 m DEM with intervals of 100 m are shown in grey. Flow displacement time series for regions P1-P4 are plotted in Figure 11. (c) Time-averaged 3D flow velocities and glacier height along red flowline. (d) Temporal evolution of horizontal flow velocity magnitude along red flowline. (e) Temporal evolution of vertical flow velocity along red flowline.

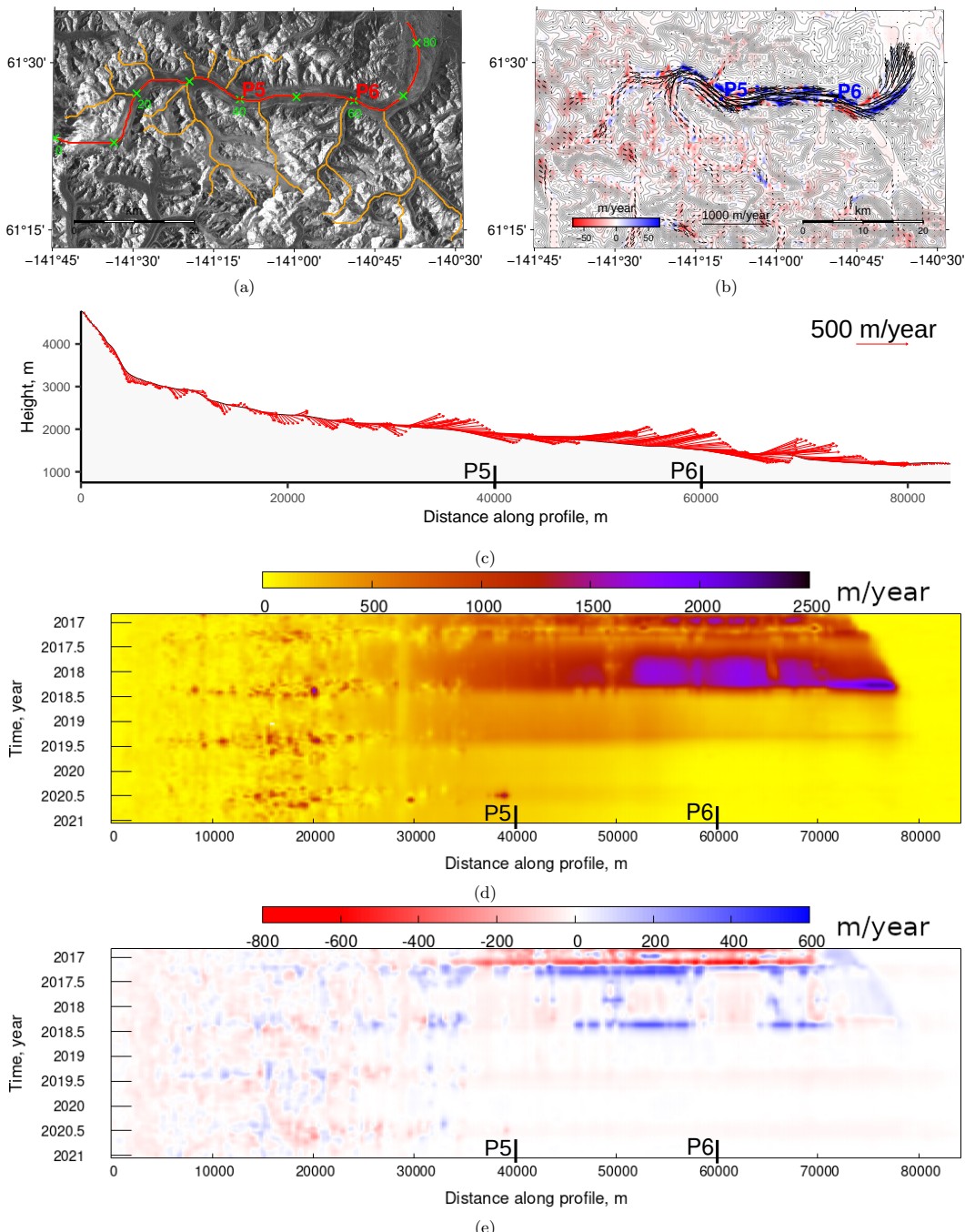

**Figure 8.** (a) Sentinel-1 SAR intensity image acquired on 20191222 over AOI2 that covers Klutlan Glacier (KtG). Flowlines are in orange and red. Markers in green show distance in kilometres along selected in red flowline. (b) Time-averaged 3D flow velocities: horizontal velocity is shown as (coarse-resolution) vector map and vertical velocity is colour-coded. Surface topographic contour lines derived from TerraSAR-x 90 m DEM with intervals of 100 m are shown in grey. Flow displacement time series for regions P5-P6 are plotted in Figure 11. (c) Time-averaged 3D flow velocities and glacier height along red flowline. (d) Temporal evolution of horizontal flow velocity magnitude along red flowline. (e) Temporal evolution of vertical flow velocity along red flowline.

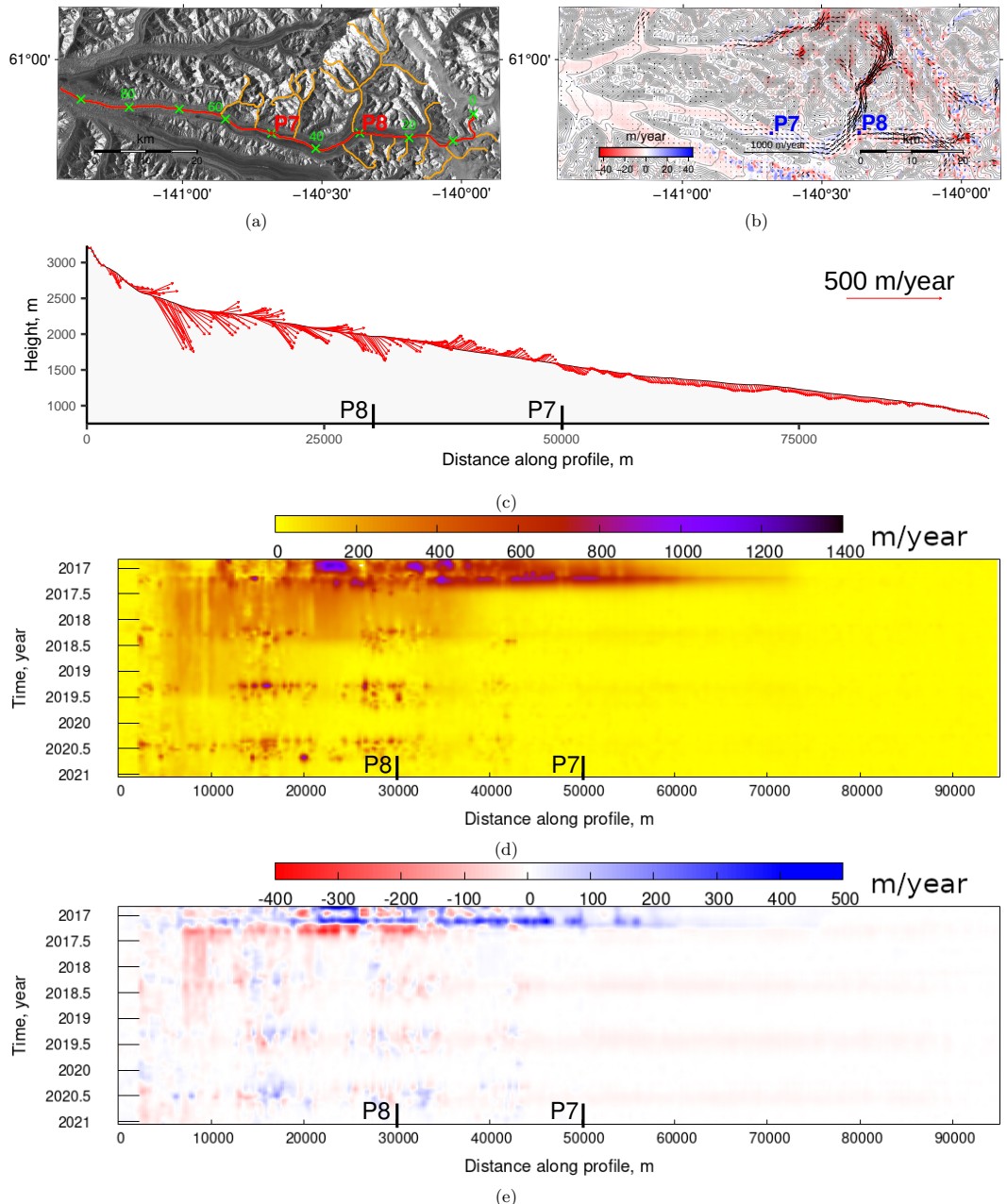

**Figure 9.** (a) Sentinel-1 SAR intensity image acquired on 20191222 over AOI3 that covers Walsh Glacier (WG). Flowlines are in orange and red. Markers in green show distance in kilometres along selected in red flowline. (b) Time-averaged 3D flow velocities: horizontal velocity is shown as (coarse-resolution) vector map and vertical velocity is colour-coded. Surface topographic contour lines derived from TerraSAR-x 90 m DEM with intervals of 100 m are shown in grey. Flow displacement time series for regions P7-P8 are plotted in Figure 11. (c) Time-averaged 3D flow velocities and glacier height along red flowline. (d) Temporal evolution of horizontal flow velocity magnitude along red flowline. (e) Temporal evolution of vertical flow velocity along red flowline.

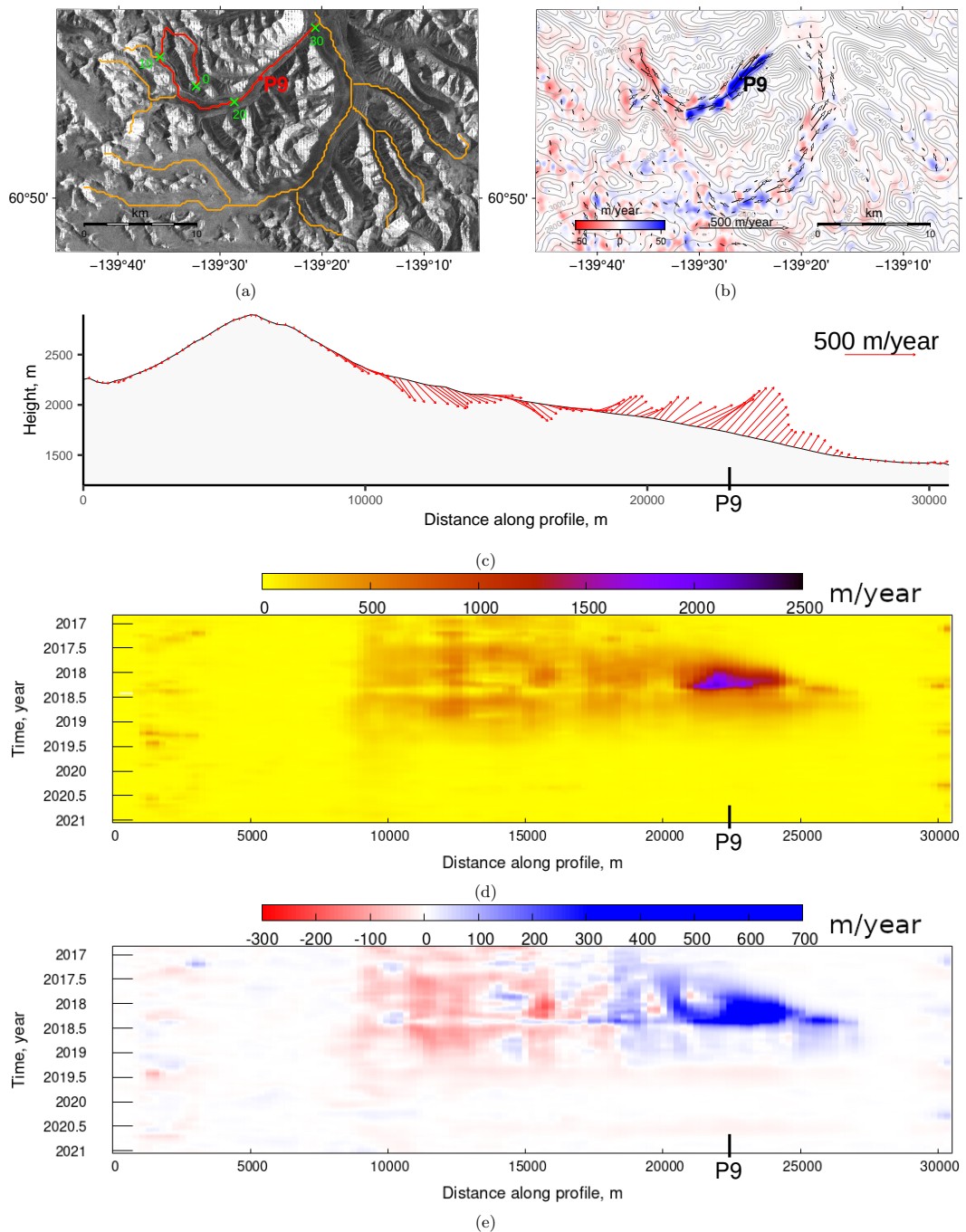

**Figure 10.** (a) Sentinel-1 SAR intensity image acquired on 20191222 over AOI4 that covers Kluane Glacier (KnG). Flowlines are in orange and red. Markers in green show distance in kilometres along selected in red flowline. (b) Time-averaged 3D flow velocities: horizontal velocity is shown as (coarse-resolution) vector map and vertical velocity is colour-coded. Surface topographic contour lines derived from TerraSAR-x 90 m DEM with intervals of 100 m are shown in grey. Flow displacement time series for region P9 are plotted in Figure 11. (c) Time-averaged 3D flow velocities and glacier height along red flowline. (d) Temporal evolution of horizontal flow velocity magnitude along red flowline. (e) Temporal evolution of vertical flow velocity along red flowline.

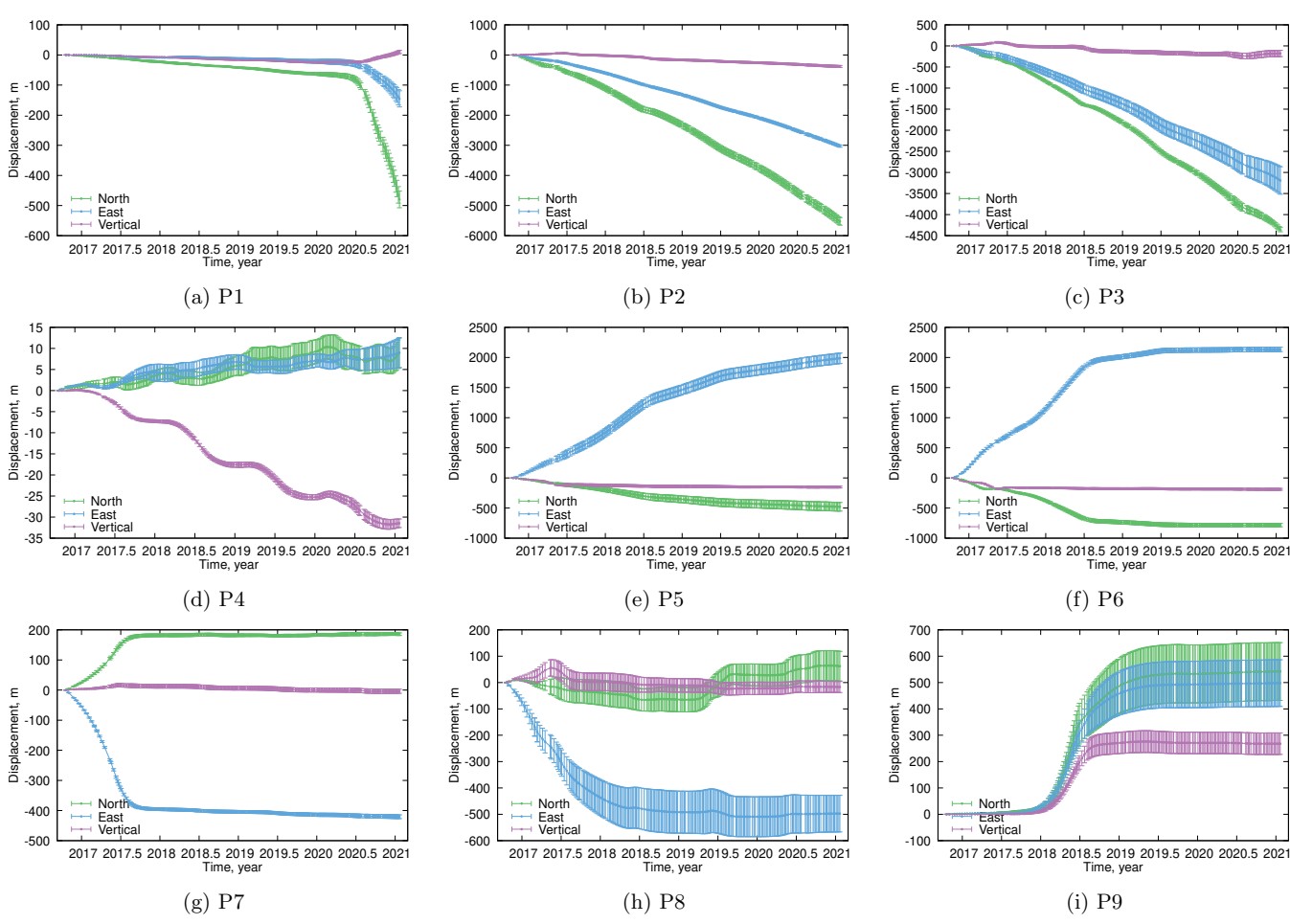

**Figure 11.** 3D flow displacement time series for regions P1-P9, which locations are shown in Figures 6-10.

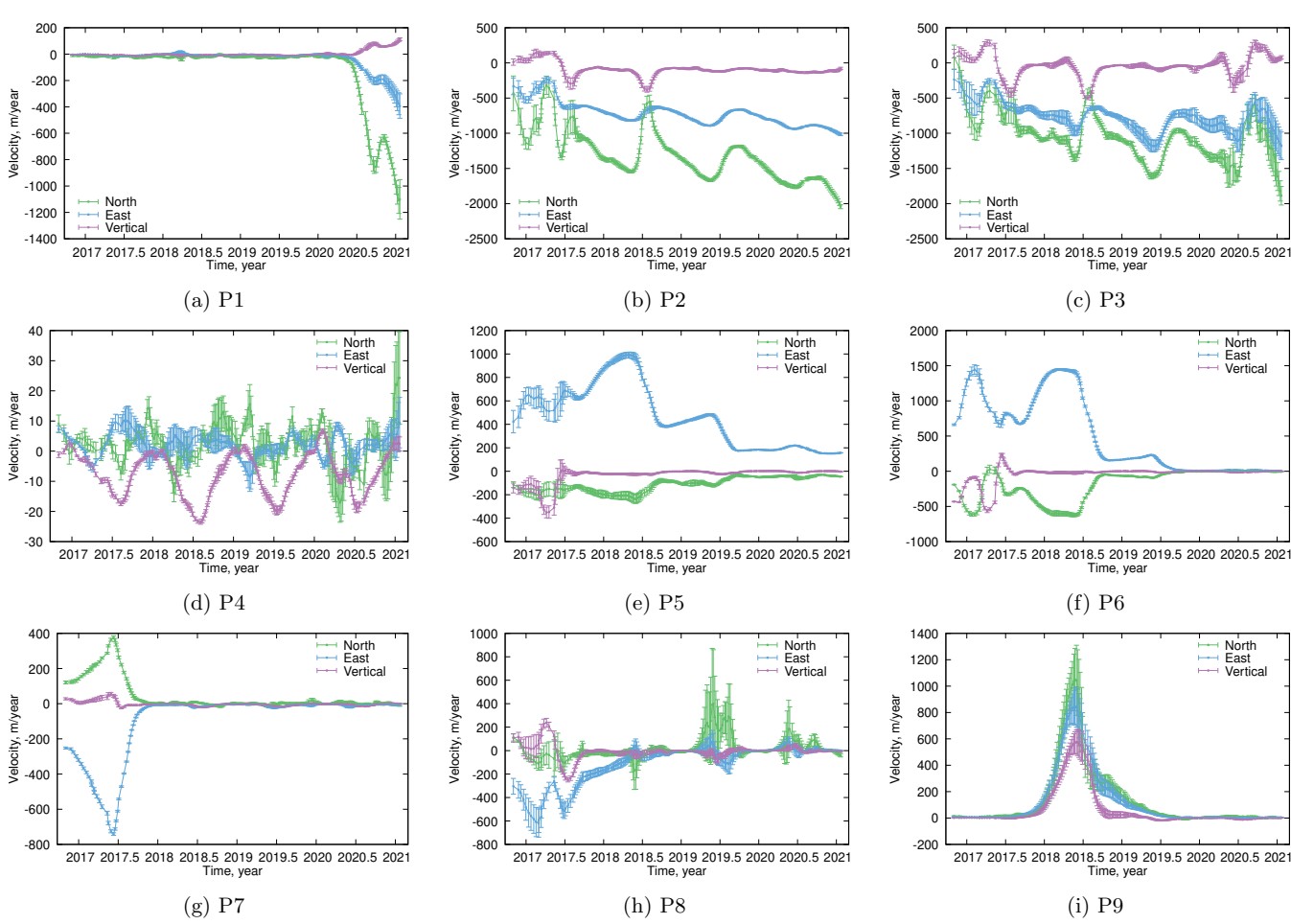

**Figure 12.** 3D flow velocity time series for regions P1-P9, which locations are shown in Figures 6-10.