# Peer review of "Measuring the state and temporal evolution of glaciers in Alaska and Yukon using SAR-derived 3D time series of glacier surface flow"

_The Cryosphere, 2020_

## Referee Comment (RC1) · Anonymous Referee #1 · 16 Nov 2020

Review of Measuring the state and temporal evolution of glaciers using SAR-derived 3D time series of glacier surface flow

By Sergey Samsonov, Kristy Tiampo, and Ryan Cassotto

This paper present a technique for producing times series of the 3D glacier motion using ascending/descending data.

Overall it's a good paper, and the technique seems sound. That said, though I would like to see a better technical discussion with respect to errors and temporal/spatial resolution as noted below. The discussion needs some work make it clear what the data is actually showing, especially with respect to what is the source of the vertical

displacement (see below).

General points I don't see much discussion of errors. Typically, with 6-day sampling and azimuth offsets, you are going to get about errors of about 20-m/yr. The vertical velocities are more driven by range offsets, but the time everything is solved for, some of those errors are going to fold into the vertical solution.

Vertical motion is a combination of the vertical motion due to surface parallel flow other factors (e.g., submergence/emergence velocity or subsidence/inflation due to subglacial water flow). In general, the surface parallel vertical displacement will be the dominant term and it will vary with the horizontal speed, especially for mountain glaciers with relatively steep slopes. The other forms of vertical displacement are the more interesting terms though. So, it would make sense to compute the surface parallel component and remove it to isolate the other types of vertical motion. Otherwise statements like "The predominately downward flow of ice observed throughout the Malaspina Glacier's massive lobe (Figure 4b.c) indicates that ablation rates have exceeded emergence velocities during our 4 year study period, implying" are potentially incorrect or at least presented without proper context. I have no reason not to believe the glacier is wasting down faster than the emergence velocity, but in general glaciers flow downhill, so that could largely due to the surface parallel flow component. Moreover, from image to image, the glacier is largely measuring the same coherent patch of speckle, which would record motion of the surface due to ice dynamics. It should not be measuring downward motion of the surface due to direct ablation (unless you phase of the ablated layer is accounted for, which does not seem to be happening here). In other words, this is like measuring the downward motion of a GPS on a pole sunk in the ice, which will not measure ablation, vs a GPS placed on the surface, which will measure ablation. Distinctions such as these need to made in the discussion.

There is some discussion about penetration effects, but it is import to note you can get some really strange offset patterns when you have soaked firn, which can be spatially coherent over large distances (can map into errors of several hundred m/yr). I am not
sure that some of what's being seen is this kind of effect (e.g. blue patches Fig. 4b).

Point, P2, seems to exhibit a seasonal cycle not seen in the other data. It also happens to be near a marginal lake. I wonder if pressure variations as the lake fills and drains are contributing to the seasonal up and down motion. Some discussion as to why this point has a strong seasonal signal would be good.

Specific points Line 48: "However, the SPF constraint is only applicable to glaciers in steady state." This statement is not correct, the SPF assumption ignore the submergence and emergence velocity and other vertical motion, which is true whether or not the glacier is in steady state. And if the glacier is not in steady state, it will still measure vertical velocity variations that are parallel to the surface.

Line 84: A 256x256 sampling window (both patches???) will provides about 3.5km resolution (256 \* 13.9 m azimuth resolution), which is further degraded by a 2-km median filter. In addition to the lack of resolution, this can cause problems where the matcher will lock on stationary rock areas more easily than the glacier to report zero velocities a km or 2 inboard of the margin. How is this dealt with. There should be some discussion of what the spatial resolution is. Certainly, the ground resolution is not 200-m as stated, even if the data posted are at 200 m.

Line 85: What corrections were applied for baseline and to calibrate the data (e.g., were control points used to remove biases).

Equation 1 – please break separate into two equations (put the cumulative as a different equation).

117 "temporal smoothing" What is the temporal resolution after regularization.

Line 224-226. I would like to see the surface parallel flow components removed before seeing discussion about kinematic waves.

Figures. The x-axis of the vector profile plots could be lined up with the color times series plots. The vector plots while pretty, don't really give a good idea of the magnitudes

TCD
of the vertical velocity. Please show the profiles also on the b panels since that actually shows magnitudes of vertical motion. Also use some kind of symbol on the c and d plots to indicate where the points PX are.

Would be helpful to see the color panels broken out separately as horizontal and vertical magnitudes. The alternating patches of slow and fast flow are strange.

TCD

---

## Referee Comment (RC2) · Brent Minchew (Referee) · 30 Dec 2020

Review of "Measuring the state and temporal evolution of glaciers using SAR-derived 3D time series of glacier surface flow" by Samsonov, Tiampo, and Cassotto

**Summary**

The authors discuss a method for inferring time-dependent 3D surface velocity fields from synthetic aperture radar (SAR) data and apply this method to data collected from Sentinel I in 2016-2020 over five outlet glaciers in Alaska: Agassiz, Seward, Malaspina, Klutlan, and Walsh. Their results show complex glacier flow fields and temporal variations, and capture a host of interesting phenomena, including seasonal variations in ice flow, a surge, and dynamical glacier states. The authors present a series of figures showing the resulting velocity fields and report on their results and a few possible implications.

The manuscript is in line with a growing area of research that has great promise to advance our understanding of the cryosphere due to the volume of information available from modern remote sensing platforms. The authors have described a potentially useful method and chosen an interesting study area. As such, this work may be of interest to the TC readership.

However, as discussed below, the paper needs a lot of work in terms of organization, presentation of the methods, and analysis of the results before it can be considered acceptable for publication in the scientific literature. Indeed, I found the paper to be frustrating to read for a variety of reasons, perhaps the main reason being that the authors seem to be trying to claim levels of success, novelty, and generality that their study does not merit rather than taking a measured approach to demonstrate that their method works, inform readers of the merits of their method/study, and to connect their work to the scholarly literature. The authors almost completely ignore the existing body of work on time-dependent 3D surface velocity fields (which their citations suggest they are aware of), with the exception of references to their own papers and a couple of passing references to a paper (Guo et al., 2020) that presents what appears to be the exact same method the authors are presenting here. As a result, the authors do not place their study into the proper context, nor do they give readers the ability to compare the strengths and weaknesses of the authors' method and those of other methods. But most importantly, the authors do not demonstrate an awareness of the documented challenges of inferring 3D time-dependent velocity fields from satellite data and thus they ignore any consideration of accuracy and precision in the measurements, mixing between the inferred velocity components, and propagation of measurement errors. Perhaps because of this oversight, the authors make claims about the viability of their method that are unsupported by the work presented in the manuscript. **Nowhere do the authors test their technique nor make any meaningful attempt to show that their method actually works**; rather, we get a few basic equations, some results from actual SAR data (which can only show that the matrix in the authors' method is invertible but cannot show that there is sufficient information to make the inferences the authors are trying to make), and then unsupported claims like (line 164) "[t]he technique in this study is a viable solution for computing 3D flow displacement time series…" The authors never discuss errors nor the limitations and challenges of their method, inferences of multidimensional flow velocities and other such matters that one would expect to find in a scientific publication. In addition, the presented analysis of the results lacks the depth and detail needed to provide new insight into the glaciers being studied or glacier dynamics in

general. The current manuscript is very short compared with the vastness and richness of the material the authors are trying to cover, and the supplementary material contains only a single figure, so the authors have plenty of space in the main text and supplement to expand on their methods and findings. More details are provided below.

**Major points:**
- The title and abstract of the paper suggest the main goal of this manuscript is to present a general method for inferring time-dependent 3D surface velocity fields of glaciers. But the methods are presented as an incremental step from previous work and not described in sufficient detail to merit publication based on the methods alone. Key information that readers need to understand the method and reproduce the results are missing. A few specific points:
  - No meaningful validation of the method is provided. In my reading, I did not find any evidence that the method produces accurate 3D velocity fields. I only found reference to one comparison between the authors' results and independent measurements taken over vastly different time scales (Gardner et al., 2018, 2019). This comparison is given in the figure in the supplement (Fig. S1). I would argue that the authors' results differ markedly in all cases from the measurements of Gardner et al., 2018, 2019. Even on the glaciers where the authors claim 'nearly identical results' there are clearly significant disparities (100s of meters per year, or roughly a factor of 2) between the data sets. The authors provide some plausible suppositions to explain these disparities but never explore these possibilities. Rather, the authors give us offhand references to filtering and other technical matters that can and should be tested and some discussion of how the flow of glaciers is expected to differ between the more recent (2016-2020) observations made by the authors and the multi-decadal average of Gardner et al., 2018, 2019. Indeed, the comparisons between the authors' velocity fields and those of Gardner et al., 2018, 2019, especially over surging glaciers, are scientifically interesting but are not viable tests of the methods presented here. If the authors wish to present a new method, especially one that is as generally applicable as they claim, they need to conduct multiple appropriate tests on synthetic data to test their method under the conditions expected in the natural environment and to convincingly show readers that their method reliably produces accurate results. I cannot stress enough that the authors do none of this work in the current manuscript.
  - The authors need to provide some discussion of the effects of the viewing geometry on the inferred results. While range and azimuth offsets are orthogonal to one another in existing SAR systems (where the radar line of sight is orthogonal to the platform velocity vector by design), ascending and descending orbits are not orthogonal (as shown in Fig. 1). Thus, the viewing geometries are nonideal and the relative orientation of the orbits and the flow direction of the glacier influences the precision of the inferred velocity components. This geometric effect is amplified by noise in the measurements, particularly the disparity in noise between range and azimuth offsets (where

range offsets generally have higher signal-to-noise ratios than azimuth offsets). These effects are discussed to some extent in Minchew et al., 2015, 2017, though the basic ideas are well known from GPS and should be given ample consideration in work of the type the authors are presenting.

o Given the orientation of satellite orbits, I expect that there are strong covariances between the inferred vertical and horizontal components of flow. This effect is likely to be most pronounced on Klutlan Glacier, whose flow direction is close to the line-of-sight direction of the radar, meaning that the range offsets pick up most of the horizontal motion and all of the vertical motion while the azimuth offsets provide relatively little constraints. The covariances between horizontal and vertical velocity components are likely to be lowest on Malaspina Glacier, which is flowing more or less south, in a direction that is close to the azimuth direction of the SAR data. This orientation is favorable to inferring 3D velocity fields as the azimuth offsets (which are purely horizontal) are doing most of the work to constrain the horizontal velocity while the range offsets provide information on vertical velocity with little direct influence from the horizontal components. More generally, it is worth noting that the authors' results seem to show that the vertical component of velocity is largest in areas of the glacier that are flowing more along latitude (i.e., east/west flow direction, which is close to alignment with the radar line of sight) than along longitude (which is close to being aligned with the orbits, or azimuth direction), which suggests covariance between the horizontal and vertical components of velocity in the east/west trending flow. Again, all of these topics are discussed in some detail in Minchew et al., 2015, 2017. The take-away is that the authors need to quantify and discuss the geometric and measurement errors for their methods to be publishable and to support any claim of generality.

o There is no discussion of errors. The authors provide error bars on the results they present in Fig. 7 but these are merely spatial variances. I would expect a modern paper on geodetic methods to discuss formal errors in the SAR offset fields, how formal errors in the SAR measurements are propagated to the inferred 3D fields, how viewing geometry impacts the results (as just discussed), and any sources of additional error that may not be accounted for in the formal error (e.g., the influences of radar penetration depth and surface moisture, as discussed by Minchew et al., 2015). The lack of treatment of errors is a considerable omission from a methods-focused paper that should be rectified before publication.

o The authors need to provide a reference to the form of the Tikhonov regularization matrix of various orders, some convincing evidence that regularization "is not critical" in this case (cf. line 119), and some discussion of why regularization doesn't make a difference in this case and the conditions under which regularization should matter. Finally, the authors need to be clear how regularization is being applied: are the authors regularizing in space or time or both? Eq. 3 suggests that regularization is only applied in time but this should be clarified in the paper.

o   It would seem that the authors are filtering in space using median filter with a window size larger than the width of some of the glaciers (line 85). A discussion of how this filter and window size were chosen and the effects of a nonlinear (median) filter and such a heavy filter on the final results would be useful.

- The authors need to discuss the existing literature and place their methods into proper context. Methods for inferring 3D, time-dependent surface velocity fields of glaciers (e.g., Minchew et al., 2015, 2017; Milillo et al., 2017; Guo et al., 2020) and generalized frameworks for inferring multi-dimensional surface velocity time-series (e.g., Greene et al, 2020; Riel et al., 2020) have been published. Only two of these papers are cited in the current manuscript and neither of these are discussed in any meaningful detail. In particular, it appears as though the methods of Milillo et al., 2017, and Guo et al., 2020, are strikingly similar to those presented by the authors in this work. Certainly, they all take the same basic approach of using the small-baseline subset (SBAS) method extended to multiple dimensions.

  This similarity between Milillo et al. (2017), Guo et al. (2020), and the current work needs to be properly explored in the manuscript. In my own reading of Guo et al., I cannot find any real difference between their approaches and those presented here. Indeed, the current authors state on line 170 "[o]ur approach is conceptually similar to the technique of Guo et al. (2020)…[h]owever, our software can additionally compute 1D, 2D…, and 3D flow velocities and displacement time-series and linear rates." It is important here to distinguish methods (and the ideas behind them) from implementation (software and tools); it appears that the authors' claim to novelty is in the implementation and the additional features of their software rather than differences in methodology. In other words, their software bundles several methods that are discussed in previous publications into a single user interface. I find this confusing because the paper seems to be about methodology, which is separate from implementation. I respect the fact that the authors have been working independently on this method for some time and that they should have their efforts rewarded by having the opportunity to publish their work in the scholarly literature. But more needs to be done to clarify the differences (if any) between the method presented here and those in the existing literature. If there are no meaningful differences, the authors should make that clear and emphasize the unique aspects of their work and results. Thorough testing of the method, exploration of precision and geometric effects, and robust uncertainty quantification (all discussed above) would make this paper unique from Guo et al. and would add value to the authors' methods.

  In contrast to the studies mentioned in the previous paragraph, Minchew et al., (2017), take a markedly different approach to inferring time-dependent 3D velocity fields. I note that this paper is cited in the current manuscript but simply in a list of other papers that apply InSAR to glacier flow; the fact that Minchew et al. (2017) present a method for inferring 3D, time-dependent velocity fields is ignored by the authors, as are the lessons learned about the challenges of accurately inferring 3D, time-dependent velocity fields from SAR data (as discussed above). It would be useful to compare the approach of Minchew et al. (2017) to the authors' approach in an insightful way as the two approaches are quite different. For example, the authors could note that Minchew

et al. assume a form for the temporal basis functions based on prior knowledge of the study area, while the current authors invert a matrix for displacement at a given time. The need for prior knowledge in the Minchew et al approach means that this method is not general and so its application is limited to areas where the assumed basis functions should be valid. But the advantage of the Minchew et al approach is interpretability of the results, straightforward connection of the results to the physics of the systems being observed, and robust uncertainly quantification, all things that are lacking in SBAS-based methods like those presented here.

A recent improvement to the work of Minchew et al. (2017) is Riel et al., 2020. I won't fault the authors for not discussing Riel et al. (2020) in the current manuscript as this is very recent work, but it would be good (though, not required) to include a brief discussion of Riel et al. (2020) in the revised draft to add context to the authors' work. Riel et al., 2020, adopt some of the methods of Riel et al., 2014, 2018, and apply them to remote sensing observations of glaciers. From a methodological perspective, this has the effect of generalizing the approach of Minchew et al. (2017) to allow for a generic set of temporal basis functions, from which a sparsity-inducing optimization is used to identify the simplest set of basis functions that describe the data. Here again, the main advantage of this approach is interpretability of the results (and robust uncertainty quantification), which provides the ability to decompose the observed signal into short and long-term variations, and features to ability to constrain transients, secular, and periodic signals. There are certainly limitations to the Riel et al (2020) method, namely that it still requires some level of prior knowledge to provide confidence in the resulting basis functions. The authors' methods may be complementary in the sense that they do not rely on basis functions. Again, the authors' method provides flexibility at the expense of interpretability of the results, where was the Minchew et al. and Riel et al. approaches sacrifice flexibility in the method for enhanced interpretability of the results. An appropriate exploration of these differences in approaches will provide readers with insight into the respective strengths and weaknesses so that they can make informed decisions about which methods may best suit their needs and where improvements can be made.

Riel et al (2020) take the time-dependent methodology further by introducing methods to quantify the propagation of waves through glaciers in the case of generalized methods (Minchew et al., 2017, quantify wave propagation but this is implicit and straightforward in the periodic basis functions). Importantly, Riel et al (2020) are able to track waves of different frequencies (because separation of frequencies is inherent in the time-series methods) and are able to show that waves with seasonal and annual periods on Jakobshavn Isbræ, Greenland, are dispersive (phase velocity varies with frequency). These waves are likely to be kinematic waves due to the long periods and contemporaneous changes in ice thickness (though there are caveats to this hypothesis and it remains to be tested), so the findings of Riel et al (2020) are also applicable to the interpretation of the results presented in the current manuscript because the authors report observations of kinematic waves.

The work of Greene et al., 2020, should also be referenced and discussed in the revised manuscript. This is a conceptually different approach to all of those mentioned

above and warrants comparison to the method being proposed by the authors. In this work, the authors apply a generalized method that disentangles periodic variations from non-periodic variations.

- Without some quantification of the errors and analysis of the covariances between horizontal and vertical velocity components, I am skeptical of the results because it's not clear that they are valid. Indeed, some results strain my physical intuition (which it is essential to note, does not mean the results are wrong and could be my own failing). For example, 600 meters of downward (vertical) displacement at a point (P7, Figure 7g) in 1.5 years seems rather extreme, even during a surge, especially for a glacier with a total flow speed of only ~1000 m/yr. How does this displacement compare with the local ice thickness and are there any observations of dramatic surface lowering of 100s of meters to validate this observation? As discussed above, I am willing to bet that there is a strong tradeoff between the inferred horizontal and vertical components that results in unrealistically large vertical displacements (i.e., horizontal displacement bleeding into the inferred vertical displacement due to the fact that the flow is close to being in line with the radar line of sight, meaning that there is not enough information in the SAR offsets to allow for accurate inferences of the 3D velocity vector). Some validation of these and other results needs to be done as does some attempt to connect the results to physical models (e.g., given mass conservation, is the horizontal flow speed consistent with the inferred vertical speeds?). Some more specific points:
  - It is hard to glean insight into the data plotted along the transects because the transects are crudely drawn and undoubtedly cross flowlines. This is likely the source of confounding results like those shown in Figure 5 (Klutlan Glacier) where the plots indicate that there are areas (~20 km, 30 km, and 36 km along the transect) that show very slow velocities (<200 m/yr) throughout the observation period but are surrounded upstream and downstream by fast-flowing regions (velocity > 1200 m/yr). A transect drawn along a flowline shouldn't show such behavior, which is the reason it's a good idea to be careful about where one draws profiles. The authors could provide a clearer view of their results by extracting data along profiles that are consistent with the dynamics of the fluid.
  - The possibility that the authors captured a kinematic wave is truly exciting but not very convincing due to the issues discussed above and the qualitative discussion surrounding the observation. Again, it would be useful for the authors to attempt some validation of this observation and some in-depth discussion of the implications. How fast is the wave propagating? Over what distances does it attenuate? Does the wave deform as it travels? Are the amplitude and speed of the wave related to ice thickness, surface slope, or other observables? Is it really a kinematic wave or do longitudinal stress gradients matter? Is it a monochromatic wave or are there multiple frequencies? What is the period (are the periods) of the wave? Even if the authors can't sort out these details, some discussion of basic wave propagation would be valuable, if for no other reason than to point others toward the possibility of using SAR to observe traveling waves of various sorts. Again, I'll point to Riel et al (2020) as an example of

thinking about observing wave propagation with SAR. (And while I'm on this point, the authors need to provide citations to support the idea that one should expect kinematic waves to accompany surges.)

**Minor comments:**
- Title: Given the lack of details and analysis of the method (e.g., lack of synthetic tests showing the validity of the method, lack of analysis of errors and limitations of the method, etc), the title seems overly generalized. What the authors have done here is slightly modify existing methods and applied them to a particular case study. Thus, the title needs to be narrowed and case study mentioned.
- Abstract: Similar comment as the previous one about the title. The authors are seriously overselling the generality and novelty of their work. Other authors have published strikingly similar methods as well as markedly different methods that aim to do the same things the authors are attempting. I think the method the authors are presenting is worth publishing eventually, but it is not as novel as the authors seem to be insisting and careful rewording is warranted. Furthermore, without extensive testing of the method (which, as detailed above, has not been done in this manuscript), the argument that the proposed method is generally applicable to the global catalog of SAR data is unsupported.
- Line 1: It's unusual to refer to the components of a vector as direction and "intensity." It's more comment to reference direction and magnitude, or in the case of velocity, direction and speed. The authors could clean up the description by simply saying "Glacier velocities adjust to a warming climate…"
- Lines 5-6: "We observe seasonal and interannual variations and the maximum horizontal and vertical flow velocity in excess of 1000 and 200 m/yr, respectfully." I don't know what to take from this sentence as there are 4 quantities mentioned (seasonal and interannual, horizontal and vertical velocities) and 5 glaciers in the case study, yet only two speeds mentioned in the sentence. These variations should be given context by comparing them with the mean flow speeds.
- Lines 20-22: A couple of comments: 1) The sentence starts with the phrase "Modern remote sensing techniques …include…" but then lists GNSS. I would not consider GNSS to be remote sensing since the quantity being measured requires a station placed on the glacier. 2) UAV is a curious addition to this list as it is not a method for measuring surface velocity. The first three items in the list are specific techniques, whereas UAVs are merely platforms that collect various types of data, all of which could be described as examples of one of the first three items in the list.
- Line 32: "non-tidewater" needs to be defined.
- Line 35: The authors need to tone down their language and to make a more objective point about the need for studies in Alaska. It is not useful or interesting to say that more studies have been conducted in the ice sheets.
- Line 40: The authors need to elucidate why they think the vertical component of velocity is important, not simply assert it. To a good approximation, ice flows along the surface

slope, so velocity fields given in two horizontal dimensions should capture the vast majority of information about the flow.

- Line 44: Rignot et al., 2011, is a 2D, not a 3D, map.
- Lines 44-46: One cannot validate remotely sensed maps of velocity using MAI as it is simply another remote sensing method.
- Section 2: It seems to me that the order of this section is backward relative to the authors' apparent desire to introduce a general method. If I understand their motivation correctly, the method should be described and evaluated first and separately from the data being used in the case study.
- Line 79: Give the SLC resolution of Sentinel-1 somewhere in this paragraph. Also worth mentioning that SLCs were collected from both Sentinel-1 satellites with a nominal repeat time of 6 days.
- Line 84: "Such a large window size…" Why? Is it because of the flow speed or something else?
- Line 85: What do the authors mean by "distinct peak?" What do the uncertainties look like (curvature of the correlation surface, difference between peak values, etc.)? It would be useful if the authors would give a few examples of the range and azimuth offsets and associated uncertainties for the study areas in the supplement.
- Line 89: The term 'resolution' appears to be improperly used here. Resolution is a statement of information content but what the authors appear to be referring to is grid spacing.
- Eqs. 1 and 2: It's useful to bold variables that indicate vectors and matrices for clarity.
- Line 90: RO and AO should represent *sets* of range and azimuth offsets. (In other words, it's useful to connect RO and AO to the rho and alpha designation for individual displacements used below.)
- Lines 90-91: What are the sizes of matrices L and A in relation to the number of observations?
- Line 91: It's useful to point out that lambda is a scalar (if, in fact, it is).
- Line 95: azimuth and incidence angles need to be defined for the non-SAR-expert.
- Eq 3: There are several differences between the variables used in the equation and those described above (e.g., upper case to lower case 's', different subscripts on rho and alpha, etc.). These need to be consistent. I don't see where the Delta t variables are defined; they need to be defined for completeness.
- Lines 111-112: "Since this method does not make any assumptions about the direction of motion, it provides the optimal solution applicable to any phenomenon." As I mention *ad nauseam* above, the authors provide absolutely no support for this statement. This is merely an assertion as there is no attempt to show that this method produces accurate or optimal solutions for glacier flow or any other phenomena. Unless the authors can provide proof through rigorous testing of the method, this and other statements need to be removed or clearly qualified with statements like "we hypothesize that…"
- Line 113: In my reading, it seems that this is the first mention of coherence in the SAR images. I think a more expanded discussion of noise and errors is required. But at the

very least, the authors should make clear to the non-expert reader what they mean by 'coherent pixel'.

- Line 120: 'visually indistinguishable' from what? Could the authors show us some examples of the effects of regularization in the supplement.
- Line 121: Presumably the authors mean that a line was fit in time at each pixel. It is not clear if the authors are average the speed in time (keeping the unit velocity vector fixed) or each velocity component individually, or some other combination. This should be clarified.
- Line 124: I'm not sure what the authors are referring to here, but it sounds like the kind of statement that would be more useful in a figure caption.
- Line 133: Does 'mean' mean time-averaged or is there some spatial averaging?
- Line 145: Report the window size in meters; the number of pixels can be given as a parenthetical, but what matters is how large the window size is in geographic space.
- Line 164: See above comments about unsupported claims.
- Line 166: What do the authors mean by "with different scales?"
- Line 168: Citation is needed.
- Lines 168-170: Unclear what method the authors are referring to.
- Lines 170-171: The authors state (*emphasis mine*) "[o]ur approach is conceptually similar to the technique of Guo et al. (2020) *that was built on our previous work*." As written, the (italicized) second half of this sentence comes across as petty and unprofessional. All science is built on the work of others. That is its nature and strength. The authors are duly cited by Guo et al. (2020), so phrases like the one italicized above are unnecessary. If the authors decide to keep such statements, it would be best to reword.
- Line 181: Again, why is it the case that large correlation windows and "strong filtering" are needed? The authors need to provide some reasoning that his related to the physical characteristics of the area of interest to support and provide insight into this statement. This is especially true given the claims of general applicability of the methods. If readers were to use this method somewhere else, is it necessarily true that large correlation windows and extensive filtering is needed? How will they know? Additionally, the term "strong filtering" needs to be quantified here.
- Lines 181-182: The fact that surges do affect the results in this study but not those of Gardner et al. means that the comparison is useful for scientific study but not useful for comparing the results between the studies or validating the method.
- Line 184: The fact that glacier flow can deviate significantly from mean flow speeds is well known and documented. This is not something that the authors have demonstrated. So, the statement needs to be reworded or removed.
- Line 185: "SAR-derived time series are often compared with GNSS-derived time series…" Some citations are needed.
- Lines 185-186: "…and both techniques are considered conceptually similar; however, there is an important difference…" This looks like a strawman argument and hearsay. The authors need to provide a citation to show that someone thinks that these methods

are similar. Otherwise, they should just make their point without acting as if they are addressing some controversy that does not exist.

- Lines 185-198: I do not see why this discussion is necessary or useful and think it should be removed entirely. Eulerian and Lagrangian coordinates are well understood (as the authors point out in line 191) and well established. Converting between Eulerian and Lagrangian coordinates is also well understood. This paragraph does not add anything new to the topic.
- Lines 199-210: I do not understand what the authors are trying to say here. So far as I can tell, they are trying to elucidate the distinction between cumulative displacement and instantaneous velocity. I don't see why that is necessary and think that this entire portion of the discussion should be overhauled or removed.
- Lines 204-207: I do not understand the point of this discussion and find it very confusing. It seems that the point being made is that flow direction might change in time in unconfined glaciers but probably not in glaciers whose flow is confined by bedrock. That's a pretty basic point that is not advanced by the current study. The connection the authors are trying make to ice streams is tenuous at best because again, this behavior is well known, methods have existed for quite some time to measure the horizontal flow speed and direction, and ice streams are defined as fast-flowing regions that lack strong lateral confinement from the bed topography.
- Lines 207-208: "Moreover, such direction changes are not easily discerned in 2D or 3D resolved velocity fields." This is a completely and utterly false statement that is both unsupported by this work and contradicted by numerous published studies. Plenty of work has shown that SAR-derived velocity unit vectors are accurate to within a couple of degrees in the horizontal (see stacks of papers by Rignot, Joughin, Rott, and other pioneers of this field). I argue that such accuracy is more than sufficient to quantify changes in flow direction.
- Lines 219-220: I don't fully understand why this interpretation is correct. In an idealized sense, SAR observations should be sensitive to the motion of the scatterers, not necessarily the surface. Indeed, it is critical when talking about SAR-derived vertical velocities to distinguish vertical velocity from changes in surface elevation. When one is discussing vertical velocities in glaciers that experience surface melt, as all glaciers in this study do, there is an additional complexity due to the dielectric influence of surface melt on the radar signal (penetration depth, or phase center). This is discussed to some extent by Rignot, Echelmeyer, and Krabill (2001) and Minchew et al. (2015). As mentioned above, some careful exploration of this component of the signal is needed here along with some qualification of these findings.
- Line 220: My previous comment notwithstanding, the dynamic state of Malaspina has been reported previously (e.g., Larsen et al., 2015; Muskett et al., 2003) and appropriate citations are needed.
- Line 224: Such large changes in vertical velocity over such short distances should be manifest in the surface topography and thus should be tested against surface elevation time-series where available. ArcticDEM would be a good place to look and would

require very little effort on the part of the authors. This could go a long way to showing that the inferred vertical velocities are accurate.

- Line 235: The authors need to qualify such statements: "We hypothesize that the downward vertical motion…represents a kinematic wave…" More needs to be done to verify that this is what is recorded in the data (see above discussion).
- Line 242: I'm not sure I agree with the statement that vertical and horizontal motion can only be derived from SAR. I do agree that nadir optical measurements (like Landsat) are insensitive to vertical, but I see no reason why time-dependent altimetry could not be fused into the data to give vertical and horizontal motion (again, the distinction would need to be made between vertical velocity of the ice and vertical motion of the surface). I think it's best for the authors to soften this statement, perhaps saying that SAR is *currently* the most viable way to infer 3D velocity.
- Line 245: As noted several times above, this qualification needs to be infused throughout the discussion section to provide a more nuanced discussion.
- Where can readers download the software?
- Where can the processed data from this study be accessed?
- Table 1: What is the range of incidence angles in the image? The last column should be labeled as the number of SLC scenes.
- Figure 1: Add citation for ASTER DEM.
- Figure 2 caption: 'octagon' is referenced but no octagons are in the figure. Delta t_i should be defined as Delta t_i = t_{i+1} – t_i
- Figure 3: 'Mean temporal resolution' should be clearly defined and distinguished from the repeat time of a given track.
- Figure 4:
  - Fix the letter labels on the panels. Panel a is missing its letter. There are two letter b's, one of which looks like it is the label for panel c.
  - The abbreviations for the different glaciers should be defined in the caption.
  - Why put the date in YYYYMMDD format just to have to explain the format in parentheses? Why not just give the name of the month and the day and year?
  - What is the local time of the acquisition (this matters for surface melt)?
  - Panel a (top left): Add distance markers along the profile for easy reference to the surrounding figures. Make the scale bar legible by changing colors or adding a box behind it.
  - Panel b: Are these the time-averaged velocities or for a specific time? What data set is used for the contour lines?
  - Panel c: same comment as for panel b about specifying that the velocities are time averaged.
  - Panel d: What are the white gaps?
- Figures 5 and 6: Same comments as for Fig 4.
- Figure 7: It would be much easier to interpret these results if the time-series were merged with Figures 4-6 and/or if the points were labeled with the letter of their respective glacier. The 'P' designation does not provide the reader with any information.

---

## Referee Comment (RC3) · Anonymous Referee #3 · 6 Jan 2021

This draft proposes a novel method for 3-D velocity mapping of glaciers using modern spaceborne SAR measurements. Instead of using surface parallel flow constraint, this method combines speckle offset tracking and MSBAS , which is also assisted with regularization. It is further validated with Sentinel-1 data over 5 glaciers in Alaska. The draft is generally well written and the methodology is reasonable. However, there are couple of issues that need to be resolved/expanded in detail.

Major comments:

1. Study area description is better to be extracted from Section I, together with the dataset description in Section 2, to form a separate section, named "Area and Data"
2. The model description in Section 2 needs to be clearly rewritten and expanded in detail. If sufficient details do not fit the section, they could be added to an appendix then.

Detailed comments:

1. Line #13: this is the same sentence as included in the abstract, thus redundant
2. Line #26: SAR-based correlation algorithms not only operate on radar backscatter, but also radar backscatter and phase (complex-valued correlation).
3. Section I: you introduced multiple methods for velocity mapping (SPO, DInSAR, MAI), but did not mention what specific one you use in this work and why you chose that one. It is clear later in Section 2 that you used SPO, but would be better to motivate it in Section 1
4. Line #74: the last sentence is also the same as that included in the abstract, i.e. redundant
5. Line #83-84: the number of pixels also need to be converted to distance in m. I see you want a square sampling interval on the ground by choosing 64 x 16 for Sentinel-1 images.
6. Line #84-86: why isn't the correlation window (256 x 256) a square window on the ground to be consistent with the sampling interval. Also, the numbers you chose are equivalently 1km x 4km on the ground. With the 2km wide median filter, you essentially got a spatial resolution around 2km or at least on the order of km. Even though you resampled the products into 200m, this does not justify the spatial resolution is 200m. That said, the spatial resolution is too coarse over fast-moving glaciers, and the resulting spatial pixels are strongly correlated.
7. Eq. 1: you should either cite a reference or explicitly show the proof of this equation. The way it current shows is introducing the equation out of the blue. When details of the proof is involved, you can also put that in an appendix if necessary.
8. Eq. 1: the matrix/vector notation should be clearly defined by providing the dimension, which should then be related to the number of ascending/descending acquisitions.
9. Eq. 3: this simplified example is not clear. First of all, it is not clear how the Sa and Sr components are coupled in that way. To do so, you probably need a separate graphic illustration besides Fig. 2 or an appendix. If you can find a citation that does exactly the same thing, that would work too. Second, the notation of the rho and alpha elements in

the column to the right of the "=" sign were never introduced since they are different from those described in Line #96-101. Third, the last three elements in the velocity vector only show the northing of velocity at t3 and easting/vertical of velocity at t4. Why is that and what happened to the missing other components at t3 and t4, and what happened to t5?

10. Line #112: "any phenomenon" This is to vague. You need to be specific what type of phenomenon

11. Line #114: the dimension is 609 x 1014 for the matrix to be inverted. As mentiond above, how to relate these numbers to your total ascending/descending acquisitions. After Eq. 1, you should add a symbolic equation that relates the matrix dimension to the number of radar acquisitions

12. Line #116: please report the specific computer setting and runtime for your case

13. Line #117-120: add a sentence explaining why regularization is needed, and what happens if not included. Any comparison of the horizontal velocity results derived from the 3-D approach with regularization to those from the 2-D methods? Please add some simple analysis

14. Line #121: what do you mean by "mean linear flow velocity" especially the word "linear"? Regarding "mean", is the 3-year mean value meaningful for those fast-moving glacier terminus? It is expected that such glaciers should have strong seasonal/interannual changes. Probably 1-year mean value is better

15. Line #123: how much coarser resolution is the horizontal one resampled to? And also why is <5m/yr removed? Velocity estimates over slow-moving areas (e.g. < 15m/yr) are usually used to tie the products and calibrate the estimation bias. How did you calibrate your Sentinel-1-derived velocity products?

16. Line #180: "every single range and azimuth offset maps must be coherent at every pixel" what does it exactly mean?

17. Line #181: "large correlation window followed by strong filtering" gives you much lower resolution and spatially correlated pixels. Isn't that problematic for fast-moving glacier terminus? Please comment and justify.

---

## Author Comment (AC1) · 7 Feb 2021

*Dear Reviewer,*

*Thank you very much for providing your valuable comments that helped us to significantly improve our manuscript. Below we provide our detailed responses to your questions in italic font.*

*Best regards,*
*Sergey Samsonov, Kristy Tiampo and Ryan Cassotto*

Review of Measuring the state and temporal evolution of glaciers using SAR-derived 3D time series of glacier surface flow

By Sergey Samsonov, Kristy Tiampo, and Ryan Cassotto

This paper present a technique for producing times series of the 3D glacier motion using ascending/descending data.

Overall it's a good paper, and the technique seems sound. That said, though I would like to see a better technical discussion with respect to errors and temporal/spatial resolution as noted below. The discussion needs some work make it clear what the data is actually showing, especially with respect to what is the source of the vertical displacement (see below).

*Reply: Thank you. We are prepared to address these issues.*

General points I don't see much discussion of errors. Typically, with 6-day sampling and azimuth offsets, you are going to get about errors of about 20-m/yr. The vertical velocities are more driven by range offsets, but the time everything is solved for, some of those errors are going to fold into the vertical solution.

*Reply: We utilize a standard speckle tracking technique, in which sources of errors are well understood. We will strengthen the discussion of errors and explain how these errors propagate to the 3D decomposed results. Note that while individual offset maps have large errors, the derived mean velocities have significantly smaller errors. Which is similar to reconstructing GNSS velocities from long time series of GNNS observations.*

Vertical motion is a combination of the vertical motion due to surface parallel flow other factors (e.g., submergence/emergence velocity or subsidence/inflation due to subglacial water flow). In general, the surface parallel vertical displacement will be the dominant term and it will vary with the horizontal speed, especially for mountain glaciers with relatively steep slopes. The other forms of vertical displacement are the more interesting terms though. So, it would make sense to compute the surface parallel component and remove it to isolate the other types of vertical motion. Other-wise statements like "The predominately downward flow of ice observed throughout the Malaspina Glacier's massive lobe (Figure 4b,c) indicates that ablation rates have exceeded emergence velocities during our 4 year study period, implying" are potentially incorrect or at least presented without proper context. I have no reason not to believe the glacier is wasting down faster than the emergence velocity, but in general glaciers flow downhill, so that could largely due to the surface parallel flow component. More-over, from image to image, the glacier is largely measuring the same coherent patch of speckle, which would record motion of the surface due to ice dynamics. It should not be measuring downward motion of the surface due to direct ablation (unless you phase of the ablated layer is accounted for, which does not seem to be happening here). In other words, this is like measuring the downward motion of a GPS on a pole sunk in the ice, which will not measure ablation, vs a GPS placed on the surface, which will measure ablation. Distinctions such as these need to made in the discussion.

*Reply: We agree that surface parallel displacement is expected to dominate along mountain glaciers with steep slopes, much like the flow patterns observed along Seward Glacier (Fig 4b). However, although bed elevations along the piedmont lobe are unknown, they are not expected to be steeply dipping, so surface parallel flow here is unlikely to be dominant. Nonetheless, our original statement requires an additional qualifier. We will amend it to read: "Low magnitude horizontal velocities combined with a predominately downward flow throughout Malaspina Glacier's massive lobe (Figure 4b,c) indicates...." Also, we have differentiated between*

*surface parallel flow (SPF) and non-SPF in another paper that was recently accepted. We will provide a reference to the manuscript that describes this work.*

There is some discussion about penetration effects, but it is import to note you can get some really strange offset patterns when you have soaked firn, which can be spatially coherent over large distances (can map into errors of several hundred m/yr). I am not sure that some of what's being seen is this kind of effect (e.g. blue patches Fig. 4b).

*Reply: That is quite possible. It is, however, not specific to our processing technique. Our final maps of vertical flow are based on averages of four full years of observations; thus, have an equal number of melt and accumulation seasons that should account for differences in penetration depth. Furthermore, the overall trend in elevation change over these four years (10's meters, Fig 7) exceeds the maximum 10 m depth penetration measured in dry firn in this area in earlier studies (e.g. Rignot et al., 2001); thus, errors due to seasonal variations in depth penetration should be small relative to the true signal. We will discuss this in the revised manuscript as something to watch out for during the interpretation of results.*

Point, P2, seems to exhibit a seasonal cycle not seen in the other data. It also happens to be near a marginal lake. I wonder if pressure variations as the lake fills and drains are contributing to the seasonal up and down motion. Some discussion as to why this point has a strong seasonal signal would be good.

*Reply: Actually, most of the glaciers at lower elevations show similar behavior (hence this point is chosen as a characteristic). This motion is caused by glacier melting during summer. Note that the magnitude of vertical motion caused by the melting process is small and nearly constant. It is hard to see in plots that have a different vertical scale (e.g. P1).*

Specific points Line 48: "However, the SPF constraint is only applicable to glaciers in steady state." This statement is not correct, the SPF assumption ignore the submer-gence and emergence velocity and other vertical motion, which is true whether or not the glacier is in steady state. And if the glacier is not in steady state, it will still measure vertical velocity variations that are parallel to the surface.

*Reply: That is correct, we now understand these processes better. We will address this comment in the revised manuscript.*

Line 84: A 256x256 sampling window (both patches???) will provides about 3.5km resolution (256 * 13.9 m azimuth resolution), which is further degraded by a 2-km median filter. In addition to the lack of resolution, this can cause problems where the matcher will lock on stationary rock areas more easily than the glacier to report zero velocities a km or 2 inboard of the margin. How is this dealt with. There should be some discussion of what the spatial resolution is. Certainly, the ground resolution is not 200-m as stated, even if the data posted are at 200 m.

*Reply: We recognize the benefits of having high-resolution results. Unfortunately, in this area, the application of a small window produces measurements that are too noisy, and if we only select pixels with high SNR the spatial coverage reduces to nothing. Likely this happens because of the warm regional climate and wet glacier surface (in comparison to Greenland or the Arctic or the Antarctic). Therefore, we are limited to using a larger window of256x256 pixels.*

*We consulted the developers of GAMMA processing software that is used to compute speckle offsets. We were advised that the window that is used to compute the offsets is not uniform, pixels in the center have larger weights than those pixels on edges. The effective resolution is about four times higher than the window size. The process of the extraction of offsets, as it is implemented in the software, is not linear. We acknowledge that the spatial resolution is reduced by using such a large window. However, this is necessary for extracting temporal information. Note that the computation of offsets, in general, is not specific in any way to the technique presented here.*

*To confirm this we computed offsets for a single pair using 64x64, 128x128, and 256x256 windows. In figure 1, below, we present these results before and after filtering. As you can see, while there are differences, overall the signal is consistent. We will add this figure to the revised manuscript. Note that filtering does not reduce the resolution significantly. Again, we found that these processing parameters are optimal for our purposes in this region; it does not mean that they would be optimal in other areas.*

[Figure]

*Figure 1: (top-left) Seward range, (top-right) Seward azimuth, (bottom-left) Klutlan range, (bottom-right) Klutlan azimuth.*

Line 85: What corrections were applied for baseline and to calibrate the data (e.g., were control points used to remove biases).

*Reply: We use precise orbits downloaded from the ESA website. We calibrate the offsets by fitting and removing the polynomial model. This approach works well in this region where most areas do not show any motion. The entire Sentinel-1 scene is processed as a whole, and it is cut into small sub-regions only for visualization in the manuscript. Note that the entire Sentinel-1 scene extends far beyond the area shown in the manuscript. The software provides alternative methods of calibration that can be employed in other, more complex, regions (e.g. calibration against multiple reference regions, Z-score).*

Equation 1 – please break separate into two equations (put the cumulative as a different equation).

*Reply: Will do.*

117 "temporal smoothing" What is the temporal resolution after regularization.

*Reply: The effective temporal resolution decreases by about a factor of two. This depends on the strength of the regularization that is controlled by a regularization parameter lambda. For some studies, the decrease of a temporal resolution would be unacceptable (e.g. timing of the particular events). In that case, a small lambda can be chosen. In this study, we believe, it is not particularly important, because trends are overall smooth. The selection of the optimal lambda can be performed using the L-curve method. It has been previously explicitly shown in our previous manuscripts (https://doi.org/10.1016/j.rse.2013.12.017, https://doi.org/10.1080/07038992.2017.1344926).*

Line 224-226. I would like to see the surface parallel flow components removed before seeing discussion about kinematic waves.

*Reply: This has already been done. We will provide a reference to the manuscript that describes this work.*

Figures. The x-axis of the vector profile plots could be lined up with the color times se-ries plots. The vector plots while pretty, don't really give a good idea of the magnitudes of the vertical velocity. Please show the profiles also on the b panels since that actually shows magnitudes of vertical motion. Also use some kind of symbol on the c and d plots to indicate where the points PX are.

*Reply: Will do.*

Would be helpful to see the color panels broken out separately as horizontal and vertical magnitudes. The alternating patches of slow and fast flow are strange.

*Reply: Will do. The alternative patches can be due to profiles not well aligned with flow lines. In the revised manuscript we will utilize an OGGM software (https://docs.oggm.org) for selecting profiles along flow lines.*

---

## Author Comment (AC2) · 7 Feb 2021

*Dear Reviewer,*

*Thank you very much for providing your valuable comments that helped us to significantly improve our manuscript. Below we provide our detailed responses to your questions in italic font.*

*Best regards,*
*Sergey Samsonov, Kristy Tiampo and Ryan Cassotto*

This draft proposes a novel method for 3-D velocity mapping of glaciers using modern spaceborne SAR measurements. Instead of using surface parallel flow constraint, this method combines speckle offset tracking and MSBAS, which is also assisted with regularization. It is further validated with Sentinel-1 data over 5 glaciers in Alaska. The draft is generally well written and the methodology is reasonable. However, there are couple of issues that need to be resolved/expanded in detail.

Major comments:

1. Study area description is better to be extracted from Section I, together with the dataset description in Section 2, to form a separate section, named "Area and Data"

*Reply: Thank you. We are prepared to address this issue in the revised manuscript.*

1. The model description in Section 2 needs to be clearly rewritten and expanded in detail. If sufficient details do not fit the section, they could be added to an appendix then.

*Reply: We are prepared to address this issue in the revised manuscript. In the past, the first author had reviewers telling him not to repeat information that can be cited.*

Detailed comments:

Line #13: this is the same sentence as included in the abstract, thus redundant

*Reply: Our understanding is that an abstract should stand on its own. We will modify the redundant sentences.*

Line #26: SAR-based correlation algorithms not only operate on radar backscatter, but also radar backscatter and phase (complex-valued correlation).

*Reply: That is correct. We will correct this in the revised manuscript.*

Section I: you introduced multiple methods for velocity mapping (SPO, DInSAR, MAI), but did not mention what specific one you use in this work and why you chose that one. It is clear later in Section 2 that you used SPO, but would be better to motivate it in Section 1

*Reply: We will add a statement in the introduction to clarify this.*

Line #74: the last sentence is also the same as that included in the abstract, i.e. redundant
*Reply: We will amend to remove redundant statements.*

Line #83-84: the number of pixels also need to be converted to distance in m. I see you want a square sampling interval on the ground by choosing 64 x 16 for Sentinel-1 images.

*Reply: Will do.*

Line #84-86: why isn't the correlation window (256 x 256) a square window on the ground to be consistent with the sampling interval. Also, the numbers you chose are equivalently 1km x 4km on the ground. With the 2km wide median filter, you essentially got a spatial resolution around 2km or at least on the order of km. Even though you resampled the products into 200m, this does not justify the spatial resolution is 200m. That said, the spatial resolution is too coarse over fast-moving glaciers, and the resulting spatial pixels are strongly correlated.

*Reply: We recognize the benefits of having high-resolution results. Unfortunately, in this area, the application of a small window produces measurements that are too noisy, and if we only select pixels with high SNR the spatial coverage reduces to nothing. Likely this happens because of the warm regional climate and wet glacier surface (in comparison to Greenland or the Arctic or the Antarctic). Therefore, we are limited to using a large window of256x256 pixels.*

*We consulted the developers of GAMMA processing software that is used to compute speckle offsets. We were advised that the window that is used to compute the offsets is not uniform, pixels in the center have large weights than pixels on edges. The effective resolution is about four times higher than the window size. The process of extraction of offsets, as it is implemented in the software, is not linear. We acknowledge that the spatial resolution is reduced by using such a large window. However, this is necessary for extracting temporal information. Note that the computation of offsets, in general, is not specific in any way to the presented here technique.*

*To confirm this we computed offsets for a single pair using 64x64, 128x128, and 256x256 windows. In figure 1 we present these results before and after filtering. As you can see, while there are differences, overall the signal is consistent. We will add this figure to the revised manuscript. Note that filtering does not reduce the resolution significantly. Again, we found that these processing parameters are optimal for our purposes, it does not mean that they would be optimal in other areas.*

[Figure]

Figure 1: (top-left) Seward range, (top-right) Seward azimuth, (bottom-left) Klutlan range, (bottom-right) Klutlan azimuth.

Eq. 1: you should either cite a reference or explicitly show the proof of this equation. The way it current shows is introducing the equation out of the blue. When details of the proof is involved, you can also put that in an appendix if necessary.

Reply: It is a basic equation with a meaning similar to V*t =D. We will clarify it in the revision.

Eq. 1: the matrix/vector notation should be clearly defined by providing the dimension, which should then be related to the number of ascending/descending acquisitions.

Reply: Will do.

Eq. 3: this simplified example is not clear. First of all, it is not clear how the Sa and Sr components are coupled in that way. To do so, you probably need a separate graphic illustration besides Fig. 2 or an appendix. If you can find a citation that does exactly the same thing, that would work too. Second, the notation of the rho and alpha elements in the column to the right of the "=" sign were never introduced since they are different from those described in Line #96 -101. Third, the last three elements in the velocity vector only show the northing of velocity at t3 and easting/vertical of velocity at t4. Why is that and what happened to the missing other components at t3 and t4, and what happened to t5?

Reply: We will clarify the notation and make it consistent.

Line #112: "any phenomenon" This is to vague. You need to be specific what type of phenomenon

*Reply: We meant to say any surface motion.*

Line #114: the dimension is 609 x 1014 for the matrix to be inverted. As mentiond above, how to relate these numbers to your total ascending/descending acquisitions. After Eq. 1, you should add a symbolic equation that relates the matrix dimension to the number of radar acquisitions

*Reply: Sounds good, we will clarify that.*

Line #116: please report the specific computer setting and runtime for your case

*Reply: We will do. For us, it takes about 24 hours of processing time on a single node with 44 cores. An MPI version of msbas software is on its way. The processing time in an MPI version will be reduced proportionally to the number of nodes.*

Line #117-120: add a sentence explaining why regularization is needed, and what happens if not included. Any comparison of the horizontal velocity results derived from the 3-D approach with regularization to those from the 2-D methods? Please add some simple analysis

*Reply: Will do.*

Line #121: what do you mean by "mean linear flow velocity" especially the word "linear"? Regarding "mean", is the 3-year mean value meaningful for those fast-moving glacier terminus? It is expected that such glaciers should have strong seasonal/interannual changes. Probably 1-year mean value is better

*Reply: With the technique presented here, we compute velocities between consecutive SAR acquisitions. Sentinel-1 data is acquired with either a six or 12 day revisit cycle, and velocities are computed for every revisit cycle interval (so-called instantaneous velocities). The flow displacement time series are then reconstructed from these instantaneous velocities. Assuming a 12 day Sentinel-1 revisit cycle, our technique produces about 365/12 = ~30 4D velocities per year. Since all these data cannot be presented in a single publication (30 velocities per year x 4D x 4 years ~ 480 figures), as a simplified representation of our results that require only four figures, we choose to compute mean velocities by fitting a line to the flow displacement time series. Along with the mean velocities for each of the four components, we compute their standard deviations and coefficients of determination ($R^2$), which help us understand if the linear model provides a good approximation. For some regions, a linear approximation cannot capture all the complexity of the motion. For these regions, we plot flow displacement time series, which describe instantaneous velocity at each moment in time. Annual or any other duration (monthly, quarterly) velocities can also be computed from our flow displacement time series by aggregating time series at different intervals.*

*Concerning selecting the length of time to estimate mean flow, a shorter period could certainly be used; however, our aim for this manuscript was to demonstrate the technique used and the overall trends that occurred over 4 years. The flow displacement time series and text in the discussion address the benefits of short term analyses such as seasonal and interannual variability.*

Line #123: how much coarser resolution is the horizontal one resampled to? And also why is <5m/yr removed? Velocity estimates over slow-moving areas (e.g. < 15m/yr) are usually used to tie the products and calibrate the estimation bias. How did you calibrate your Sentinel-1-derived velocity products?

*Reply: The resolution and masking out is performed only for improving visualization, otherwise, the image gets oversaturated with details. We use precise orbits downloaded from the ESA website. We calibrate the offsets by fitting and removing the polynomial model. This approach works well in this region where most areas do not show any motion. The entire Sentinel-1 scene is processed as a whole, and it is cut into small sub-regions only for visualization in the manuscript. Note that the entire Sentinel-1 scene extends far beyond the area shown in the manuscript. The software provides alternative methods of calibration that can be employed in other, more complex, regions (e.g. calibration against multiple reference regions, Z-score).*

Line #180: "every single range and azimuth offset maps must be coherent at every pixel" what does it exactly mean?

*Reply: This means that if a pixel is incoherent on one of the offset maps (e.g. 20190201-20190213) it will be excluded from the processing and all results will have NaN value at that pixel. In general, our processing software can handle partially incoherent pixels (it will be filled by the regularization); however, in this study, we choose to utilize only pixels coherent in all offset maps so their precision is identical. The technique that utilizes partially coherent pixels will be discussed in the follow-up publications.*

Line #181: "large correlation window followed by strong filtering" gives you much lower resolution and spatially correlated pixels. Isn't that problematic for fast-moving glacier terminus? Please comment and justify.

*Reply: That is correct. However, it is a necessity to use a large window and filtering as processing with a low correlation window produces very noisy results in this region. This has already been discussed above. It will be addressed in the revised manuscript. Also, in the revised version we will be using a filter with the Gaussian window, we found that it performs better for small and large glaciers. Finally, none of the glaciers in our study area is tidewater and thus do not experience the rapid flow that typifies tidewater glacier termini.*

---

## Author Response (AR1)

*Dear Reviewer 1,*

*Thank you very much for providing your valuable comments that helped us to significantly improve our manuscript. Below we provide our detailed responses to your questions in italic font. Here is a quick summary of changes:*

- *Addressed reviewers' comments to the best of our knowledge;*
- *Added recent Sentinel-1 data, mainly to investigate what is happening at region P1 at the Malaspina Glacier;*
- *Recomputed offset maps using smaller 128x128 window and the Gaussian filter with 1.3 km 6-sigma width;*
- *Detected another surging Kluane Glacier and analyzed it in-detail;*
- *Used OGGM software to extract flow lines and performed all analysis for selected flow lines;*
- *Simplified interpretation by removing reference to kinematic waves, which require more attention and, which possibly will be addressed in a separate publication.*
- *Provided animations for four AOIs.*

*Best regards,*
*Sergey Samsonov, Kristy Tiampo, and Ryan Cassotto*

Review of Measuring the state and temporal evolution of glaciers using SAR-derived 3D time series of glacier surface flow

By Sergey Samsonov, Kristy Tiampo, and Ryan Cassotto

This paper present a technique for producing times series of the 3D glacier motion using ascending/descending data.

Overall it's a good paper, and the technique seems sound. That said, though I would like to see a better technical discussion with respect to errors and temporal/spatial resolution as noted below. The discussion needs some work make it clear what the data is actually showing, especially with respect to what is the source of the vertical displacement (see below).

*Reply: Thank you. We addressed these issues.*

General points I don't see much discussion of errors. Typically, with 6-day sampling and azimuth offsets, you are going to get about errors of about 20-m/yr. The vertical velocities are more driven by range offsets, but the time everything is solved for, some of those errors are going to fold into the vertical solution.

*Reply: We utilize a standard speckle tracking technique, which sources of errors are well understood from previous studies. Note that while individual offset maps have large errors (30 m/year in range and 120 m/year in azimuth, see discussion), the derived mean velocities have significantly smaller errors (0.7, 0.3, and 0.2 m/year, while the maximum values are 21, 18 and 7 m/year, for northern, eastward and vertical components, respectively). This is similar to reconstructing GNSS velocities from long time series of GNNS observations.*

Vertical motion is a combination of the vertical motion due to surface parallel flow other factors (e.g., submergence/emergence velocity or subsidence/inflation due to subglacial water flow). In general, the surface parallel vertical displacement will be the dominant term and it will vary with the horizontal speed, especially for mountain glaciers with relatively steep slopes. The other forms of

vertical displacement are the more interesting terms though. So, it would make sense to compute the surface parallel component and remove it to isolate the other types of vertical motion. Other-wise statements like "The predominately downward flow of ice observed throughout the Malaspina Glacier's massive lobe (Figure 4b,c) indicates that ablation rates have exceeded emergence velocities during our 4 year study period, implying" are potentially incorrect or at least presented without proper context. I have no reason not to believe the glacier is wasting down faster than the emergence velocity, but in general glaciers flow downhill, so that could largely due to the surface parallel flow component. More-over, from image to image, the glacier is largely measuring the same coherent patch of speckle, which would record motion of the surface due to ice dynamics. It should not be measuring downward motion of the surface due to direct ablation (unless you phase of the ablated layer is accounted for, which does not seem to be happening here). In other words, this is like measuring the downward motion of a GPS on a pole sunk in the ice, which will not measure ablation, vs a GPS placed on the surface, which will measure ablation. Distinctions such as these need to made in the discussion.

*Reply: We have differentiated between surface parallel flow (SPF) and non-SPF in another paper that was recently published (https://doi.org/10.1016/j.rse.2021.112343). Here we can refer to panel "c" in Figures 6-9, where it can be seen that the slope of flow vectors differs from the slope of the topography. If motion was parallel to the surface then flow vectors would also be parallel to the slope. Also, please see four animations that show temporal variability of the flow lines relative to the topographic slope. We addressed the difference between Eulerian (SAR) and Lagrangian (GNSS) representations in the Discussion section.*

There is some discussion about penetration effects, but it is import to note you can get some really strange offset patterns when you have soaked firn, which can be spatially coherent over large distances (can map into errors of several hundred m/yr). I am not sure that some of what's being seen is this kind of effect (e.g. blue patches Fig. 4b).

*Reply: That is quite possible. It is, however, not specific to our processing technique. Our final maps of vertical flow are based on averages of four full years of observations; thus, have an equal number of melt and accumulation seasons that should account for differences in penetration depth.*

*This is how we addressed this issue in the previous manuscript (https://doi.org/10.1016/j.rse.2021.112343):*
*"The penetration depth likely changes throughout the year due to seasonal temperature changes but this change is small over 6 to 12 days, over which the individual offset maps are computed. The error due to the seasonal variability in penetration depth is removed by differencing primary and secondary observations. This can be deduced from observing seasonally correlated signals only in a few regions (at low and high elevations). Higher temperatures during summer would result in higher water content and less penetration depth, which would appear as an upward movement in the flow displacement time series. We, however, observe downward motion during summer. The radar penetration depth could also change throughout the day due to surface melt and meltwater percolation variations. Ascending and descending data is acquired at different times; it is processed separately and combined only during MSBAS analysis. Any error due to diurnal variations in penetration depth is also removed by differencing primary and secondary observations since both images are acquired at precisely the same time of day."*

Point, P2, seems to exhibit a seasonal cycle not seen in the other data. It also happens to be near a marginal lake. I wonder if pressure variations as the lake fills and drains are contributing to the seasonal up and down motion. Some discussion as to why this point has a strong seasonal signal would be good.

*Reply: Note that point numbering has changed, P2 is now closer to current P4. Actually, many of the glaciers in our study area show similar behaviour (hence this point is chosen as a characteristic). This is evident in the time-series shown in the supplemental movies. The motion is believed to be caused by melt-induced seasonal variations and it was modelled in (https://doi.org/10.1016/j.rse.2021.112343). Note that the magnitude of vertical motion caused by the melting process is small and nearly constant. It is hard to see in plots that have a different vertical scale (e.g. P1).*

Specific points Line 48: "However, the SPF constraint is only applicable to glaciers in steady state." This statement is not correct, the SPF assumption ignore the submer-gence and emergence velocity and other vertical motion, which is true whether or not the glacier is in steady state. And if the glacier is not in steady state, it will still measure vertical velocity variations that are parallel to the surface.

*Reply: That is correct, we now understand these processes better. We revised the manuscript accordingly.*

Line 84: A 256x256 sampling window (both patches???) will provides about 3.5km resolution (256 * 13.9 m azimuth resolution), which is further degraded by a 2-km median filter. In addition to the lack of resolution, this can cause problems where the matcher will lock on stationary rock areas more easily than the glacier to report zero velocities a km or 2 inboard of the margin. How is this dealt with. There should be some discussion of what the spatial resolution is. Certainly, the ground resolution is not 200-m as stated, even if the data posted are at 200 m.

*Reply: In the revised version we reduced the correlation window to 128x128 pixels and used a Gaussian filter with a width of 1.3 km (6-sigma). We recognize the benefits of having high-resolution results. Unfortunately, in this area, the application of a small window produces measurements that are too noisy, and if we only select pixels with high SNR the spatial coverage reduces to nothing. Therefore, we are limited to using a larger window.*

*We consulted the developers of the GAMMA processing software that is used to compute speckle offsets. We were advised that the window that is used to compute the offsets is not uniform, pixels in the centre have larger weights than those pixels on edges. The effective resolution is about four times higher than the window size; thus, the effective resolution is X by Y. The process of the extraction of offsets, as it is implemented in the software, is not linear. We acknowledge that the spatial resolution is reduced by using such a large window. However, this is necessary for extracting temporal information. Note that the computation of offsets, in general, is not specific in any way to the technique presented here.*

*To confirm this we computed offsets for a single pair using 64x64, 128x128, and 256x256 correlation windows. In figure 1, below, we present these results before and after filtering. As you can see, while there are differences, overall the signal is consistent. Note that filtering does not reduce the resolution significantly. Again, we found that these processing parameters are optimal for our purposes in this region; however, it does not mean that they would be optimal in other areas.*

[Figure]

*Figure 1: (top-left) Seward range, (top-right) Seward azimuth, (bottom-left) Klutlan range, (bottom-right) Klutlan azimuth.*

Line 85: What corrections were applied for baseline and to calibrate the data (e.g., were control points used to remove biases).

*Reply: We use precise orbits downloaded from the ESA website. We calibrate the offsets by fitting and removing the polynomial model. This approach works well in this region where most areas do not show any motion. The entire Sentinel-1 scene is processed as a whole, and it is cut into small sub-regions only for visualization in the manuscript. Note that the entire Sentinel-1 scene extends far beyond the area shown in the manuscript. The software provides alternative methods of calibration that can be employed in other, more complex, regions (e.g. calibration against multiple reference regions, Z-score).*

Equation 1 – please break separate into two equations (put the cumulative as a different equation).

*Reply: Done. Now we have two equation 1a and 1b.*

117 "temporal smoothing" What is the temporal resolution after regularization.

*Reply: The effective temporal resolution decreases by about a factor of two. This depends on the strength of the regularization that is controlled by a regularization parameter lambda. For some studies, the decrease of a temporal resolution would be unacceptable (e.g. timing of the particular events). In that case, a small lambda can be chosen. In this study, we believe, it is not particularly important, because trends are overall smooth. The selection of the optimal lambda can be performed using the L-curve method, which has been explicitly shown in our previous manuscripts (https://doi.org/10.1016/j.rse.2013.12.017, https://doi.org/10.1080/07038992.2017.1344926).*

Line 224-226. I would like to see the surface parallel flow components removed before seeing discussion about kinematic waves.

*Reply: This issue was raised by multiple reviewers but it was not given a proper detail in our manuscript. Since it is beyond of the original scope of the manuscript we decided to remove the discussion on kinematic waves entirely.*

Figures. The x-axis of the vector profile plots could be lined up with the color times series plots. The vector plots while pretty, don't really give a good idea of the magnitudes of the vertical velocity. Please show the profiles also on the b panels since that actually shows magnitudes of vertical motion. Also use some kind of symbol on the c and d plots to indicate where the points PX are.

*Reply: To address this issue we introduced distance markers in green in panels "a" in Figures 6-9. These distance markers correspond to distance along profile in panels "c"-"e" in Figures 6-9.*

Would be helpful to see the color panels broken out separately as horizontal and vertical magnitudes. The alternating patches of slow and fast flow are strange.

*Reply: We have done it – panels "d" and "e" in Figures 6-9.  The alternating patches along the y-axis show seasonal variations in flow while the same alternating pattern along the x-axis shows spatial variability along flow.  Note that in the revised manuscript we utilize an OGGM software (https://docs.oggm.org) for selecting profiles along flow lines to improve sampling along flow, which has improved the results.*

*Dear Reviewer 2,*

*Thank you very much for providing your valuable comments that helped us to significantly improve our manuscript. Below we provide our detailed responses to your questions in italic font. Here is a quick summary of changes:*

- *Addressed reviewers' comments to the best of our knowledge;*
- *Added recent Sentinel-1 data, mainly to investigate what is happening at region P1 at the Malaspina Glacier;*
- *Recomputed offset maps using smaller 128x128 window and the Gaussian filter with 1.3 km 6-sigma width;*
- *Detected another surging Kluane Glacier and analyzed it in-detail;*
- *Used OGGM software to extract flow lines and performed all analysis for selected flow lines;*
- *Simplified interpretation by removing reference to kinematic waves, which require more attention and, which possibly will be addressed in a separate publication.*
- *Provided animations for four AOIs.*

*Best regards,*
*Sergey Samsonov, Kristy Tiampo, and Ryan Cassotto*

Review of "Measuring the state and temporal evolution of glaciers using SAR-derived 3D time series of glacier surface flow" by Samsonov, Tiampo, and Cassotto

**Summary**

The authors discuss a method for inferring time-dependent 3D surface velocity fields from synthetic aperture radar (SAR) data and apply this method to data collected from Sentinel I in 2016-2020 over five outlet glaciers in Alaska: Agassiz, Seward, Malaspina, Klutlan, and Walsh. Their results show complex glacier flow fields and temporal variations, and capture a host of interesting phenomena, including seasonal variations in ice flow, a surge, and dynamical glacier states. The authors present a series of figures showing the resulting velocity fields and report on their results and a few possible implications.

The manuscript is in line with a growing area of research that has great promise to advance our understanding of the cryosphere due to the volume of information available from modern remote sensing platforms. The authors have described a potentially useful method and chosen an interesting study area. As such, this work may be of interest to the TC readership.

However, as discussed below, the paper needs a lot of work in terms of organization, presentation of the methods, and analysis of the results before it can be considered acceptable for publication in the scientific literature. Indeed, I found the paper to be frustrating to read for a variety of reasons, perhaps the main reason being that the authors seem to be trying to claim levels of success, novelty, and

generality that their study does not merit rather than taking a measured approach to demonstrate that their method works, inform readers of the merits of their method/study, and to connect their work to the scholarly literature. The authors almost completely ignore the existing body of work on time-dependent 3D surface velocity fields (which their citations suggest they are aware of), with the exception of references to their own papers and a couple of passing references to a paper (Guo et al., 2020) that presents what appears to be the exact same method the authors are presenting here. As a result, the authors do not place their study into the proper context, nor do they give readers the ability to compare the strengths and weaknesses of the authors' method and those of other methods. But most importantly, the authors do not demonstrate an awareness of the documented challenges of inferring 3D time-dependent velocity fields from satellite data and thus they ignore any consideration of accuracy and precision in the measurements, mixing between the inferred velocity components, and propagation of measurement errors. Perhaps because of this oversight, the authors make claims about the viability of their method that are unsupported by the work presented in the manuscript. Nowhere do the authors test their technique nor make any meaningful attempt to show that their method actually works; rather, we get a few basic equations, some results from actual SAR data (which can only show that the matrix in the authors' method is invertible but cannot show that there is sufficient information to make the inferences the authors are trying to make), and then unsupported claims like (line 164) "[t]he technique in this study is a viable solution for computing 3D flow displacement time series…" The authors never discuss errors nor the limitations and challenges of their method, inferences of multidimensional flow velocities and other such matters that one would expect to find in a scientific publication. In addition, the presented analysis of the results lacks the depth and detail needed to provide new insight into the glaciers being studied or glacier dynamics in general. The current manuscript is very short compared with the vastness and richness of the material the authors are trying to cover, and the supplementary material contains only a single figure, so the authors have plenty of space in the main text and supplement to expand on their methods and findings. More details are provided below.

*Reply: Thank you for sharing your opinion. We answer your questions below.*

**Major points:**

- The title and abstract of the paper suggest the main goal of this manuscript is to present a general method for inferring time-dependent 3D surface velocity fields of glaciers. But the methods are presented as an incremental step from previous work and not described in sufficient detail to merit publication based on the methods alone. Key information that readers need to understand the method and reproduce the results are missing. A few specific points:

o No meaningful validation of the method is provided. In my reading, I did not find any evidence that the method produces accurate 3D velocity fields. I only found reference to one comparison between the authors' results and independent measurements taken over vastly different time scales (Gardner et al., 2018, 2019). This comparison is given in the figure in the supplement (Fig. S1). I would argue that the authors' results differ markedly in all cases from the measurements of Gardner et al., 2018, 2019. Even on the glaciers where the authors claim 'nearly identical results' there are clearly significant disparities

(100s of meters per year, or roughly a factor of 2) between the data sets. The authors provide some plausible suppositions to explain these disparities but never explore these possibilities. Rather, the authors give us offhand references to filtering and other technical matters that can and should be tested and some discussion of how the flow of glaciers is expected to differ between the more recent (2016-2020) observations made by the authors and the multi-decadal average of Gardner et al., 2018, 2019. Indeed, the comparisons between the authors' velocity fields and those of Gardner et al., 2018, 2019, especially over surging glaciers, are scientifically interesting but are not viable tests of the methods presented here. If the authors wish to present a new method, especially one that is as generally applicable as they claim, they need to conduct multiple appropriate tests on synthetic data to test their method under the conditions expected in the natural environment and to convincingly show readers that their method reliably produces accurate results. I cannot stress enough that the authors do none of this work in the current manuscript.

*Reply: The technique presented here is based on 10 years of research and is incremental in nature. Many issues raised by the reviewer have been discussed in previous publications. In the revised version we provide synthetic tests that demonstrate that 3D flow displacement time series can be reconstructed from ascending/descending range/azimuth data very well. See figures below for a preview, figures are numbered 1-9 from top left to bottom right corners, the actual geometry matrix from the manuscript is used: figures 1-3 individual components are reconstructed from a harmonic input without noise (see legend for input signal); 4-6 same as 1-3 but with 10% noise; 7-9 all three components are reconstructed with zero, 10% and 30% noise. As you can see the reconstruction is very good. Similar plots were produced for the 2D tectonic time series in the first MSBAS paper* https://doi.org/10.1111/j.1365-246X.2012.05669.x*.*

[Figure]

o The authors need to provide some discussion of the effects of the viewing geometry on the inferred results. While range and azimuth offsets are orthogonal to one another in existing SAR systems (where the radar line of sight is orthogonal to the platform velocity vector by design), ascending and descending orbits are not orthogonal (as shown in Fig. 1). Thus, the viewing geometries are nonideal and the relative orientation of the orbits and the flow direction of the glacier influences the precision of the inferred velocity components. This geometric effect is amplified by noise in the measurements, particularly the disparity in noise between range and azimuth offsets (where range offsets generally have higher signal-to-noise ratios than azimuth offsets). These effects are discussed to some extent in Minchew et al., 2015, 2017, though the basic ideas are well known from GPS and should be given ample consideration in work of the type the authors are presenting.

*Reply: The effect of viewing geometry has also been discussed in previous papers and are cited here. We also show that the rank of the geometry matrix for a case with one set of velocities is 3. This means that the solution exists, and is unique and stable. See also the synthetic tests above that fully support this statement. We are happy to refer the readers to Minchew et al., 2015 paper for additional discussion by citing it in our manuscript.*

1. Given the orientation of satellite orbits, I expect that there are strong covariances between the inferred vertical and horizontal components of flow. This effect is likely to be most pronounced on Klutlan Glacier, whose flow direction is close to the line-of-sight direction of the radar, meaning that the range offsets pick up most of the horizontal motion and all of the vertical motion while the azimuth offsets provide relatively little constraints. The covariances between horizontal and vertical velocity components are likely to be lowest on Malaspina Glacier, which is flowing more or less south, in a direction that is close to the azimuth direction of the SAR data. This orientation is favorable to inferring 3D velocity fields as the azimuth offsets (which are purely horizontal) are doing most of the work to constrain the horizontal velocity while the range offsets provide information on vertical velocity with little direct influence from the horizontal components. More generally, it is worth noting that the authors' results seem to show that the vertical component of velocity is largest in areas of the glacier that are flowing more along latitude (i.e., east/west flow direction, which is close to alignment with the radar line of sight) than along longitude (which is close to being aligned with the orbits, or azimuth direction), which suggests covariance between the horizontal and vertical components of velocity in the east/west trending flow. Again, all of these topics are discussed in some detail in Minchew et al., 2015, 2017. The take-away is that the authors need to quantify and discuss the geometric and measurement errors for their methods to be publishable and to support any claim of generality.

*Reply: The synthetic tests and the properties of the geometry matrix show that viewing geometry is sufficient for reconstructing any 3D flow displacements time series. As a result, there should be no need to present the covariance matrix for synthetic tests since the results in the figure are very clear, covariance terms are zero.*

1. There is no discussion of errors. The authors provide error bars on the results they present in Fig. 7 but these are merely spatial variances. I would expect a modern paper on geodetic methods to discuss formal errors in the SAR offset fields, how formal errors in the SAR measurements are propagated to the inferred 3D fields, how viewing geometry impacts the results (as just discussed), and any sources of additional error that may not be accounted for in the formal error (e.g., the influences of radar penetration depth and surface moisture, as discussed by Minchew et al., 2015). The lack of treatment of errors is a considerable omission from a methods-focused paper that should be rectified before publication.

*Reply: We discuss errors in the revised manuscript. In the original version we omitted this because at the level that we wanted to present the precision and accuracy is sufficiently high and does not raise any concerns, as it can be seen from time series. All the errors that the reviewer mentioned are not unique to this technique and have been previously discussed by the authors, and others, in earlier work. However, we agree that this is the body of work that is not likely to be familiar to many in this audience. We now provide a synopsis and appropriate references.*

1.  The authors need to provide a reference to the form of the Tikhonov regularization matrix of various orders, some convincing evidence that regularization "is not critical" in this case (cf. line 119), and some discussion of why regularization doesn't make a difference in this case and the conditions under which regularization should matter. Finally, the authors need to be clear how regularization is being applied: are the authors regularizing in space or time or both? Eq. 3 suggests that regularization is only applied in time but this should be clarified in the paper. It would seem that the authors are filtering in space using median filter with a window size larger than the width of some of the glaciers (line 85). A discussion of how this filter and window size were chosen and the effects of a nonlinear (median) filter and such a heavy filter on the final results would be useful.

*Reply: This has been discussed and explicitly shown for the 2D case of tectonic deformation (mathematically identical) in https://doi.org/10.1080/07038992.2017.1344926, and referenced here. Theoretically, it is possible to regularize in time and space but the resulting matrices become so big that they cannot be handled by modern computers. We also provided a discussion on the filter selection.*

- The authors need to discuss the existing literature and place their methods into proper context. Methods for inferring 3D, time-dependent surface velocity fields of glaciers (e.g., Minchew et al., 2015, 2017; Milillo et al., 2017; Guo et al., 2020) and generalized frameworks for inferring multi-dimensional surface velocity time-series (e.g., Greene et al, 2020; Riel et al., 2020) have been published. Only two of these papers are cited in the current manuscript and neither of these are discussed in any meaningful detail. In particular, it appears as though the methods of Milillo et al., 2017, and Guo et al., 2020, are strikingly similar to those presented by the authors in this work. Certainly, they all take the same basic approach of using the small-baseline subset (SBAS) method extended to multiple dimensions.

*Reply: We provide additional background information on the history of multi-dimensional time-dependent surface flow in the revised manuscript. The SBAS is a general linear inversion technique, simply speaking it is V\*t=D.*

This similarity between Milillo et al. (2017), Guo et al. (2020), and the current work needs to be properly explored in the manuscript. In my own reading of Guo et al., I cannot find any real difference between their approaches and those presented here. Indeed, the current authors state on line 170 "[o]ur approach is conceptually similar to the technique of Guo et al. (2020)…[h]owever, our software can additionally compute 1D, 2D…, and 3D flow velocities and displacement time-series and linear rates." It is important here to distinguish methods (and the ideas behind them) from implementation (software and tools); it appears that the authors' claim to novelty is in the implementation and the additional features of their software rather than differences in methodology. In other words, their software bundles several methods that are discussed in previous publications into a single user interface. I find this confusing because the paper seems to be about methodology, which is separate from implementation. I respect the fact that the authors have been working independently on this method for some time and that they should have their efforts rewarded by having the opportunity to publish their work in the scholarly literature. But more needs to be done to clarify the differences (if any)

between the method presented here and those in the existing literature. If there are no meaningful differences, the authors should make that clear and emphasize the unique aspects of their work and results. Thorough testing of the method, exploration of precision and geometric effects, and robust uncertainty quantification (all discussed above) would make this paper unique from Guo et al. and would add value to the authors' methods.

 In contrast to the studies mentioned in the previous paragraph, Minchew et al., (2017), take a markedly different approach to inferring time-dependent 3D velocity fields. I note that this paper is cited in the current manuscript but simply in a list of other papers that apply InSAR to glacier flow; the fact that Minchew et al. (2017) present a method for inferring 3D, time-dependent velocity fields is ignored by the authors, as are the lessons learned about the challenges of accurately inferring 3D, time-dependent velocity fields from SAR data (as discussed above). It would be useful to compare the approach of Minchew et al. (2017) to the authors' approach in an insightful way as the two approaches are quite different. For example, the authors could note that Minchew et al. assume a form for the temporal basis functions based on prior knowledge of the study area, while the current authors invert a matrix for displacement at a given time. The need for prior knowledge in the Minchew et al approach means that this method is not general and so its application is limited to areas where the assumed basis functions should be valid. But the advantage of the Minchew et al approach is interpretability of the results, straightforward connection of the results to the physics of the systems being observed, and robust uncertainly quantification, all things that are lacking in SBAS-based methods like those presented here.

A recent improvement to the work of Minchew et al. (2017) is Riel et al., 2020. I won't fault the authors for not discussing Riel et al. (2020) in the current manuscript as this is very recent work, but it would be good (though, not required) to include a brief discussion of Riel et al. (2020) in the revised draft to add context to the authors' work. Riel et al., 2020, adopt some of the methods of Riel et al., 2014, 2018, and apply them to remote sensing observations of glaciers. From a methodological perspective, this has the effect of generalizing the approach of Minchew et al. (2017) to allow for a generic set of temporal basis functions, from which a sparsity-inducing optimization is used to identify the simplest set of basis functions that describe the data. Here again, the main advantage of this approach is interpretability of the results (and robust uncertainty quantification), which provides the ability to decompose the observed signal into short and long-term variations, and features to ability to constrain transients, secular, and periodic signals. There are certainly limitations to the Riel et al (2020) method, namely that it still requires some level of prior knowledge to provide confidence in the resulting basis functions. The authors' methods may be complementary in the sense that they do not rely on basis functions. Again, the authors' method provides flexibility at the expense of interpretability of the results, where was the Minchew et al. and Riel et al. approaches sacrifice flexibility in the method for enhanced interpretability of the results. An appropriate exploration of these differences in approaches will provide readers with insight into the respective strengths and weaknesses so that they can make informed decisions about which methods may best suit their needs and where improvements can be made.

Riel et al (2020) take the time-dependent methodology further by introducing methods to quantify the propagation of waves through glaciers in the case of generalized methods (Minchew et al., 2017, quantify wave propagation but this is implicit and straightforward in the periodic basis functions).

Importantly, Riel et al (2020) are able to track waves of different frequencies (because separation of frequencies is inherent in the time-series methods) and are able to show that waves with seasonal and annual periods on Jakobshavn Isbræ, Greenland, are dispersive (phase velocity varies with frequency). These waves are likely to be kinematic waves due to the long periods and contemporaneous changes in ice thickness (though there are caveats to this hypothesis and it remains to be tested), so the findings of Riel et al

(2020) are also applicable to the interpretation of the results presented in the current manuscript because the authors report observations of kinematic waves.

The work of Greene et al., 2020, should also be referenced and discussed in the revised manuscript. This is a conceptually different approach to all of those mentioned
 above and warrants comparison to the method being proposed by the authors. In this work, the authors apply a generalized method that disentangles periodic variations from non-periodic variations.

*Reply: The data contains some fixed amount of information, you can use it to extract N parameters with high precision or M parameters with less precision, assuming N<M. Minchew et al. (2017) and Riel et al. (2020) chose to solve for a small number of parameters that can be resolved with high precision. Here we choose to solve for the maximum possible number of parameters, again that are naturally resolved with less precision. Our technique can be used in any environment; the quality of the results is, of course, dependent on the input data. Other techniques are limited in scope as the reviewer said himself in his comment. Based on our reading, the technique of Milillo et al. (2017) is close to Minchew et al. (2017) while the technique of Guo et al. (2020) is closer to ours; we detail those differences in the revision.*

- Without some quantification of the errors and analysis of the covariances between horizontal and vertical velocity components, I am skeptical of the results because it's not clear that they are valid. Indeed, some results strain my physical intuition (which it is essential to note, does not mean the results are wrong and could be my own failing). For example, 600 meters of downward (vertical) displacement at a point (P7, Figure 7g) in 1.5 years seems rather extreme, even during a surge, especially for a glacier with a total flow speed of only ~1000 m/yr. How does this displacement compare with the local ice thickness and are there any observations of dramatic surface lowering of 100s of meters to validate this observation? As discussed above, I am willing to bet that there is a strong tradeoff between the inferred horizontal and vertical components that results in unrealistically large vertical displacements (i.e., horizontal displacement bleeding into the inferred vertical displacement due to the fact that the flow is close to being in line with the radar line of sight, meaning that there is not enough information in the SAR offsets to allow for accurate inferences of the 3D velocity vector). Some validation of these and other results needs to be done as does some attempt to connect the results to physical models (e.g., given mass conservation, is the horizontal flow speed consistent with the inferred vertical speeds?).

*Reply: We understand the reviewer's confusion. It took us some time to comprehend this phenomenon. The flow displacement time series in Figure 10 show the cumulative displacement measured at the*

*surface within an Eulerian pixel. It demonstrates how surface flow has varied over time in 3 dimensions and is not indicative of the surface elevation changes.*

*A good analogy is river rapids where there are areas where water flows almost vertical. There, the vertical flow velocity is fast (kms/hour) and the river depth is less than one meter (can be just a few cm for small streams). Similarly, for glaciers, it is just a matter of inflow and outflow, it has little to do with glacier thickness. We return to this in a later reply statement.*

Some more specific points:

o It is hard to glean insight into the data plotted along the transects because the transects are crudely drawn and undoubtedly cross flowlines. This is likely the source of confounding results like those shown in Figure 5 (Klutlan Glacier) where the plots indicate that there are areas (~20 km, 30 km, and 36 km along the transect) that show very slow velocities (<200 m/yr) throughout the observation period but are surrounded upstream and downstream by fast-flowing regions (velocity > 1200 m/yr). A transect drawn along a flowline shouldn't show such behavior, which is the reason it's a good idea to be careful about where one draws profiles. The authors could provide a clearer view of their results by extracting data along profiles that are consistent with the dynamics of the fluid.

*Reply: We agree, however, the manual extraction of flow lines is subjected to error, we improved this by using flow lines provide by OGGM software ([https://docs.oggm.org/en/latest/flowlines.html](https://docs.oggm.org/en/latest/flowlines.html)). While we observe some discrepancies, we believe it is of a sufficient quality and also reproducible.*

o The possibility that the authors captured a kinematic wave is truly exciting but not very convincing due to the issues discussed above and the qualitative discussion surrounding the observation. Again, it would be useful for the authors to attempt some validation of this observation and some in-depth discussion of the implications. How fast is the wave propagating? Over what distances does it attenuate? Does the wave deform as it travels? Are the amplitude and speed of the wave related to ice thickness, surface slope, or other observables? Is it really a kinematic wave or do longitudinal stress gradients matter? Is it a monochromatic wave or are there multiple frequencies? What is the period (are the periods) of the wave? Even if the authors can't sort out these details, some discussion of basic wave propagation would be valuable, if for no other reason than to point others toward the possibility of using SAR to observe traveling waves of various sorts. Again, I'll point to Riel et al (2020) as an example ofthinking about observing wave propagation with SAR. (And while I'm on this point, the authors need to provide citations to support the idea that one should expect kinematic waves to accompany surges.)

*Reply: This issue was raised by multiple reviewers but it is not given a proper amount of attention in our manuscript. Since it is beyond of the original scope of the manuscript, we decided to remove the discussion about kinematic waves entirely.*

**Minor comments:**

Title: Given the lack of details and analysis of the method (e.g., lack of synthetic tests showing the validity of the method, lack of analysis of errors and limitations of the method, etc), the title seems overly generalized. What the authors have done here is slightly modify existing methods and applied them to a particular case study. Thus, the title needs to be narrowed and case study mentioned.

*Reply: We addressed all these issues in the revised manuscript.*

- Abstract: Similar comment as the previous one about the title. The authors are seriously overselling the generality and novelty of their work. Other authors have published strikingly similar methods as well as markedly different methods that aim to do the same things the authors are attempting. I think the method the authors are presenting is worth publishing eventually, but it is not as novel as the authors seem to be insisting and careful rewording is warranted. Furthermore, without extensive testing of the method (which, as detailed above, has not been done in this manuscript), the argument that the proposed method is generally applicable to the global catalog of SAR data is unsupported.

*Reply: We provided synthetic tests (figure above) that demonstrates that the inversion always produces unique and stable solution, therefore the technique is general. The ability to compute the individual offset maps depends on the region, data availability, and is outside of the scope of the study.*

Line 1: It's unusual to refer to the components of a vector as direction and "intensity." It's more comment to reference direction and magnitude, or in the case of velocity, direction and speed. The authors could clean up the description by simply saying "Glacier velocities adjust to a warming climate…"

*Reply: Corrected.*

Lines 5-6: "We observe seasonal and interannual variations and the maximum horizontal and vertical flow velocity in excess of 1000 and 200 m/yr, respectfully." I don't know what to take from this sentence as there are 4 quantities mentioned (seasonal and interannual, horizontal and vertical velocities) and 5 glaciers in the case study, yet only two speeds mentioned in the sentence. These variations should be given context by comparing them with the mean flow speeds.

*Reply: Corrected.*

Lines 20-22: A couple of comments: 1) The sentence starts with the phrase "Modern remote sensing techniques …include…" but then lists GNSS. I would not consider GNSS to be remote sensing since the quantity being measured requires a station placed on the glacier. 2) UAV is a curious addition to this list as it is not a method for measuring surface velocity. The first three items in the list are specific techniques, whereas UAVs are merely platforms that collect various types of data, all of which could be described as examples of one of the first three items in the list.

*Reply: Corrected.*

Line 32: "non-tidewater" needs to be defined.

*Reply: Corrected.*

Line 35: The authors need to tone down their language and to make a more objective point about the need for studies in Alaska. It is not useful or interesting to say that more studies have been conducted in the ice sheets.

*Reply: Corrected.*

Line 40: The authors need to elucidate why they think the vertical component of velocity is important, not simply assert it. To a good approximation, ice flows along the surfaceslope, so velocity fields given in two horizontal dimensions should capture the vast majority of information about the flow.

*Reply: Corrected.*

Line 44: Rignot et al., 2011, is a 2D, not a 3D, map.

*Reply: Corrected.*

Lines 44-46: One cannot validate remotely sensed maps of velocity using MAI as it is simply another remote sensing method.

*Reply: Corrected.*

Section 2: It seems to me that the order of this section is backward relative to the authors' apparent desire to introduce a general method. If I understand their motivation correctly, the method should be described and evaluated first and separately from the data being used in the case study.

*Reply: Corrected. The method now is presented first.*

Line 79: Give the SLC resolution of Sentinel-1 somewhere in this paragraph. Also worth mentioning that SLCs were collected from both Sentinel-1 satellites with a nominal repeat time of 6 days.

*Reply: Actually, it is mainly 12 days for this region. We provided SLC resolution.*

Line 84: "Such a large window size..." Why? Is it because of the flow speed or something else?

*Reply: We consulted the developers of GAMMA processing software that is used to compute speckle offsets. We were advised that the window that is used to compute the offsets is not uniform, pixels in the centre have larger weights that pixels on edges. The effective resolution is about four times higher than*

*the window size. The process of extraction of offsets, as it is implemented in the software, is not linear. We acknowledge that the spatial resolution is reduced by using such large window. However, this is necessary for extracting the temporal information. Note that the computation of offsets, in general, is not specific in any way to the presented technique here. We also reduced the window to 128x128 pixels in the revised manuscript.*

Line 85: What do the authors mean by "distinct peak?" What do the uncertainties look like (curvature of the correlation surface, difference between peak values, etc.)? It would be useful if the authors would give a few examples of the range and azimuth offsets and associated uncertainties for the study areas in the supplement.

*Reply: To address this we provided signal-to-noise ratio (SNR) maps for all deformation products in the supplementary files. We also computed average uncertainties for our range and azimuth offset maps and for computed velocities. We believe it is not worth providing examples of SNR function for particular pixels because there are so many pixels and images (~10^8), selecting a few will not be representative. However, they look like a figure below.*

[Figure]

Figure S1. 2D cross-correlation plots calculated with a small window (left) and a large window (right)

Line 89: The term 'resolution' appears to be improperly used here. Resolution is a statement of information content but what the authors appear to be referring to is grid spacing.

*Reply: That is correct, we now use grid spacing.*

Eqs. 1 and 2: It's useful to bold variables that indicate vectors and matrices for clarity.

*Reply: We rewrote these equations using capital letters for matrices.*

Line 90: RO and AO should represent sets of range and azimuth offsets. (In other words, it's useful to connect RO and AO to the rho and alpha designation for individual displacements used below.)

*Reply: Corrected.*

Lines 90-91: What are the sizes of matrices L and A in relation to the number of observations?

*Reply: This information is now provided.*

Line 91: It's useful to point out that lambda is a scalar (if, in fact, it is).

*Reply: It is, corrected.*

Line 95: azimuth and incidence angles need to be defined for the non-SAR-expert.

*Reply: Corrected. "The azimuth angle is the compass heading of the satellite, measured from the north; it discerns ascending vs descending orbits. The incidence angle is the angle between the ground normal and the look direction from the satellite; it is one of the acquisition parameters of the side-looking SAR sensor."*

Eq 3: There are several differences between the variables used in the equation and those described above (e.g., upper case to lower case 's', different subscripts on rho and alpha, etc.). These need to be consistent. I don't see where the Delta t variables are defined; they need to be defined for completeness.

*Reply: Corrected. Thank you for checking these fine details.*

Lines 111-112: "Since this method does not make any assumptions about the direction of motion, it provides the optimal solution applicable to any phenomenon." As I mention ad nauseam above, the authors provide absolutely no support for this statement. This is merely an assertion as there is no attempt to show that this method produces accurate or optimal solutions for glacier flow or any other phenomena. Unless the authors can provide proof through rigorous testing of the method, this and other statements need to be removed or clearly qualified with statements like "we hypothesize that..."

*Reply: We believe that the synthetic test does this job. Also, the technique is a basic transformation of a coordinate system.*

Line 113: In my reading, it seems that this is the first mention of coherence in the SAR images. I think a more expanded discussion of noise and errors is required. But at the very least, the authors should make clear to the non-expert reader what they mean by 'coherent pixel'.

*Reply: We removed the word "coherent" to reduce confusion. In the previous version it was used in a broader sense meaning "of a good quality", rather than in a specific sense used in DInSAR (the magnitude of correlation coefficient).*

Line 120: 'visually indistinguishable' from what? Could the authors show us some examples of the effects of regularization in the supplement.

*Reply: In this case, visually indistinguishable from each other. 'Virtually indistinguishable' might be a better term, we used new phrasing in the revision. In the figure below you can see results for P2 (Figure*

*11b) computed using first and second order regularization. We believe it is not worth providing this figure in the supplementary files because it does not convey new information.*

[Figure]

Line 121: Presumably the authors mean that a line was fit in time at each pixel. It is not clear if the authors are average the speed in time (keeping the unit velocity vector fixed) or each velocity component individually, or some other combination. This should be clarified.

*Reply: With the technique presented here, we compute velocities between consecutive SAR acquisitions. Sentinel-1 data is acquired with either a six or 12 day revisit cycle, and velocities are computed for every revisit cycle interval (so-called instantaneous velocities). The flow displacement time series are then reconstructed from these instantaneous velocities. Assuming a 12 day Sentinel-1 revisit cycle, our technique produces 365/12 = ~30 3D velocities per year. Since all these data cannot be presented in a single publication (30 velocities per year x 3D x 4 years ~ 360 figures), as a simplified representation of our results that require only four figures, we choose to compute mean velocities by fitting a line to the flow displacement time series. Along with the mean velocities for each of the four components, we compute their standard deviations and coefficients of determination ($R^2$), which help us understand if the linear model provides a good approximation. For some regions, a linear approximation cannot capture all the complexity of the motion. For these regions, we plot flow displacement time series, which describe instantaneous velocity at each moment in time. Annual or any other duration (monthly, quarterly) velocities can also be computed from our flow displacement time series by aggregating time series at different intervals. The linear rate of each velocity component is computed individually.*

Line 124: I'm not sure what the authors are referring to here, but it sounds like the kind of statement that would be more useful in a figure caption.

*Reply: Corrected.*

Line 133: Does 'mean' mean time-averaged or is there some spatial averaging?

*Reply: Linear model fitted to the entire time series – time-averaged, see comment above.*

Line 145: Report the window size in meters; the number of pixels can be given as a parenthetical, but what matters is how large the window size is in geographic space.

*Reply: Corrected.*

Line 164: See above comments about unsupported claims.

*Reply: Corrected.*

Line 166: What do the authors mean by "with different scales?"

*Reply: At different spatial scales. InSAR is limited to measuring cm-scale displacements, while SPO can measure m-scale displacements. But this statement was deleted for simplicity.*

Line 168: Citation is needed.

*Reply: Corrected.*

Lines 168-170: Unclear what method the authors are referring to.

*Reply: Corrected.*

Lines 170-171: The authors state (emphasis mine) "[o]ur approach is conceptually similar to the technique of Guo et al. (2020) that was built on our previous work." As written, the (italicized) second half of this sentence comes across as petty and unprofessional. All science is built on the work of others. That is its nature and strength. The authors are duly cited by Guo et al. (2020), so phrases like the one italicized above are unnecessary. If the authors decide to keep such statements, it would be best to reword.

*Reply: Our objective was to emphasize that the first version of this manuscript was submitted before Guo et al. (2020) was published. Therefore, it is not based on Guo et al. (2020) work but is similar to it. This statement was deleted.*

Line 181: Again, why is it the case that large correlation windows and "strong filtering" are needed? The authors need to provide some reasoning that his related to the physical characteristics of the area of interest to support and provide insight into this statement. This is especially true given the claims of general applicability of the methods. If readers were to use this method somewhere else, is it necessarily true that large correlation windows and extensive filtering is needed? How will they know? Additionally, the term "strong filtering" needs to be quantified here.

*Reply: Computation of offset maps is not a part of the method that we present. We chose to start from SLC data, but we can also start from the velocity maps produced by someone else. We provide additional guidelines for users who may want to use this technique in other regions.*

Lines 181-182: The fact that surges do affect the results in this study but not those of Gardner et al. means that the comparison is useful for scientific study but not useful for comparing the results between the studies or validating the method.

*Reply: There are no other 3D ice surface velocities from this region or period to compare against. Gardner et al results were provided because they provide 2D resolved velocities from this region, as recommended by the editor during submission.*

Line 184: The fact that glacier flow can deviate significantly from mean flow speeds is well known and documented. This is not something that the authors have demonstrated. So, the statement needs to be reworded or removed.

*Reply: Corrected.*

Line 185: "SAR-derived time series are often compared with GNSS-derived time series..." Some citations are needed.

*Reply: Added.*

Lines 185-186: "...and both techniques are considered conceptually similar; however, there is an important difference..." This looks like a strawman argument and hearsay. The authors need to provide a citation to show that someone thinks that these methods are similar. Otherwise, they should just make their point without acting as if they are addressing some controversy that does not exist.

*Reply: Added references.*

Lines 185-198: I do not see why this discussion is necessary or useful and think it should be removed entirely. Eulerian and Lagrangian coordinates are well understood (as the authors point out in line 191) and well established. Converting between Eulerian and Lagrangian coordinates is also well understood. This paragraph does not add anything new to the topic.

*Reply: We believe this is an important discussion, as we have interacted with other reviewers who are not as astute regarding this issue, or the cryosphere community in general.*

Lines 199-210: I do not understand what the authors are trying to say here. So far as I can tell, they are trying to elucidate the distinction between cumulative displacement and instantaneous velocity. I don't see why that is necessary and think that this entire portion of the discussion should be overhauled or removed.

*Reply: Removed.*

Lines 204-207: I do not understand the point of this discussion and find it very confusing. It seems that the point being made is that flow direction might change in time in unconfined glaciers but probably not in glaciers whose flow is confined by bedrock. That's a pretty basic point that is not advanced by the current study. The connection the authors are trying make to ice streams is tenuous at best because again, this behavior is well known, methods have existed for quite some time to measure the horizontal flow speed and direction, and ice streams are defined as fast-flowing regions that lack strong lateral confinement from the bed topography.

*Reply: We removed this text.*

Lines 207-208: "Moreover, such direction changes are not easily discerned in 2D or 3D resolved velocity fields." This is a completely and utterly false statement that is both unsupported by this work and contradicted by numerous published studies. Plenty of work has shown that SAR-derived velocity unit vectors are accurate to within a couple of degrees in the horizontal (see stacks of papers by Rignot, Joughin, Rott, and other pioneers of this field). I argue that such accuracy is more than sufficient to quantify changes in flow direction.

*Reply: We removed this text as well.*

Lines 219-220: I don't fully understand why this interpretation is correct. In an idealized sense, SAR observations should be sensitive to the motion of the scatterers, not necessarily the surface. Indeed, it is critical when talking about SAR-derived vertical velocities to distinguish vertical velocity from changes in surface elevation. When one is discussing vertical velocities in glaciers that experience surface melt, as all glaciers in this study do, there is an additional complexity due to the dielectric influence of surface melt on the radar signal (penetration depth, or phase center). This is discussed to some extent by Rignot, Echelmeyer, and Krabill (2001) and Minchew et al. (2015). As mentioned above, some careful exploration of this component of the signal is needed here along with some qualification of these findings.

*Reply: We addressed this issue in a separate manuscript. Here, we simply want to point out that the flow of ice occurs at a steeper angle than the surface topography (in the absence of structural boundaries that may force flow in that direction), therefore signifying ice loss. We rewrote this text.*

Line 220: My previous comment notwithstanding, the dynamic state of Malaspina has been reported previously (e.g., Larsen et al., 2015; Muskett et al., 2003) and appropriate citations are needed.

*Reply: Corrected.*

Line 224: Such large changes in vertical velocity over such short distances should be manifest in the surface topography and thus should be tested against surface elevation time-series where available.

ArcticDEM would be a good place to look and would require very little effort on the part of the authors. This could go a long way to showing that the inferred vertical velocities are accurate.

*Reply: Elevation data for our period is limited. Operation IceBridge data, publicly available through the NSIDC, is only current through 2012 and ArcticDEM has only a few points. However, we refer again to the analogy between ice flow and river rapids where vertical and horizontal velocities fluctuate but no change in surface elevation occurs over time, it is all about inflow vs outflow rate change. Another consideration, as our reflective surface remains at some depth the localized changes in elevation rapidly are filled with dry snow that remains transparent for SAR but is not transparent for LIDAR and LANDSAT. Such snow can be moved, for example, by the wind. There is a lot of uncertainty in what SAR-derived results represent. With our technique it is possible to see previously unseen effects of a secondary magnitude, some of them are likely due to SAR-ground interaction. These are the reasons we are cautious about providing in-depth interpretation and instead concentrate on a technique.*

Line 235: The authors need to qualify such statements: "We hypothesize that the downward vertical motion...represents a kinematic wave..." More needs to be done to verify that this is what is recorded in the data (see above discussion).

*Reply: As mentioned above, this issue was raised by multiple reviewers but it is not given a proper amount of attention in our manuscript. Since it is beyond of the original scope of the manuscript, we decided to remove the discussion about kinematic waves entirely.*

Line 242: I'm not sure I agree with the statement that vertical and horizontal motion can only be derived from SAR. I do agree that nadir optical measurements (like Landsat) are insensitive to vertical, but I see no reason why time-dependent altimetry could not be fused into the data to give vertical and horizontal motion (again, the distinction would need to be made between vertical velocity of the ice and vertical motion of the surface). I think it's best for the authors to soften this statement, perhaps saying that SAR is currently the most viable way to infer 3D velocity.

*Reply: We slightly rewrote this statement.*

Line 245: As noted several times above, this qualification needs to be infused throughout the discussion section to provide a more nuanced discussion.

*Reply: Corrected.*

Where can readers download the software?

*Reply: We will provide a link to the repository with software after the manuscript is accepted, in case changes are needed to be made. In any case the software can be accessed by contacting the first author. Mendeley Data repository does not allow modification after submission.*

Where can the processed data from this study be accessed?

*Reply: We will provide a link to the repository with data after the manuscript is accepted, in case changes are needed to be made. Mendeley Data repository does not allow modification after submission.*

Table 1: What is the range of incidence angles in the image? The last column should be labeled as the number of SLC scenes.

*Reply: Usually SAR beams are identified by the mean incidence angle but in the inversion we use precise values for each pixel. Gamma software provides two quantities for each pixel: SAR look vector elevation angle at each map pixel (lv_theta: PI/2 -> up  -PI/2 -> down, the elevation angle is measured between the surface and the look vector pointing at the radar) and SAR look vector orientation angle at each map pixel, 0 -> East  PI/2 -> North). These can be converted to azimuth and incidence angles and these two quantities are provided for each orbit with the data. Corrected heading.*

Figure 1: Add citation for ASTER DEM.

*Reply: Added.*

Figure 2 caption: 'octagon' is referenced but no octagons are in the figure. Delta t_i should be defined as Delta t_i = t_{i+1} – t_i

*Reply: Corrected.*

Figure 3: 'Mean temporal resolution' should be clearly defined and distinguished from the repeat time of a given track.

*Reply: Corrected.*

Figure 4:
Fix the letter labels on the panels. Panel a is missing its letter. There are two letter b's, one of which looks like it is the label for panel c.

*Reply: Corrected. Now figures 6-9.*

The abbreviations for the different glaciers should be defined in the caption. o Why put the date in YYYYMMDD format just to have to explain the format in parentheses? Why not just give the name of the month and the day and year? o What is the local time of the acquisition (this matters for surface melt)?

*Reply: Abbreviations - corrected.  Format of date is preserved for consistency. Time, we believe, is the secondary order effect (compare to the annual cycle) and is not provided.*

o Panel a (top left): Add distance markers along the profile for easy reference to the surrounding figures. Make the scale bar legible by changing colors or adding a box behind it.

*Reply: Corrected. The intensity image was manipulated to display variability in a range pleasant for human eyes. It does not represent any particular units (such as dB), therefore, scale and units are not provided.*

o Panel b: Are these the time-averaged velocities or for a specific time? What data set is used for the contour lines?

*Reply: These are time-averaged velocities. For contour lines we used TerraSAR-x 90 m DEM – now mentioned in the caption.*

o Panel c: same comment as for panel b about specifying that the velocities are time averaged.

*Reply: Corrected.*

o Panel d: What are the white gaps?

*Reply: These were the values above the scale range. Corrected in the revised version.*

Figures 5 and 6: Same comments as for Fig 4.

*Reply: Corrected.*

Figure 7: It would be much easier to interpret these results if the time-series were merged with Figures 4-6 and/or if the points were labeled with the letter of their respective glacier. The 'P' designation does not provide the reader with any information.

*Reply: I guess we continue to use this notation for consistency because it was used in the multiple previous MSBAS-related papers. Since it is not a significant factor and it would require changing most figures, we left the labels unchanged. But we agree with the comment and will be more accurate in our future publications.*

*Dear Reviewer 3,*

*Thank you very much for providing your valuable comments that helped us to significantly improve our manuscript. Below we provide our detailed responses to your questions in italic font. Here is a quick summary of changes:*

- *Addressed reviewers' comments to the best of our knowledge;*
- *Added recent Sentinel-1 data, mainly to investigate what is happening at region P1 at the Malaspina Glacier;*
- *Recomputed offset maps using smaller 128x128 window and the Gaussian filter with 1.3 km 6-sigma width;*
- *Detected another surging Kluane Glacier and analyzed it in-detail;*
- *Used OGGM software to extract flow lines and performed all analysis for selected flow lines;*
- *Simplified interpretation by removing reference to kinematic waves, which require more attention and, which possibly will be addressed in a separate publication.*
- *Provided animations for four AOIs.*

*Best regards,*
*Sergey Samsonov, Kristy Tiampo, and Ryan Cassotto*

This draft proposes a novel method for 3 -D velocity mapping of glaciers using modern spaceborne SAR measurements. Instead of using surface parallel flow constraint, this method combines speckle offset tracking and MSBAS , which is also assisted with regularization. It is further validated with Sentinel-1 data over 5 glaciers in Alaska. The draft is generally well written and the methodology is reasonable. However, there are couple of issues that need to be resolved/expanded in detail.

Major comments:

1. Study area description is better to be extracted from Section I, together with the dataset description in Section 2, to form a separate section, named "Area and Data"

*Reply: We followed your advice and created a separate section "Study Area and Data".*

1. The model description in Section 2 needs to be clearly rewritten and expanded in detail. If sufficient details do not fit the section, they could be added to an appendix then.

*Reply: We rewrote the section "Model" entirely.*

Detailed comments:

Line #13: this is the same sentence as included in the abstract, thus redundant

*Reply: We rewrote the redundant sentence in the Introduction.*

Line #26: SAR-based correlation algorithms not only operate on radar backscatter, but also radar backscatter and phase (complex-valued correlation).

*Reply: Corrected.*

Section I: you introduced multiple methods for velocity mapping (SPO, DInSAR, MAI), but did not mention what specific one you use in this work and why you chose that one. It is clear later in Section 2 that you used SPO, but would be better to motivate it in Section 1

*Reply: We commented in the second paragraph of the Introduction that we use the SPO technique and in the first paragraph of the Model section explained reasons (no need for phase unwrapping, produces range and azimuth results).*

Line #74: the last sentence is also the same as that included in the abstract, i.e. redundant

*Reply: Corrected.*

Line #83-84: the number of pixels also need to be converted to distance in m. I see you want a square sampling interval on the ground by choosing 64 x 16 for Sentinel-1 images.

*Reply: This is approximately equal to 200x200m. This information is now provided in the last paragraph of Study area and Data section.*

Line #84-86: why isn't the correlation window (256 x 256) a square window on the ground to be consistent with the sampling interval. Also, the numbers you chose are equivalently 1km x 4km on the ground. With the 2km wide median filter, you essentially got a spatial resolution around 2km or at least on the order of km. Even though you resampled the products into 200m, this does not justify the spatial resolution is 200m. That said, the spatial resolution is too coarse over fast-moving glaciers, and the resulting spatial pixels are strongly correlated.

*Reply: Such a large window was required to obtain a distinct, statistically-significant peak of the 2D cross-correlation function; its square shape produced similar precision in range and azimuth directions in radar coordinates, and azimuth precision four times lower than range precision in geocoded products. We found that 128x128 (as in the revised version of the manuscript) is sufficient. If we chose to reduce the number of pixels in the azimuth direction M (to make square window on the ground) we would need to increase the number of pixels in the range direction N to keep M\*N=128\*128, but that would affect the precision in an unpredictable way.*

In the revised version we reduced the correlation window to 128x128 pixels and used a Gaussian filter with a width to Gaussian 1.3 km (6-sigma). We recognize the benefits of having high-resolution results. Unfortunately, in this area, the application of a small window produces measurements that are too noisy, and if we only select pixels with high SNR the spatial coverage reduces to nothing. Therefore, we are limited to using a larger window.

We consulted the developers of the GAMMA processing software that is used to compute speckle offsets. We were advised that the window that is used to compute the offsets is not uniform, pixels in the centre have larger weights than those pixels on edges. The effective resolution is about four times higher than the window size. The process of the extraction of offsets, as it is implemented in the software, is not linear. We acknowledge that the spatial resolution is reduced by using such a large window. However, this is necessary for extracting temporal information. Note that the computation of offsets, in general, is not specific in any way to the technique presented here.

To confirm this we computed offsets for a single pair using 64x64, 128x128, and 256x256 correlation windows. In figure 1, below, we present these results before and after filtering. As you can see, while there are differences, overall the signal is consistent. Note that filtering does not reduce the resolution significantly. Again, we found that these processing parameters are optimal for our purposes in this region; however, it does not mean that they would be optimal in other areas.

[Figure]

*Figure 1: (top-left) Seward range, (top-right) Seward azimuth, (bottom-left) Klutlan range, (bottom-right) Klutlan azimuth.*

Eq. 1: you should either cite a reference or explicitly show the proof of this equation. The way it current shows is introducing the equation out of the blue. When details of the proof is involved, you can also put that in an appendix if necessary.

*Reply: While it looks unconventional, it is a basic equation with a meaning similar to V\*t =D, that we believe does not require further derivation. It is used in many SBAS and MSBAS publications and its explicit representation can be deduced from the example (equation 3). We provided clarifications about this equation in the third paragraph of the Model section. Also, the Fialko et al., 2001 paper is cited that explains in detail how azimuth and range offsets are used to solve for the 3D deformation.*

Eq. 1: the matrix/vector notation should be clearly defined by providing the dimension, which should then be related to the number of ascending/descending acquisitions.

*Reply: We provided the following clarification. "In matrix A the number of columns is equal to the number of available SLC images minus 1 multiplied by three, and the number of rows is equal to the total number of range and azimuth offset maps computed from those SLC images." We also explained the size of the matrix in this particular case.*

Eq. 3: this simplified example is not clear. First of all, it is not clear how the Sa and Sr components are coupled in that way. To do so, you probably need a separate graphic illustration besides Fig. 2 or an appendix. If you can find a citation that does exactly the same thing, that would work too. Second, the notation of the rho and alpha elements in the column to the right of the "=" sign were never introduced since they are different from those described in Line #96 -101. Third, the last three elements in the velocity vector only show the northing of velocity at t3 and easting/vertical of velocity at t4. Why is that and what happened to the missing other components at t3 and t4, and what happened to t5?

*Reply: This comes from the geodetic analysis of seismic events and it is very well described in (Fialko et al., 2001; Bechor and Zebker, 2006), which are now referenced in our manuscript. We now explicitly show RO and AO in our simplified example (lines 85-90). Each row in A represents one range or one azimuth offset map. We believe it is now clearer.*

Line #112: "any phenomenon" This is to vague. You need to be specific what type of phenomenon

*Reply: We meant to say any surface motion.*

Line #114: the dimension is 609 x 1014 for the matrix to be inverted. As mentiond above, how to relate these numbers to your total ascending/descending acquisitions. After Eq. 1, you should add a symbolic equation that relates the matrix dimension to the number of radar acquisitions

*Reply: After adding the most recent Sentinel-1 data to the revised version of the manuscript (we wanted to see what is happening at Malaspina Glacier at region P1) the dimensions of matrix became 666×1109. This means that we have 223 SLC images (223-1)\*3=666 and 108 ascending range and azimuth offset maps and 115 descending range and azimuth offset maps = 108+108+115+115=446 and the regularization rows are (223-2)\*3= 663. The total amount of rows is 446+663 = 1109. This now is explained in the Model section.*

Line #116: please report the specific computer setting and runtime for your case

*Reply: For us, it takes about 24 hours of processing time on a single node with 44 cores. An Message Passing Inteface (MPI) version of msbas software has also been developed. The processing time in an MPI version is reduced proportionally to the number of nodes.*

Line #117-120: add a sentence explaining why regularization is needed, and what happens if not included. Any comparison of the horizontal velocity results derived from the 3-D approach with regularization to those from the 2-D methods? Please add some simple analysis

*Reply: It is a somewhat specific and complex issue from the field of linear algebra, which most users probably do not want to know unless they want to develop their own software. There are three theoretically possible cases: the number of equations is less, equal or greater than the number of unknowns. In the equal case, the matrix is square and no regularization is required. In the greater case, the least square solution is found using SVD – this is common in 1D MSBAS (more interferograms than SLCs). In the lesser case (as always in 2D and 3D MSBAS), the solution is found using the truncated-SVD, which is identical to the zeroth-order Tikhonov regularization. If we want to fill the temporal gaps, we need to apply higher order regularization (first and second-orders work equally well in this case). From the computational point of view there is no difference between the 2D and 3D problem. The need for regularization arises because SAR images from different tracks are acquired at different times, which results in more unknowns than equations, producing a rank-deficient, under-determined problem.*

Line #121: what do you mean by "mean linear flow velocity" especially the word "linear"? Regarding "mean", is the 3-year mean value meaningful for those fast-moving glacier terminus? It is expected that such glaciers should have strong seasonal/interannual changes. Probably 1-year mean value is better

*Reply: With the technique presented here, we compute velocities between consecutive SAR acquisitions. Sentinel-1 data is acquired with either a six or 12 day revisit cycle, and velocities are computed for every revisit cycle interval (so-called instantaneous velocities). The flow displacement time series are then reconstructed from these instantaneous velocities. Assuming a 12 day Sentinel-1 revisit cycle, our technique produces about 365/12 = ~30 3D velocities per year. Since all these data cannot be presented in a single publication (30 velocities per year x 3D x 4 years ~ 360 figures), as a simplified representation of our results that require only three figures, we choose to compute mean velocities by fitting a line to the flow displacement time series, which we then divide by the length of our record. Along with the mean velocities for each of the four components, we compute their standard deviations and coefficients of*

determination (R2), which help us understand if the linear model provides a good approximation. For some regions, a linear approximation cannot capture all the complexity of the motion. For these regions, we plot flow displacement time series, which describe instantaneous velocity at each moment in time. Annual or any other duration (monthly, quarterly) velocities can also be computed from our flow displacement time series by aggregating time series at different intervals.

Concerning selecting the length of time to estimate mean flow, a shorter period could certainly be used; however, our aim for this manuscript was to demonstrate the technique used and the overall trends that occurred over 4 years. The flow displacement time series (particularly Figure 11) and text in the discussion address the benefits of short term analyses such as seasonal and inter-annual variability. Also four supplementary animations show instantaneous velocities for each of the studied glaciers.

Line #123: how much coarser resolution is the horizontal one resampled to? And also why is <5m/yr removed? Velocity estimates over slow-moving areas (e.g. < 15m/yr) are usually used to tie the products and calibrate the estimation bias. How did you calibrate your Sentinel-1-derived velocity products?

Reply: The resolution and masking out is performed only for improving visualization (after processing is finished), otherwise, images in the figures get oversaturated with details. We use precise orbits downloaded from the ESA website. We calibrate the offsets by fitting and removing the polynomial model. This approach works well in this region where most areas do not show any motion. The entire Sentinel-1 scene is processed as a whole, and it is cut into small sub-regions only for visualization in the manuscript. Note that the entire Sentinel-1 scene extends far beyond the area shown in the manuscript. The software provides alternative methods of calibration that can be employed in other, more complex, regions (e.g. calibration against multiple reference regions, Z-score). You can see an example of the complete data set at the original resolution in Figure 5 and in supplementary files.

Line #180: "every single range and azimuth offset maps must be coherent at every pixel" what does it exactly mean?

Reply: This means that if a pixel is incoherent on one of the offset maps (e.g. 20190201-20190213) it will be excluded from the processing and all results will have NaN value at that pixel. This approach ensures we used only the highest quality results. In general, our processing software can handle partially incoherent pixels (it will be filled by the regularization); however, in this study, we choose to utilize only pixels coherent in all offset maps so their precision is identical. The technique that utilizes partially coherent pixels will be discussed in the follow-up publications.

Line #181: "large correlation window followed by strong filtering" gives you much lower resolution and spatially correlated pixels. Isn't that problematic for fast-moving glacier terminus? Please comment and justify.

Reply: That is correct. However, it is a necessity to use a large window and filtering as processing with a low correlation window produces very noisy results in this region. This has already been discussed above.

*In the revised version we use smaller window and a filter with the Gaussian window, we found that it performs better for small and large glaciers. Finally, with the exception of a handful of tidewater glaciers (Hubbard, Tsaa, Guyot, and Taan), the majority of glaciers in our study area are land terminating and thus do not experience the rapid flow that typifies tidewater glacier termini.*

---

## Referee Report (RR1)

*Dear Reviewer 3,*

*Thank you very much for providing your valuable comments that helped us to significantly improve our manuscript. Below we provide our detailed responses to your questions in italic font. Here is a quick summary of changes:*

- *Addressed reviewers' comments to the best of our knowledge;*

- *Added recent Sentinel-1 data, mainly to investigate what is happening at region P1 at the*

*Malaspina Glacier;*

- *Recomputed offset maps using smaller 128x128 window and the Gaussian filter with 1.3 km 6-*

*sigma width;*

- *Detected another surging Kluane Glacier and analyzed it in-detail;*

- *Used OGGM software to extract flow lines and performed all analysis for selected flow lines;*

- *Simplified interpretation by removing reference to kinematic waves, which require more*

*attention and, which possibly will be addressed in a separate publication.*

- *Provided animations for four AOIs.*

*Best regards,*
*Sergey Samsonov, Kristy Tiampo, and Ryan Cassotto*

This draft proposes a novel method for 3-D velocity mapping of glaciers using modern spaceborne SAR measurements. Instead of using surface parallel flow constraint, this method combines speckle offset tracking and MSBAS, which is also assisted with regularization. It is further validated with

Sentinel-1 data over 5 glaciers in Alaska. The draft is generally well written and the methodology is reasonable. However, there are couple of issues that need to be resolved/expanded in detail.

Major comments:

1. Study area description is better to be extracted from Section I, together with the dataset description in Section 2, to form a separate section, named "Area and Data"

*Reply: We followed your advice and created a separate section "Study Area and Data".*

2. The model description in Section 2 needs to be clearly rewritten and expanded in detail. If sufficient details do not fit the section, they could be added to an appendix then.

*Reply: We rewrote the section "Model" entirely.*

The equations/notations in Section 2 have been rewritten and also additional references have been added. It looks a bit better however, there are still places that look confusing or unclear. For example,

1. please rewrite the $1^{st}$ sentence of $4^{th}$ paragraph in Section 2 to use notation of M x N to represent the dimension of matrix A. You could explain what M or N means using the number of unknowns/observables, e.g. number of SLC images.
2. You did not answer our comment why the velocity vector in Eq. 3 only included $V_n^3$, $V_e^4$, $V_v^4$. What about the other missing terms at $t_3$, $t_4$, $t_5$?
3. The RO and AO vector definitions using rho and alpha elements need a vector transpose as they are column vectors.
4. Define right after Eq. 3 what those vector/matrix mean and note clearly the dimension using the above-mentioned number of unknowns/observables.
5. You now added the total dimension of 666 x 1109 in Section 2. However, 666 actually corresponds to the column dimension and 1109 the row dimension, which is opposite to the convention of the using row x column. Please reverse the order unless there is a reason for it. Also, you need to put your response to our comment about how these numbers are calculated based on the number of unknowns/observables into the main text.
6. In Fig. 1, and also the simplified example of Section 2, you have 3+4=7 SLC images, so according to the statement ($1^{st}$ sentence of $4^{th}$ paragraph in Section 2), the number of columns should be (7-1)*3=18, which is not equal to the actual number (12) in Eq. 3.
7. Please also put your response to our comment about regularization into the text.
8. It also seems that the actual value of the regularization parameter, lambda, does not matter. Because it got cancelled out in each regularization equation, where there are only two non-zero terms (both terms have lambda's that will be cancelled out) and all zero values for the other terms. Not sure why your reported value of 0.1 matters.

Detailed comments:
Line #13: this is the same sentence as included in the abstract, thus redundant

*Reply: We rewrote the redundant sentence in the Introduction.*

Line #26: SAR-based correlation algorithms not only operate on radar backscatter, but also radar backscatter and phase (complex-valued correlation).

*Reply: Corrected.*

Section I: you introduced multiple methods for velocity mapping (SPO, DInSAR, MAI), but did not mention what specific one you use in this work and why you chose that one. It is clear later in Section 2 that you used SPO, but would be better to motivate it in Section 1

*Reply: We commented in the second paragraph of the Introduction that we use the SPO technique and in the first paragraph of the Model section explained reasons (no need for phase unwrapping, produces range and azimuth results).*

We only found one sentence in Section 2 and did not see that you chose to use SPO in Section 1.

Line #74: the last sentence is also the same as that included in the abstract, i.e. redundant

*Reply: Corrected.*

Line #83-84: the number of pixels also need to be converted to distance in m. I see you want a square sampling interval on the ground by choosing 64 x 16 for Sentinel-1 images.

*Reply: This is approximately equal to 200x200m. This information is now provided in the last paragraph of Study area and Data section.*

Line #84-86: why isn't the correlation window (256 x 256) a square window on the ground to be consistent with the sampling interval. Also, the numbers you chose are equivalently 1km x 4km on the ground. With the 2km wide median filter, you essentially got a spatial resolution around 2km or at least on the order of km. Even though you resampled the products into 200m, this does not justify the spatial resolution is 200m. That said, the spatial resolution is too coarse over fast-moving glaciers, and the resulting spatial pixels are strongly correlated.

*Reply: Such a large window was required to obtain a distinct, statistically-significant peak of the 2D cross-correlation function; its square shape produced similar precision in range and azimuth directions in radar coordinates, and azimuth precision four times lower than range precision in geocoded products. We found that 128x128 (as in the revised version of the manuscript) is sufficient. If we chose to reduce the number of pixels in the azimuth direction M (to make square window on the ground) we would need to increase the number of pixels in the range direction N to keep M\*N=128\*128, but that would affect the precision in an unpredictable way.*

*In the revised version we reduced the correlation window to 128x128 pixels and used a Gaussian filter with a width to Gaussian 1.3 km (6-sigma). We recognize the benefits of having high-resolution results. Unfortunately, in this area, the application of a small window produces measurements that are too noisy, and if we only select pixels with high SNR the spatial coverage reduces to nothing. Therefore, we are limited to using a larger window.*

*We consulted the developers of the GAMMA processing software that is used to compute speckle offsets. We were advised that the window that is used to compute the offsets is not uniform, pixels in the centre have larger weights than those pixels on edges. The effective resolution is about four times higher than the window size. The process of the extraction of offsets, as it is implemented in the software, is not linear. We acknowledge that the spatial resolution is reduced by using such a large*

*window. However, this is necessary for extracting temporal information. Note that the computation of offsets, in general, is not specific in any way to the technique presented here.*

*To confirm this we computed offsets for a single pair using 64x64, 128x128, and 256x256 correlation windows. In figure 1, below, we present these results before and after filtering. As you can see, while there are differences, overall the signal is consistent. Note that filtering does not reduce the resolution significantly. Again, we found that these processing parameters are optimal for our purposes in this region; however, it does not mean that they would be optimal in other areas.*

[Figure]

*Figure 1: (top-left) Seward range, (top-right) Seward azimuth, (bottom-left) Klutlan range, (bottom-right) Klutlan azimuth.*

1. Based on what you clarified, using a large window might be okay for your area. But using a filter with larger width (km) is not recommended. Are you saying you replaced the previous median filter (2km width) with a Gaussian one (1.3km 6-sigma)? If so, 3-sigma Gaussian is roughly 650m, which might be okay but still a bit large.
2. From above figure (bottom right), it seems using your new window of 128 x 128 with filtering gives quite different results compared to 64 x 64 without filtering. So the question arises: the large window might be insufficient for this area and also the filter width might be too coarse.
3. You probably want to mention this as a limitation of the current processing and discuss how to improve the results in the future.

Eq. 1: you should either cite a reference or explicitly show the proof of this equation. The way it current shows is introducing the equation out of the blue. When details of the proof is involved, you can also put that in an appendix if necessary.

*Reply: While it looks unconventional, it is a basic equation with a meaning similar to V\*t =D, that we believe does not require further derivation. It is used in many SBAS and MSBAS publications and its explicit representation can be deduced from the example (equation 3). We provided clarifications about this equation in the third paragraph of the Model section. Also, the Fialko et al., 2001 paper is cited that explains in detail how azimuth and range offsets are used to solve for the 3D deformation.*

Eq. 1: the matrix/vector notation should be clearly defined by providing the dimension, which should then be related to the number of ascending/descending acquisitions.

*Reply: We provided the following clarification. "In matrix A the number of columns is equal to the number of available SLC images minus 1 multiplied by three, and the number of rows is equal to the total number of range and azimuth offset maps computed from those SLC images." We also explained the size of the matrix in this particular case.*

==Please refer to our above comments on rewriting this sentence and also the problem of applying this sentence in calculating the dimension for the particular case.==

Eq. 3: this simplified example is not clear. First of all, it is not clear how the Sa and Sr components are coupled in that way. To do so, you probably need a separate graphic illustration besides Fig. 2 or an appendix. If you can find a citation that does exactly the same thing, that would work too. Second, the notation of the rho and alpha elements in the column to the right of the "=" sign were never introduced since they are different from those described in Line #96 -101. Third, the last three elements in the velocity vector only show the northing of velocity at t3 and easting/vertical of velocity at t4. Why is that and what happened to the missing other components at t3 and t4, and what happened to t5?

*Reply: This comes from the geodetic analysis of seismic events and it is very well described in (Fialko et al., 2001; Bechor and Zebker, 2006), which are now referenced in our manuscript. We now explicitly show RO and AO in our simplified example (lines 85-90). Each row in A represents one range or one azimuth offset map. We believe it is now clearer.*

==As mentioned above, you did not answer our comment why the velocity vector in Eq. 3 only included $V_n^3$, $V_e^4$, $V_v^4$. What about the other missing terms at t3, t4, t5? Also, as mentioned above, please denote number of unknowns/observables (e.g. number of SLC images as N) and use N to express each vector/matrix dimension right after Eq. 3. This is pretty standard way of introducing vector/matrix notation in writing scientific articles.==

Line #112: "any phenomenon" This is to vague. You need to be specific what type of phenomenon

*Reply: We meant to say any surface motion.*

==Please reflect that change not only in the current response but also in the revised manuscript, otherwise it is still confusing to others.==

Line #114: the dimension is 609 x 1014 for the matrix to be inverted. As mentiond above, how to relate these numbers to your total ascending/descending acquisitions. After Eq. 1, you should add a symbolic equation that relates the matrix dimension to the number of radar acquisitions

*Reply: After adding the most recent Sentinel-1 data to the revised version of the manuscript (we wanted to see what is happening at Malaspina Glacier at region P1) the dimensions of matrix became 666×1109. This means that we have 223 SLC images (223-1)\*3=666 and 108 ascending range and azimuth offset maps and 115 descending range and azimuth offset maps = 108+108+115+115=446 and the regularization rows are (223-2)\*3= 663. The total amount of rows is 446+663 = 1109. This now is explained in the Model section.*

As mentioned above, you need to move your response to the revised text as well. Once you define number of unknowns/observables as N or M as suggested above (e.g. number of SLC images as N), it is pretty straightforward to make this calculation by substituting N=223.

Line #116: please report the specific computer setting and runtime for your case

*Reply: For us, it takes about 24 hours of processing time on a single node with 44 cores. An Message Passing Inteface (MPI) version of msbas software has also been developed. The processing time in an MPI version is reduced proportionally to the number of nodes.*

Line #117-120: add a sentence explaining why regularization is needed, and what happens if not included. Any comparison of the horizontal velocity results derived from the 3-D approach with regularization to those from the 2-D methods? Please add some simple analysis

*Reply: It is a somewhat specific and complex issue from the field of linear algebra, which most users probably do not want to know unless they want to develop their own software. There are three theoretically possible cases: the number of equations is less, equal or greater than the number of unknowns. In the equal case, the matrix is square and no regularization is required. In the greater case, the least square solution is found using SVD – this is common in 1D MSBAS (more interferograms than SLCs). In the lesser case (as always in 2D and 3D MSBAS), the solution is found using the truncated-SVD, which is identical to the zeroth-order Tikhonov regularization. If we want to fill the temporal gaps, we need to apply higher order regularization (first and second-orders work equally well in this case). From the computational point of view there is no difference between the 2D and 3D problem. The need for regularization arises because SAR images from different tracks are acquired at different times, which results in more unknowns than equations, producing a rank-deficient, under-determined problem.*

Even though only some readers might be interested in this topic, you still need to include it in the text to be complete. Also it is not trivial and widely used in the literature on ice velocity mapping.

Line #121: what do you mean by "mean linear flow velocity" especially the word "linear"? Regarding "mean", is the 3-year mean value meaningful for those fast-moving glacier terminus? It is expected that such glaciers should have strong seasonal/interannual changes. Probably 1-year mean value is better

*Reply: With the technique presented here, we compute velocities between consecutive SAR acquisitions. Sentinel-1 data is acquired with either a six or 12 day revisit cycle, and velocities are computed for every revisit cycle interval (so-called instantaneous velocities). The flow displacement time series are then reconstructed from these instantaneous velocities. Assuming a 12 day Sentinel-1 revisit cycle, our technique produces about 365/12 = ~30 3D velocities per year. Since all these data cannot be presented in a single publication (30 velocities per year x 3D x 4 years ~ 360 figures), as a simplified representation of our results that require only three figures, we choose to compute mean*

*velocities by fitting a line to the flow displacement time series, which we then divide by the length of our record. Along with the mean velocities for each of the four components, we compute their standard deviations and coefficients of*

*determination (R2), which help us understand if the linear model provides a good approximation. For some regions, a linear approximation cannot capture all the complexity of the motion. For these regions, we plot flow displacement time series, which describe instantaneous velocity at each moment in time. Annual or any other duration (monthly, quarterly) velocities can also be computed from our flow displacement time series by aggregating time series at different intervals.*

*Concerning selecting the length of time to estimate mean flow, a shorter period could certainly be used; however, our aim for this manuscript was to demonstrate the technique used and the overall trends that occurred over 4 years. The flow displacement time series (particularly Figure 11) and text in the discussion address the benefits of short term analyses such as seasonal and inter-annual variability. Also four supplementary animations show instantaneous velocities for each of the studied glaciers.*

It is now clear to us. However, it is strongly recommended to rename the term "mean linear flow velocity". Alternatively, you should add a few more sentences from the above response to the main text otherwise, the readers might still feel confused and thought it was a statistical averaging mean value.

Line #123: how much coarser resolution is the horizontal one resampled to? And also why is <5m/yr removed? Velocity estimates over slow-moving areas (e.g. < 15m/yr) are usually used to tie the products and calibrate the estimation bias. How did you calibrate your Sentinel-1-derived velocity products?

*Reply: The resolution and masking out is performed only for improving visualization (after processing is finished), otherwise, images in the figures get oversaturated with details. We use precise orbits downloaded from the ESA website. We calibrate the offsets by fitting and removing the polynomial model. This approach works well in this region where most areas do not show any motion. The entire Sentinel-1 scene is processed as a whole, and it is cut into small sub-regions only for visualization in the manuscript. Note that the entire Sentinel-1 scene extends far beyond the area shown in the manuscript. The software provides alternative methods of calibration that can be employed in other, more complex, regions (e.g. calibration against multiple reference regions, Z-score). You can see an example of the complete data set at the original resolution in Figure 5 and in supplementary files.*

Please objectively report your above calibration approach and clearly state that this is a limitation of the current processing chain in the main text. It seems too empirical and will be problematic for fast moving glacier areas. We would like to see some validation of the velocity results by comparing to other reference velocity measurements with some accuracy or error analysis, which is completely missing in this work.

Line #180: "every single range and azimuth offset maps must be coherent at every pixel" what does it exactly mean?

*Reply: This means that if a pixel is incoherent on one of the offset maps (e.g. 20190201-20190213) it will be excluded from the processing and all results will have NaN value at that pixel. This approach ensures we used only the highest quality results. In general, our processing software can handle*

*partially incoherent pixels (it will be filled by the regularization); however, in this study, we choose to utilize only pixels coherent in all offset maps so their precision is identical. The technique that utilizes partially coherent pixels will be discussed in the follow-up publications.*

Line #181: "large correlation window followed by strong filtering" gives you much lower resolution and spatially correlated pixels. Isn't that problematic for fast-moving glacier terminus? Please comment and justify.

*Reply: That is correct. However, it is a necessity to use a large window and filtering as processing with a low correlation window produces very noisy results in this region. This has already been discussed above.*

*In the revised version we use smaller window and a filter with the Gaussian window, we found that it performs better for small and large glaciers. Finally, with the exception of a handful of tidewater glaciers (Hubbard, Tsaa, Guyot, and Taan), the majority of glaciers in our study area are land terminating and thus do not experience the rapid flow that typifies tidewater glacier termini.*

As mentioned above, although it seems to work for your area (note it is not convincing without a formal error analysis), you should explicitly add this as a limitation of the current processing routine, and explain how to improve it in the future for fast glacier outlets. The current revision of the manuscript still lacks a formal discussion about current limitations and how to improve.

---

## Referee Report (RR2)

Review of "Measuring the state and temporal evolution of glaciers in Alaska and Yukon using SAR-derived 3D time series of glacier surface flow" by Samsonov, Tiampo, and Cassotto

The authors discuss a method for inferring time-dependent 3D surface velocity fields from synthetic aperture radar (SAR) data and apply this method to data collected from Sentinel I in 2016-2020 over five outlet glaciers in Alaska: Agassiz, Seward, Malaspina, Klutlan, and Walsh. Their results show complex glacier flow fields and temporal variations, and capture a host of interesting phenomena, including seasonal variations in ice flow, a surge, and dynamical glacier states. The manuscript is in line with a growing area of research that has great promise to advance our understanding of the cryosphere due to the volume of information available from modern remote sensing platforms. The authors have described an interesting and useful method and focus on an area where glaciers are exhibiting fascinating dynamic behavior. As such, this work will likely be of interest to the TC readership.

This revised version of the manuscript is much improved from the original version. The authors added more context and details for their methods, attempted a minimal synthetic test to show that the method works, and improved the presentation and discussion of their results. The tone and precision of the writing in this draft better positions the work in the broader context of remote-sensing methodology and scientific knowledge. Overall, I enjoyed reading this draft and think that it is moving toward being suitable for publication, though I have some comments below that may be useful to consider.

- I think the authors need to do more to distinguish the methodology presented in this work and that of Guo et al., 2020. The authors merely mention Guo et al. in the introduction and say that they are presenting here an 'independently developed version of the algorithm' (line 46) in this manuscript. Some discussion of the differences between this algorithm and Guo et al. are needed as the current wording suggests that the authors developed an identical algorithm. If the algorithms are identical, the authors need to say so, otherwise they risk confusion for readers looking to implement or further develop the methods. In line 46, the authors point out that their software contains options to call other methods (published elsewhere) and Tikonov regularization schemes, but this is irrelevant to the distinction in the methods and algorithms. If there are no significant differences between these methods and those of Guo et al., 2020, it would seem that the presented method is not 'novel' (as stated in lines 119 and 326), and such statements should be removed.
- The synthetic tests appear a bit perfunctory and are certainly not as generally applicable as the authors' language suggests. At the very least, the sythetics need to be explained more thoroughly and in greater detail. I have a few comments:
  - One of the challenges of capturing the signals that the authors attempt to capture is that the three components of the velocity vector defined in an east-north-up (or other geographically referenced coordinate system) are not independent of one another. Rather, one would expect that the components of the velocity vector covary as they are representing the 3D flow of a glacier that is responding to some combination of internal and external forcing. So, while it's an interesting exercise to evaluate whether the method is capable of inferring time-varying signals in different velocity components that are unrelated to one another (i.e., different periods of variability), it's not a realistic test of a method meant to be applied to the natural environment. In other words, it's not that hard to infer different components when their time-varying functions are orthogonal. The challenge is in separating variability in the individual components when it is the speed (magnitude of the velocity vector) of the glacier that is varying with some given frequency and amplitude. Such a test has not been conducted and needs to be to show that the method works.
  - The covariance matrix (mentioned on lines 140 and 144) is never defined in the paper and needs to be if it's to be discussed. This is especially true given the authors finding that geometry doesn't matter in their method based on the covariance terms taking on a

value of zero. This is a surprising finding that contradicts decades of work in GPS positioning and other work in inferring multi-dimensional surface velocity fields from SAR data (as referenced in a previous review), so a little more discussion would be useful as would an explanation for why it is the authors' method doesn't suffer the same challenges as well-established methods that attempt to do essentially the same thing.

- o Lines 142-143: Why would the rank be 3 if you have ascending and descending range and azimuth offsets? These represent 4 unique viewing geometries, so one would expect the rank to be 4.
- o The authors' mention of the tensor rank seems to indicate a misunderstanding of the point of geometric influence and the role of the covariance matrix. The rank simply shows that there are 4 unique viewing geometries, which is enough information to invert for 3D velocity field. That is obvious and has never been in question. Rather, the question lies in the accuracy and precision of the inferred components. In other words, given a finite signal-to-noise ratio, is there enough information to constrain the 3D velocity vector components in time? The authors have shown that the answer is maybe, but the physical contrivance of their synthetic tests (as noted in my comment above starting with 'One of the challenges…') leaves the question open. The rank of matrix A can only provide a negative answer to this question as it is merely a necessary (not sufficient) condition for inferring the information that the authors purport to infer.
- o Further to my point about the covariance matrix, one of the major shortcomings of the methodology presented here is the lack of any formal uncertainties. What I mean is that it's possible to compute the uncertainties in the offset fields, and in a methodology as developed as MSBAS, these uncertainties should be carried through to some formal error estimate for the resulting velocity fields, and this is where imperfect observational geometries (as is virtually always the case with satellite observations due to the non-orthogonal angles between orbits) will amplify errors. Thus, spatial variability, as quantified here by the authors, is better than nothing, but not as good as formal UQ.

Minor comments:
- • I still fail to see the value in the discussion contained in the paragraph beginning in line 275 as the distinction between Eulerian and Lagrangian coordinates is well established, but perhaps the authors are aware of some related controversy that needs to be addressed. The wording in this manuscript (line 281) suggests that no such controversy exists for the TC readership (and I know of now such issues), so I still contend that this should be removed from the paper. That said, the authors are clearly intent on making this well-known point, and I won't bring it up again. I'll simply end by saying that if the authors are intent on making this point, they should at least say something about the conversion between Eulerian and Lagrangian coordinates.
- • A few sentences in this manuscript are identical to those found in Samsonov et al. 2021 (Remote Sensing of the Environment). This isn't a major issue as these are minor sentences that give context, but a bit of editing will avoid the appearance of copy-paste between two published works. This comment should not be taken by the editor or anyone else as an ethical issue, merely a logistical detail as the published manuscript was probably written around the same time as this one and it's easy for these things to happen.

---

## Author Response (AR2)

*Dear Dr. MacGregor*

*Thank you very much for the opportunity to revise our manuscript. We successfully addressed all the comments. Below, please find our response to your comments in italic font.*

*Best regards,*
*Sergey Samsonov, Kristy Tiampo, and Ryan Cassotto*

Editor Decision: Publish subject to revisions (further review by editor and referees) (10 Jun 2021) by Joseph MacGregor
Comments to the Author:
Dear Dr. Samsonov et al.,

I've now received and reviewed three thorough reports from referees concerning your revised MS on a new derivation of 3-D glacier flow in Alaska from SAR from the same three referees. This is unusual and indicates the broad interest in the topic you've presented. However, all three indicate that major revisions are required, and no referee felt that any element of the manuscript could yet be rated excellent, so further revision is clearly still needed.
*Reply: We also feel that the reviewers' reports are especially thorough. We are very thankful to the reviewers for the effort and time.*

I agree that the manuscript is currently a bit dissonant with unusual complexity in the SAR methods and relatively little detailed discussion of the sometimes large signals observed.
*Reply: We disagree to some extent about the complexity of the technique because all the equations are presented explicitly and can be programmed directly from the manuscript. The real challenge was the huge amount of data and various programming challenges that we successfully solved (for example, the matrices are huge). Our original intention was to introduce the software and let the community use it, we did not want to go into the details, because from the theoretical point of view the problem is very simple, it is the computational aspects that are very complicated. Nevertheless, we followed the reviewer's guidance and revised accordingly.*

While reviewer #1 suggest breaking out the manuscript so that the glacier signals can be explored more fully, I leave it to the authors' discretion as to how to proceed.
*Reply: The strength of our technique is in the ability to produce regional maps (again, the amount of processing is huge). While we cover a lot of glaciers, we focused our study on only four glaciers, three of which experienced surge behavior during our observation period. Because of the added benefit of observing vertical flow from 3D SAR observations of surging glaciers, we believe it makes sense to present all of them in the same manuscript.*

Please make a clear decision as to whether to highlight some of the unusually large signals observed and explain their possible physical origins more, or clarify whether they may be due to uncertainty in the method and require further future investigation.

*Reply: We do not observe unusually large signals. For example, in Figures S7-S14, we compared our SAR-derived results with the Landsat-derived (from ITS_LIVE) results during 2017 and 2018. Both results are in a good agreement and the Landsat-derived results most of the time show slightly larger velocities, which we explained (Landsat data is acquired mainly during the sunny (summer) days when the surface flow is above the average rate). The largest velocity that we observe here is about 3000 m/year or less than 10 m/day. These are very typical maximum velocities for glaciers.*

A particular concern of referees #2 and #3 is the lack of a formal uncertainty analysis, which I agree is required for this relatively new combination of method + application.

*Reply: All velocities are now provided with standard deviations (Figures S1-S3) and the coefficients of determination $R^2$, Figures (S4-S6). Previously this information was not shown in the manuscript but was included in the supplementary files (to be submitted to the data repository). You can also see error bars in time series (Figures 11-12). To the best of our knowledge, it is very unusual to have an analytical representation of the uncertainty analysis in numerical computations. We have not seen anything like this for SBAS-derived manuscripts, and, even if it was available it would not be of any benefit, because the precision of the input data is the same for all pixels, the transform is also the same, therefore, the output data will also have the same precision. Clearly, one precision value for all pixels does not provide much value. Instead, we use statistical analysis to estimate precision. The average and maximum values of standard deviations are also reported in the discussion section. Additionally, we use two independent datasets - from ascending and descending orbits, which cross-validate each other. We also provide a more detailed analysis in the synthetic data section and believe that the resulting discussion is sufficient. We now also compare our velocities ts with the Landsat-derived velocities (Figures S7-18) during 2017, 2018 and the entire period.*

This also dovetails into reviewer #1's concern indicating that a simple smoothing might be just as good as the regularization.

*Reply: As noted in our reply, the truncated SVD method is the standard method employed for inversion of DInSAR pixels with uneven time intervals today, and has been so for many years, because of the ill-posed nature of the problem combined with the very large date set. To the best of our knowledge regularization is the only available mathematical technique for solving ill-posed problems (Truncated SVD is a particular case of regularization).*

For these and other concerns, I expect them all to be addressed fully in a revised version, and I note that reviewer #2 indicated that some of their earlier comments were not fully addressed.

*Reply: We double-checked that all comments are addressed.*

Figures 6-9 are overall quite effective, but I would like to see P1-9 labeled in the lower panels. In Figures 10 and 11, please indicate which glacier/distance P1-9 are associated with (e.g., Malaspina

terminal lobe for P1). As they stand, the signals shown are quite complex and need some context without having to refer back to previous figures.

*Reply: We made these changes in the revised manuscript.*

Regards,
Joe MacGregor
NASA/GSFC

*Dear Reviewer,*

*Thank you very much for providing your valuable comments that helped us to significantly improve our manuscript. Below we provide our detailed responses to your questions in italic font.*

*Best regards,*
*Sergey Samsonov, Kristy Tiampo, and Ryan Cassotto.*

Review of Measuring the state and temporal evolution of glaciers using SAR-derived 3D time series of glacier surface flow

By Sergey Samsonov, Kristy Tiampo, and Ryan Cassotto

This paper presents a technique for producing times series of the 3D glacier motion using ascending/descending data.

I feel this paper still has a lot of issues.

In particular, the discussion of the glacier behavior is pretty shallow. There are lots of things going on in the data, but the discussion is very weak. Some of it would really benefit from trying to decompose the surface parallel and surface elevation change signals. I have made some comments, but really it just needs some more work to express the main points better.

*Reply: The strength of our manuscript is in presenting a new processing technique applied to a new region that produces new and interesting results. We agree that data shows so many interesting signals, which only emphasizes the strength of this technique. However, a detailed description of these signal is beyond the scope of this manuscript. Some of them will be discussed in separate studies that are underway (led by graduate students).*

*The interesting signals in the data sets are exactly the impression we want to make with our manuscript. Processing SAR data for studying just one or a few small glaciers is how it was done in the past. With the global coverage provided by Sentinel-1 (and forthcoming NISAR), increased availability of inexpensive processing power (e.g. clusters), and software that can handle the processing of these large data sets we can produce deformation products on a regional or even continental scale. Here, we are simply trying to introduce the technique and the results produced on such large spatial scales.*

With no context, a lot of the data look wonky. For example, a 200-m displacement of the surface at P9 is a fairly fantastic signal that warrants a lot more explanation. Perhaps not if it's a surge bulge. But how about finding some independent confirmation (Arctic DEM for example).

*Reply: As mentioned in the discussion, many of the glaciers in our study area (Kluane, Klutlan, Walsh) were, in fact, surging during part of our observation period (the Sentinel-1 record). And we provided citations from earlier studies that verified the onset of surge activity from 2D flow velocity records using sensors that precede Sentinel-1 measurements. We also searched the ICESat-2, Operation IceBridge (OIB), and ArcticDEM records for surface elevation measurements coincident with surging activity. Unfortunately, records of surface elevation change do not exist during the interesting surge activity as it either precedes the launch of ICESat-2, are not coincident with OIB flight operations, or were not acquired by commercial satellites that form the basis of ArcticDEM time-tagged strip data. Therefore, comparison with surface elevation data from the time periods of*

*active surge activity is not possible due to a lack of data. In Figures S7-S14, we compared our SAR-derived results with the Landsat-derived (from ITS_LIVE) results during 2017 and 2018. Both results are in a good agreement and the Landsat-derived results most of the time show slightly larger velocities, which we explained (Landsat data is acquired mainly during the sunny (summer) days when the surface flow is above the average rate). The largest velocity that we observe here is about 3000 m/year or less than 10 m/day. These are very typical maximum velocities for glaciers.*

Really this would be a better paper if it stuck to results from one (maybe two glaciers) and explained those well. Then the remaining results could be turned into a second more science rather than technique-focused paper.

*Reply: Here the focus is on the design and presentation of the advanced processing techniques. We believe that this is where our contribution to the science community is the strongest.*

As a point of curiosity, I really wouldn't mind seeing a discussion of how the results would be superior to taking the staggered ascending/descending offset and interpolating them to common times, and then applying the basic (e.g., Gray) equations at those times to get a solution. Either way, the velocity solution at a point in time is just a linear combination of the offsets that went into it. With no regularization, I didn't go through all of the math, but my intuition says the results shouldn't different greatly (or even be identical) to a simple 2-pt linear interpolator (or perhaps some higher-order interpolation). And it computationally, this latter approach would be far faster. A simple smoothing filter could be applied in place of the regularization. This could be coded up in an afternoon and compared with the results.

*Reply: There are a number of issues with implementing offset interpolation to common times approach, below are just a few of them:*

- *Measurements contain error – interpolating noisy data is far from obvious;*
- *Some or many measurements can be missed – interpolating over variable time periods (gaps) is far from obvious;*
- *Some reversible signals (e.g. a shift in trend) can be captured by only some data sets – interpolating data that does not capture reversible signals will miss those signals;*
- *Most importantly, the interpolation doubles the amount of input data so the inversion will actually take twice longer. For example, if you have 100 SAR images from one orbit and another 100 SAR images from another orbit, you will get 99 range and azimuth offset maps from each orbit. If you decide to interpolate to the intermediate times you get 199 range and azimuth offset maps from each orbit. That is twice the amount of data. The number of rows in the transform matrix doubles, which is computationally more expensive.*

*If one decides to approach this problem by writing equations that govern this process one will end up with the equations that we provided. Then, one can try to solve these equations using unconventional computational methods, e.g. testing various interpolation and smoothing methods, or one can use the conventional, well-established mathematical approach for solving under-determined and over-determined problems, which requires computation of the SVD.*

*In case of interpolation, how would you choose the optimal parameters from the magnitude of possibilities (e.g. interpolation length, smoothing filters)? It is fast to perform one interpolation, what about interpolating it 250 time periods x four data sets x 10,000,000 pixels ~ 10^10 (10 billion) times? Then you still have to solve the inverse problem with twice the number of rows (which takes as much time as SVD in our case) and then still perform filtering. In addition, truncated SVD is the standard method employed for DInSAR time series inversion today, and has been so for many years, because of the ill-posed nature of the problem combined with the very large date set. While investigation of modifications of the inversion method and interpolation with advanced processing techniques might prove worthwhile in the future, it is outside the scope of this work.*

I have made some specific points

Specific Points: Line numbers refer to the marked-up version.

Abstract: "no single technique" I disagree with this statement. Laurence Gray published such ability - 1.Gray, L. Using multiple RADARSAT InSAR pairs to estimate a full three-dimensional solution for glacial ice movement. Geophys Res Lett 38, n/a-n/a (2011). One can certainly argue that this paper improves on the technique, but times series of estimates can be derived using the method he developed. The method presented here improves a bit on lining up things in time, but only to the extent that built-in assumptions about the rate of change apply. I think a fairer statement for the abstract would be "We have improved upon earlier methods to measure the evolution of surface flow …"

*Reply: We modified the abstract as recommended.*

Abstract: The abstract makes it sounds a bit like 30 years of data have been ignored for want of this technique. There are massive gaps in the record, which is why the measurements have not been made. Please tweak the text to not oversell the available data.

*Reply: We corrected an abstract by removing "nearly 30 years of".*

Line 27 "m-scale" is ambiguous as the way it's worded sounds like it's referring to the horizontal scale. Change to "comprise displacement measurements sub-meter to meter-scale precision with … and cm-scale precision with …". (sensors like TSX do considerably better than m-scale).

*Reply: Corrected.*

Line 71. How about inserting "open source" before "software" then delete "It is provided to the …."

*Reply: Corrected.*

Line 132 remove "compass" as it implies magnetic north when I believe true north is meant. "satellite heading" would also save some words.

*Reply: Corrected.*

Line 133 "ground normal" not clear if you mean with respect to the DEM or the ellipsoid. Especially since it's said to be a sensor parameter, it sounds like the ellipsoid. Please be clear about what is meant as it's a rather critical distinction.

*Reply: Corrected, replaced with "nadir".*

Equation 3. Please check V column its – not clear why V3 for N, but V4 for e & v.

*Reply: Thank you, these were grammatical errors and they are now corrected.*

Line 169: given you present a method in Equation 3 with a constraint. Could you be clear about what you mean by "unconstrained"

*Reply: Equation 3 uses a numerical constraint (slowly changing velocities). However, we refer to the geometrical constraint – such as "surface-parallel flow" studied in our previous manuscripts. This has been*

*specified in the text.*

Line 200: "not limited by the acquisition geometry". The accuracy certainly does depend on the geometry (for example if the angle between ascending and descending is 1 deg vs 30). Please clarify.

*Reply: We clarified by saying that we refer to the actual Sentinel-1 geometry. "the Sentinel-1 suboptimal acquisition geometry".*

Line 245 "The magnitude" not "A magnitude"

*Reply: Corrected.*

Line 316. How does 1/10 t 1/30 of pixel precision translate into 4 m for an ~14 m azimuth pixel.

*Reply: The 1/10 to 1/30 of pixel is the reported actual measurement precision. In our study we are interested in ground displacements of glaciers. Therefore, the background motion outside of glaciers is considered as noise, e.g. snow drift, landslides, etc. Presence of this background motion contributes to the larger error. Perhaps if we were interested in studying snow drift outside of glaciers (which is irregular in time and space) or similar process we would claim a better precision, close to 1/10 to 1/30 of pixel. We added the following text "Our precision is lower than reported in \citep{strozzi2002} because we intentionally interpret the motion outside of glaciers (e.g. irregular snowdrift, landslides) as noise."*

Line 320-325. It's a bit unclear about what the various values are (eg. Maximum values 21, 18.. – maximum of what).

*Reply: For each pixel on the map during the 2017-2021 period, we compute a linear velocity and its standard deviation (Figures S1-S3) using the linear regression technique. Therefore, for each pixel and each component of the velocity, we get a standard deviation. To report these values in the manuscript we compute the average standard deviation for all pixels. Additionally, we provide the largest standard deviation among all pixels in the map - this shows the worst-case scenario, the largest error.*

Line 326. This comparison with GNSS is very unclear and for it to be at all true, it needs a lot of qualification (how GNSS with mm to cm/day can compare with SAR and 4 m over 12 days).

*Reply: Conceptually it is the same. Linear velocity is a slope of displacement time series. The precision of a computed slope depends on the number of measurements in the time series when each observation is affected by noise. It is equally true for GNSS and SAR-derived displacement time series. If the reviewer wants, we can remove the reference to GNSS without any loss of clarity in the manuscript; however, we believe the interpretation of this statement is straightforward and unit independent.*

Line 335. Please be a bit more clear about the Gardner product – are we looking at the 3-decade+ average of all the available LS?

*Reply: In the previous version we indeed used the 3+ decade averaged velocities. In the current version we additionally use Landsat-derived velocities computed during 2017 and 2018 (Figures S7-S14). This now is explained in the Discussion.*

Line 355. The definition of Eulerian/Lagrangian is not quite right (https://en.wikipedia.org/wiki/Lagrangian_and_Eulerian_specification_of_the_flow_field). And you really risk offending a lot of SE people who know the distinction.

*Reply: According to the referenced website: "The Eulerian specification of the flow field is a way of looking at fluid motion that focuses on specific locations in the space through which the fluid flows as time passes. This can be visualized by sitting on the bank of a river and watching the water pass the fixed location." and "the Lagrangian specification of the flow field is a way of looking at fluid motion where the observer follows an individual fluid parcel as it moves through space and time. Plotting the position of an individual parcel through time gives the pathline of the parcel. This can be visualized as sitting in a boat and drifting down a river."*
*This is exactly in agreement with what we say. The Eulerian specification is used in SAR measurements, where location a SAR pixel is a location of the observer "on the side of the river". The Lagrangian specification is used in GNSS where we measure "the pathline" of the receiver. We could address this comment in more detail if we better understood the reviewer's question.*

Line 380: "downward flow in the lower ablation" What matters is whether the flow is downward after accounting for the surface parallel flow component.

*Reply: That is correct, we added "with the slop steeper than surface topography".*

Line 385. To make statements about emergence velocity you have to account for surface parallel flow (i.e., the residual after subtracting surface parallel flow).

*Reply: That is correct. However, even without accounting for surface parallel flow here, the observation of downward flow having a greater slope than the surface topography implies emergent velocities are significantly less that what it expected and in a balanced glacier system.*

Figure 2. The caption needs further explanation. A lot of the other captions are a bit too terse.

*Reply: We modified the captions in Figures 2 and 3. We think that other captions are self-explanatory.*

*Dear Reviewer,*

*Thank you very much for providing your valuable comments (in red) that helped us to significantly improve our manuscript. Below we provide our detailed responses to your questions for consistency in red italic font. Text in black is from the first round of review.*

*Best regards,*
*Sergey Samsonov, Kristy Tiampo, and Ryan Cassotto*

We thank the authors for their response and revising the manuscript according to our comments.

1. There are still some parts of the manuscript (such as the equations/notations in Section 2) that are confusing or misleading.

2. Also, for some responses, the authors only replied to us but did not reflect any changes in the manuscript which may still be misleading or confusing to other readers. Please be sure to both address the reviewers' comments and reflect (even if the comment may sound simple) any possible changes in the manuscript.

3. The revised manuscript still lacks a section on formal error/accuracy analysis (by comparing to other reference velocity measurements) and a discussion section about current limitations and how to improve.

Below we attached the authors' response and only added highlighted comments to those questions that were not fully addressed.

*Reply: We addressed these comments in detail below.*

*Dear Reviewer 3,*

*Thank you very much for providing your valuable comments that helped us to significantly improve our manuscript. Below we provide our detailed responses to your questions in italic font. Here is a quick summary of changes:*

- *Addressed reviewers' comments to the best of our knowledge;*

- *Added recent Sentinel-1 data, mainly to investigate what is happening at region P1 at the*

*Malaspina Glacier;*

- *Recomputed offset maps using smaller 128x128 window and the Gaussian filter with 1.3 km 6-sigma width;*

- *Detected another surging Kluane Glacier and analyzed it in detail;*

- *Used OGGM software to extract flow lines and performed all analysis for selected flow lines;*

- *Simplified interpretation by removing reference to kinematic waves, which require more attention and, which possibly will be addressed in a separate publication.*

- *Provided animations for four AOIs.*

*Best regards,*
*Sergey Samsonov, Kristy Tiampo, and Ryan Cassotto*

This draft proposes a novel method for 3 -D velocity mapping of glaciers using modern spaceborne SAR measurements. Instead of using surface parallel flow constraint, this method combines speckle offset tracking and MSBAS , which is also assisted with regularization. It is further validated with Sentinel-1 data over 5 glaciers in Alaska. The draft is generally well written and the methodology is reasonable. However, there are couple of issues that need to be resolved/expanded in detail.

Major comments:

1.      Study area description is better to be extracted from Section I, together with the dataset description in Section 2, to form a separate section, named "Area and Data"

*Reply: We followed your advice and created a separate section "Study Area and Data".*

2.      The model description in Section 2 needs to be clearly rewritten and expanded in detail. If sufficient details do not fit the section, they could be added to an appendix then.

*Reply: We rewrote the section "Model" entirely.*

The equations/notations in Section 2 have been rewritten and also additional references have been added. It looks a bit better however, there are still places that look confusing or unclear.

For example,

1. please rewrite the 1st sentence of 4th paragraph in Section 2 to use notation of M x N to represent the dimension of matrix A. You could explain what M or N means using the number of unknowns/observables, e.g. number of SLC images.

   *Reply: Corrected as requested. "In the transform matrix A with N columns and M rows..."*

2. You did not answer our comment why the velocity vector in Eq. 3 only included $V_n3$, $V_e4$, $V_v4$.

   *Reply: Corrected the index, it should have been $V_n3$, $V_e3$, $V_v3$.*

3. The RO and AO vector definitions using rho and alpha elements need a vector transpose as they are column vectors.

*Reply: We use "{}" symbology to indicate "set" rather than "vector". Then a set can be converted to a row or column vector depending on the need. When we want to be more precise, we use "()" to indicate matrix (and vector is a matrix).*

4. Define right after Eq. 3 what those vector/matrix mean and note clearly the dimension using the above-mentioned number of unknowns/observables.

   *Reply: This is now provided.*

5. You now added the total dimension of 666 x 1109 in Section 2. However, 666 actually corresponds to the column dimension and 1109 the row dimension, which is opposite to the convention of the using row x column. Please reverse the order unless there is a reason for it. Also, you need to put your response to our comment about how these numbers are calculated based on the number of unknowns/observables into the main text.

   *Reply: Reversed as suggested. Provided explanation on how these numbers are calculated. "Thus, the total number of azimuth and range offset maps $M$ equals 446..."*

6. In Fig. 1, and also the simplified example of Section 2, you have 3+4=7 SLC images, so according to the statement (1st sentence of 4th paragraph in Section 2), the number of columns should be (7-1)*3=18, which is not equal to the actual number (12) in Eq. 3.

   *Reply: The number of SLC images is computed after applying the boundary correction. It is now clarified in the text. "Note that the boundary correction reduces the number of SLC images by two..."*

7. Please also put your response to our comment about regularization into the text.

   *Reply: Added. "The need for regularization arises...".*

8. It also seems that the actual value of the regularization parameter, lambda, does not matter. Because it got cancelled out in each regularization equation, where there are only two nonzero terms (both terms have lambda's that will be cancelled out) and all zero values for the other terms. Not sure why your reported value of 0.1 matters.

   *Reply: This is a standard methodology used for solving ill-posed inverse problem (see e.g. Tikhonov et. al., Numerical Methods for the Solution of Ill-Posed Problems). In general, we are looking for a solution that minimizes changes in the velocity between consecutive epochs (first objective) while fitting our input data (second objective). Lambda in this case is a parameter that weighs a contribution between the first and second objectives. Large lambda emphasizes the first objective, which produces smooth solution. Lambda can be looked at as a low-pass filter strength. The selection of optimal lambda is done as part of the inversion process.*

Detailed comments:
Line #13: this is the same sentence as included in the abstract, thus redundant

*Reply: We rewrote the redundant sentence in the Introduction.*

Line #26: SAR-based correlation algorithms not only operate on radar backscatter, but also radar backscatter and phase (complex-valued correlation).

*Reply: Corrected.*

Section I: you introduced multiple methods for velocity mapping (SPO, DInSAR, MAI), but did not mention what specific one you use in this work and why you chose that one. It is clear later in Section 2 that you used SPO, but would be better to motivate it in Section 1

*Reply: We commented in the second paragraph of the Introduction that we use the SPO technique and in the first paragraph of the Model section explained reasons (no need for phase unwrapping, produces range and azimuth results).*

We only found one sentence in Section 2 and did not see that you chose to use SPO in Section 1.

*Reply: Corrected by adding to introduction "We use in this study the SPO technique because it produces deformation maps in range and azimuth directions that do not require phase unwrapping."*

Line #74: the last sentence is also the same as that included in the abstract, i.e. redundant

*Reply: Corrected.*

Line #83-84: the number of pixels also need to be converted to distance in m. I see you want a square sampling interval on the ground by choosing 64 x 16 for Sentinel-1 images.

*Reply: This is approximately equal to 200x200m. This information is now provided in the last paragraph of Study area and Data section.*

Line #84-86: why isn't the correlation window (256 x 256) a square window on the ground to be consistent with the sampling interval. Also, the numbers you chose are equivalently 1km x 4km on the ground. With the 2km wide median filter, you essentially got a spatial resolution around 2km or at least on the order of km. Even though you resampled the products into 200m, this does not justify the spatial resolution is 200m. That said, the spatial resolution is too coarse over fast-moving glaciers, and the resulting spatial pixels are strongly correlated.

*Reply: Such a large window was required to obtain a distinct, statistically-significant peak of the 2D cross-correlation function; its square shape produced similar precision in range and azimuth directions in radar coordinates, and azimuth precision four times lower than range precision in geocoded products. We found that 128x128 (as in the revised version of the manuscript) is sufficient. If we chose to reduce the number of pixels in the azimuth direction M (to make square window on the ground) we would need to increase the number of pixels in the range direction N to keep M\*N=128\*128, but that would affect the precision in an unpredictable way.*

*In the revised version we reduced the correlation window to 128x128 pixels and used a Gaussian filter with a width to Gaussian 1.3 km (6-sigma). We recognize the benefits of having high-resolution results. Unfortunately, in this area, the application of a small window produces measurements that are too noisy, and if we only select pixels with high SNR the spatial coverage reduces to nothing. Therefore, we are limited to using a larger window.*

*We consulted the developers of the GAMMA processing software that is used to compute speckle offsets. We were advised that the window that is used to compute the offsets is not uniform, pixels in the centre have larger weights than those pixels on edges. The effective resolution is about four times higher than the window size. The process of the extraction of offsets, as it is implemented in the software, is not linear. We acknowledge that the spatial resolution is reduced by using such a large window. However, this is necessary for extracting temporal information. Note that the computation of offsets, in general, is not specific in any way to the technique presented here.*

*To confirm this we computed offsets for a single pair using 64x64, 128x128, and 256x256 correlation windows. In figure 1, below, we present these results before and after filtering. As you can see, while there are differences, overall the signal is consistent. Note that filtering does not reduce the resolution significantly. Again, we found that these processing parameters are optimal for our purposes in this region; however, it does not mean that they would be optimal in other areas.*

[Figure]

*Figure 1: (top-left) Seward range, (top-right) Seward azimuth, (bottom-left) Klutlan range, (bottomright) Klutlan azimuth.*

1. Based on what you clarified, using a large window might be okay for your area. But using a filter with larger width (km) is not recommended. Are you saying you replaced the previous median filter (2km width) with a Gaussian one (1.3km 6-sigma)? If so, 3-sigma Gaussian is roughly 650m, which might be okay but still a bit large.
2. From above figure (bottom right), it seems using your new window of 128 x 128 with filtering gives quite different results compared to 64 x 64 without filtering. So the question arises: the large window might be insufficient for this area and also the filter width might be too coarse.
3. You probably want to mention this as a limitation of the current processing and discuss how to improve the results in the future.

*Reply: We agree. We added the following text to the discussion: "One of the practical computational challenges of the SPO technique is the selection of pixels, which offsets are computed with high confidence. After multiple tests, we determined that the SNR function works very well but only when the search window is large. However, such a large window applied to the medium resolution SAR data limits the spatial resolution of the results. It is possible to use high-resolution SAR data and the 128$\times$28 pixels search window to overcome this limitation and achieve a high spatial resolution of results; however, such SAR data is not yet readily available on a global scale. The utilization of high-resolution SAR data also allows using a spatial filter with large, in terms of pixels, window size."*

Eq. 1: you should either cite a reference or explicitly show the proof of this equation. The way it current shows is introducing the equation out of the blue. When details of the proof is involved, you can also put that in an appendix if necessary.

*Reply: While it looks unconventional, it is a basic equation with a meaning similar to V\*t =D, that we believe does not require further derivation. It is used in many SBAS and MSBAS publications and its explicit representation can be deduced from the example (equation 3). We provided clarifications about this equation in the third paragraph of the Model section. Also, the Fialko et al., 2001 paper is cited that explains in detail how azimuth and range offsets are used to solve for the 3D deformation.*

Eq. 1: the matrix/vector notation should be clearly defined by providing the dimension, which should then be related to the number of ascending/descending acquisitions.

*Reply: We provided the following clarification. "In matrix A the number of columns is equal to the number of available SLC images minus 1 multiplied by three, and the number of rows is equal to the total number of range and azimuth offset maps computed from those SLC images." We also explained the size of the matrix in this particular case.*

Please refer to our above comments on rewriting this sentence and also the problem of applying this sentence in calculating the dimension for the particular case.

*Reply: This comment has been addressed and commented above. The entire modelling section was rewritten.*

Eq. 3: this simplified example is not clear. First of all, it is not clear how the Sa and Sr components are coupled in that way. To do so, you probably need a separate graphic illustration besides Fig. 2 or an appendix. If you can find a citation that does exactly the same thing, that would work too. Second, the notation of the rho and alpha elements in the column to the right of the "=" sign were never introduced since they are different from those described in Line #96 -101. Third, the last three elements in the velocity vector only show the northing of velocity at t3 and easting/vertical of velocity at t4. Why is that and what happened to the missing other components at t3 and t4, and what happened to t5?

*Reply: This comes from the geodetic analysis of seismic events and it is very well described in (Fialko et al., 2001; Bechor and Zebker, 2006), which are now referenced in our manuscript. We now explicitly show RO and AO in our simplified example (lines 85-90). Each row in A represents one range or one azimuth offset map. We believe it is now clearer.*

As mentioned above, you did not answer our comment why the velocity vector in Eq. 3 only included $V_n3$, $V_e4$, $V_v4$. What about the other missing terms at t3, t4, t5? Also, as mentioned above, please denote

number of unknowns/observables (e.g. number of SLC images as N) and use N to express each vector/matrix dimension right after Eq. 3. This is pretty standard way of introducing vector/matrix notation in writing scientific articles.

*Reply: This comment has been addressed and commented above. The entire modelling section was rewritten.*

Line #112: "any phenomenon" This is to vague. You need to be specific what type of phenomenon

*Reply: We meant to say any surface motion.*

Please reflect that change not only in the current response but also in the revised manuscript, otherwise it is still confusing to others.

*Reply: Corrected.*

Line #114: the dimension is 609 x 1014 for the matrix to be inverted. As mentiond above, how to relate these numbers to your total ascending/descending acquisitions. After Eq. 1, you should add a symbolic equation that relates the matrix dimension to the number of radar acquisitions

*Reply: After adding the most recent Sentinel-1 data to the revised version of the manuscript (we wanted to see what is happening at Malaspina Glacier at region P1) the dimensions of matrix became 666×1109. This means that we have 223 SLC images (223-1)\*3=666 and 108 ascending range and azimuth offset maps and 115 descending range and azimuth offset maps = 108+108+115+115=446 and the regularization rows are (223-2)\*3= 663. The total amount of rows is 446+663 = 1109. This now is explained in the Model section.*

As mentioned above, you need to move your response to the revised text as well. Once you define number of unknowns/observables as N or M as suggested above (e.g. number of SLC images as N), it is pretty straightforward to make this calculation by substituting N=223.

*Reply: This comment has been addressed and commented above. The entire modelling section was rewritten.*

Line #116: please report the specific computer setting and runtime for your case

*Reply: For us, it takes about 24 hours of processing time on a single node with 44 cores. An Message Passing Inteface (MPI) version of msbas software has also been developed. The processing time in an MPI version is reduced proportionally to the number of nodes.*

Line #117-120: add a sentence explaining why regularization is needed, and what happens if not included. Any comparison of the horizontal velocity results derived from the 3-D approach with regularization to those from the 2-D methods? Please add some simple analysis

*Reply: It is a somewhat specific and complex issue from the field of linear algebra, which most users probably do not want to know unless they want to develop their own software. There are three theoretically possible cases: the number of equations is less, equal or greater than the number of unknowns. In the equal case, the matrix is square and no regularization is required. In the greater case,*

*the least square solution is found using SVD – this is common in 1D MSBAS (more interferograms than SLCs). In the lesser case (as always in 2D and 3D MSBAS), the solution is found using the truncated-SVD, which is identical to the zeroth-order Tikhonov regularization. If we want to fill the temporal gaps, we need to apply higher order regularization (first and second-orders work equally well in this case). From the computational point of view there is no difference between the 2D and 3D problem. The need for regularization arises because SAR images from different tracks are acquired at different times, which results in more unknowns than equations, producing a rank-deficient, underdetermined problem.*

Even though only some readers might be interested in this topic, you still need to include it in the text to be complete. Also it is not trivial and widely used in the literature on ice velocity mapping.

*Reply: We added this information to text as requested.*

Line #121: what do you mean by "mean linear flow velocity" especially the word "linear"? Regarding "mean", is the 3-year mean value meaningful for those fast-moving glacier terminus? It is expected that such glaciers should have strong seasonal/interannual changes. Probably 1-year mean value is better

*Reply: With the technique presented here, we compute velocities between consecutive SAR acquisitions. Sentinel-1 data is acquired with either a six or 12 day revisit cycle, and velocities are computed for every revisit cycle interval (so-called instantaneous velocities). The flow displacement time series are then reconstructed from these instantaneous velocities. Assuming a 12 day Sentinel-1 revisit cycle, our technique produces about 365/12 = ~30 3D velocities per year. Since all these data cannot be presented in a single publication (30 velocities per year x 3D x 4 years ~ 360 figures), as a simplified representation of our results that require only three figures, we choose to compute mean velocities by fitting a line to the flow displacement time series, which we then divide by the length of our record. Along with the mean velocities for each of the four components, we compute their standard deviations and coefficients of determination (R2), which help us understand if the linear model provides a good approximation. For some regions, a linear approximation cannot capture all the complexity of the motion. For these regions, we plot flow displacement time series, which describe instantaneous velocity at each moment in time. Annual or any other duration (monthly, quarterly) velocities can also be computed from our flow displacement time series by aggregating time series at different intervals.*

*Concerning selecting the length of time to estimate mean flow, a shorter period could certainly be used; however, our aim for this manuscript was to demonstrate the technique used and the overall trends that occurred over 4 years. The flow displacement time series (particularly Figure 11) and text in the discussion address the benefits of short term analyses such as seasonal and inter-annual variability. Also four supplementary animations show instantaneous velocities for each of the studied glaciers.*

It is now clear to us. However, it is strongly recommended to rename the term "mean linear flow velocity". Alternatively, you should add a few more sentences from the above response to the main text otherwise, the readers might still feel confused and thought it was a statistical averaging mean value.

*Reply: This information is now provided in the model section. "For simplicity of presentation, a linear trend is computed by applying linear regression to the derived values in such a way as to illustrate the 3D displacement time series and three linear rate maps are used for visualizing the results. Note, that in the case of ..."*

Line #123: how much coarser resolution is the horizontal one resampled to? And also why is <5m/yr removed? Velocity estimates over slow-moving areas (e.g. < 15m/yr) are usually used to tie the products and calibrate the estimation bias. How did you calibrate your Sentinel-1-derived velocity products?

*Reply: The resolution and masking out is performed only for improving visualization (after processing is finished), otherwise, images in the figures get oversaturated with details. We use precise orbits downloaded from the ESA website. We calibrate the offsets by fitting and removing the polynomial model. This approach works well in this region where most areas do not show any motion. The entire Sentinel-1 scene is processed as a whole, and it is cut into small sub-regions only for visualization in the manuscript. Note that the entire Sentinel-1 scene extends far beyond the area shown in the manuscript. The software provides alternative methods of calibration that can be employed in other, more complex, regions (e.g. calibration against multiple reference regions, Z-score). You can see an example of the complete data set at the original resolution in Figure 5 and in supplementary files.*

Please objectively report your above calibration approach and clearly state that this is a limitation of the current processing chain in the main text. It seems too empirical and will be problematic for fast moving glacier areas. We would like to see some validation of the velocity results by comparing to other reference velocity measurements with some accuracy or error analysis, which is completely missing in this work.

*Reply: The need for calibration has nothing to do with our approach; it is a requirement for any SAR-derived deformation product (InSAR and SPO). The travel time of the SAR pulse depends on the precise location of the satellite and the state of the atmosphere. Because the state of the atmosphere changes and the position of a satellite is known imprecisely (and then imprecise coregistration and many other factors), the travel time difference between the first and second acquisitions (which is the deformation product after some manipulation) is determined with very low accuracy. But the measurement precision is very high. There is, of course, a fundamental difference between accuracy and precision, and in the SAR case, it manifests in such a way that all measurements are determined up to a constant. The process of determining this constant value is called calibration. Only in certain cases, a calibration process is difficult (e.g. large earthquakes can produce deformation across the entire image). In our case, ice flows along small (compare to the entire image) valleys, all we need to do is to measure offset in areas where there is no ice flow, from millions of pixels laying outside of glacier valleys we need to determine just one calibration constant. This can be done very accurately. We would be happy to compare our results to any other results. However, to the best of our knowledge, such data does not exist. All results used in this study are provided with confidence intervals, standard deviations, and coefficients of determination. In the revised version we also provide correlation and covariance matrices for synthetic tests. Most of the observed signals have been previously reported in cited papers.*

Line #180: "every single range and azimuth offset maps must be coherent at every pixel" what does it exactly mean?

*Reply: This means that if a pixel is incoherent on one of the offset maps (e.g. 20190201-20190213) it will be excluded from the processing and all results will have NaN value at that pixel. This approach ensures we used only the highest quality results. In general, our processing software can handle partially incoherent pixels (it will be filled by the regularization); however, in this study, we choose to utilize only pixels coherent in all offset maps so their precision is identical. The technique that utilizes partially coherent pixels will be discussed in the follow-up publications.*

It is the term "coherent" that sounds confusing to us. Please define the coherence you are referring to here or use another word to convey the exact idea.

*Reply: Coherent pixel is a pixel in which displacement is determined with high-confidence and which is retained for further processing. It is now corrected in the text.*

Line #181: "large correlation window followed by strong filtering" gives you much lower resolution and spatially correlated pixels. Isn't that problematic for fast-moving glacier terminus? Please comment and justify.

*Reply: That is correct. However, it is a necessity to use a large window and filtering as processing with a low correlation window produces very noisy results in this region. This has already been discussed above.*

*In the revised version we use smaller window and a filter with the Gaussian window, we found that it performs better for small and large glaciers. Finally, with the exception of a handful of tidewater glaciers (Hubbard, Tsaa, Guyot, and Taan), the majority of glaciers in our study area are land terminating and thus do not experience the rapid flow that typifies tidewater glacier termini.*

As mentioned above, although it seems to work for your area (note it is not convincing without a formal error analysis), you should explicitly add this as a limitation of the current processing routine, and explain how to improve it in the future for fast glacier outlets. The current revision of the manuscript still lacks a formal discussion about current limitations and how to improve.

*Reply: We believe we now addressed this comment in detail in discussion section ("One of the practical computational challenges..."). This is not a limitation of our technique but of SPO technique in general, our technique can be applied to data with any spatial resolution equally well. The proposed solution is to use SAR data acquired with higher spatial resolution. At the moment, such data is available only for some regions and it is not free.*

*Dear Reviewer,*

*Thank you very much for providing your valuable comments that helped us to significantly improve our manuscript. Below we provide our detailed responses to your questions in italic font.*

*Best regards,*
*Sergey Samsonov, Kristy Tiampo, and Ryan Cassotto.*

Review of "Measuring the state and temporal evolution of glaciers in Alaska and Yukon using SARderived 3D time series of glacier surface flow" by Samsonov, Tiampo, and Cassotto

The authors discuss a method for inferring time-dependent 3D surface velocity fields from synthetic aperture radar (SAR) data and apply this method to data collected from Sentinel I in 2016-2020 over five outlet glaciers in Alaska: Agassiz, Seward, Malaspina, Klutlan, and Walsh. Their results show complex glacier flow fields and temporal variations, and capture a host of interesting phenomena, including seasonal variations in ice flow, a surge, and dynamical glacier states. The manuscript is in line with a growing area of research that has great promise to advance our understanding of the cryosphere due to the volume of information available from modern remote sensing platforms. The authors have described an interesting and useful method and focus on an area where glaciers are exhibiting fascinating dynamic behavior. As such, this work will likely be of interest to the TC readership.

       This revised version of the manuscript is much improved from the original version. The authors added more context and details for their methods, attempted a minimal synthetic test to show that the method works, and improved the presentation and discussion of their results. The tone and precision of the writing in this draft better positions the work in the broader context of remote-sensing methodology and scientific knowledge. Overall, I enjoyed reading this draft and think that it is moving toward being suitable for publication, though I have some comments below that may be useful to consider.

- I think the authors need to do more to distinguish the methodology presented in this work and that of Guo et al., 2020. The authors merely mention Guo et al. in the introduction and say that they are presenting here an 'independently developed version of the algorithm' (line 46) in this manuscript. Some discussion of the differences between this algorithm and Guo et al. are needed as the current wording suggests that the authors developed an identical algorithm. If the algorithms are identical, the authors need to say so, otherwise they risk confusion for readers looking to implement or further develop the methods. In line 46, the authors point out that their software contains options to call other methods (published elsewhere) and Tikonov regularization schemes, but this is irrelevant to the distinction in the methods and algorithms. If there are no significant differences between these methods and those of Guo et al., 2020, it would seem that the presented method is not 'novel' (as stated in lines 119 and 326), and such statements should be removed.

*Reply: As you can see, the Guo et al., 2020 manuscript does not provide details about their methodology. We, however, noticed that their methodology applies weights in the transforms matrix based on pixel spacing (which is not justified in the text but given as a fact). Our technique does not require such weights. In any case, it is not critical for us to call the technique novel, so we removed "novel" from this manuscript and commented about the weights.*

- The synthetic tests appear a bit perfunctory and are certainly not as generally applicable as the authors' language suggests. At the very least, the sythetics need to be explained more thoroughly and in greater detail. I have a few comments:
  - One of the challenges of capturing the signals that the authors attempt to capture is that the three components of the velocity vector defined in an east-north-up (or other geographically referenced coordinate system) are not independent of one another. Rather, one would expect that the components of the velocity vector covary as they are representing the 3D flow of a glacier that is responding to some combination of internal and external forcing. So, while it's an interesting exercise to evaluate whether the method is capable of inferring time-varying signals in different velocity components that are unrelated to one another (i.e., different periods of variability), it's not a realistic test of a method meant to be applied to the natural environment. In other words, it's not that hard to infer different components when their time-varying functions are orthogonal. The challenge is in separating variability in the individual components when it is the speed (magnitude of the velocity vector) of the glacier that is varying with some given frequency and amplitude. Such a test has not been conducted and needs to be to show that the method works.

*Reply: As requested, we performed an additional synthetic test where north, east and vertical components of displacement are {2f(t), f(t), 0.5f(t)}, where f(t) is a combination of harmonic and linear functions (Figs 3 d-f). As you can see the reconstruction of this signal is very good.*

*The numerical analysis of covariance matrix is only meaningful when the actual variables are independent, as in our previous orthogonal tests. We previously showed that covariance terms for orthogonal tests are equal to zero (this can be seen from time series figures – reconstruction is very good).*

*When all three components of the displacement vector change coherently by design the covariance terms are naturally not equal to zero. What valuable information can be extracted from the covariance matrix in this case? In that case it is only meaningful to analyze the correlation matrix. It is expected that in this case all terms should be equal to 1. We present these covariance and correlation results in Table 2 and also discuss in the text.*

  - The covariance matrix (mentioned on lines 140 and 144) is never defined in the paper and needs to be if it's to be discussed. This is especially true given the authors finding that geometry doesn't matter in their method based on the covariance terms taking on a value of zero. This is a surprising finding that contradicts decades of work in GPS positioning and other work in inferring multi-dimensional surface velocity fields from SAR data (as referenced in a previous review), so a little more discussion would be useful as would an explanation for why it is the authors' method doesn't suffer the same challenges as well-established methods that attempt to do essentially the same thing. o Lines 142-143: Why would the rank be 3 if you have ascending and descending range and azimuth offsets? These represent 4 unique viewing geometries, so one would expect the rank to be 4.

*Reply: Our covariance matrix is defined in a standard way as a square matrix that gives the covariance between each pair of elements of a given random vector. It is symmetric and positive semi-definite and its*

*main diagonal contains variances (i.e., the covariance of each element with itself). The random vector in our case consists of three components of velocities measured at each epoch.*

*As you can see in Table 2, the covariance terms for uncorrelated input signal increase with the magnitude of the noise. However, they still remain small at the current noise level. This can be seen in our results as well. Each map pixel is processed independently in time yet the displacement (or velocity) field appears correlated (consistent velocity over all glaciers in all three components).*

*In MxN matrix the maximum possible rank is min(M,N), in our case the matrix is 3x4 (i.e. three unknown components of velocity and four equations), therefore the maximum possible rank is 3. The problem is over-determined and the solution is found in the least-square sense.*

> o The authors' mention of the tensor rank seems to indicate a misunderstanding of the point of geometric influence and the role of the covariance matrix. The rank simply shows that there are 4 unique viewing geometries, which is enough information to invert for 3D velocity field. That is obvious and has never been in question. Rather, the question lies in the accuracy and precision of the inferred components. In other words, given a finite signal-to-noise ratio, is there enough information to constrain the 3D velocity vector components in time? The authors have shown that the answer is maybe, but the physical contrivance of their synthetic tests (as noted in my comment above starting with 'One of the challenges…') leaves the question open. The rank of matrix A can only provide a negative answer to this question as it is merely a necessary (not sufficient) condition for inferring the information that the authors purport to infer.

*Reply: In general, the number of nonzero singular values of matrix equals the rank of the matrix. The conditioning value measures how much the output value of the function can change for a small change in the input argument, it is equal to the largest singular value divided by the smallest singular value. So, when the smallest singular value is very small comparing to the largest singular value the conditioning number is large and the numerical problem is unstable (i.e. sensitive to the noise). In our case the conditioning number is small so the solution is stable.*

> o Further to my point about the covariance matrix, one of the major shortcomings of the methodology presented here is the lack of any formal uncertainties. What I mean is that it's possible to compute the uncertainties in the offset fields, and in a methodology as developed as MSBAS, these uncertainties should be carried through to some formal error estimate for the resulting velocity fields, and this is where imperfect observational geometries (as is virtually always the case with satellite observations due to the nonorthogonal angles between orbits) will amplify errors. Thus, spatial variability, as quantified here by the authors, is better than nothing, but not as good as formal UQ.

*Reply: All velocities are now provided with standard deviations (Figures S1-S3) and the coefficients of determination R2, Figures (S4-S6). Previously this information was not shown in the manuscript but was included in the supplementary files (to be submitted to the data repository). You can also see error bars in time series (Figures 11-12). We agree that it would be great to have an equation that measures uncertainty in a formal way.  Note, however, that since the precision of input data (offsets) can be considered equal for all pixels because it only depends on sensor parameters then the precision of computed variables (3D velocities) is also constant. Intuitively, there is not much value in a single*

*precision number. Practically, the spatial variability is the only currently available methodology for SBAS-derived methods.*

Minor comments:
- I still fail to see the value in the discussion contained in the paragraph beginning in line 275 as the distinction between Eulerian and Lagrangian coordinates is well established, but perhaps the authors are aware of some related controversy that needs to be addressed. The wording in this manuscript (line 281) suggests that no such controversy exists for the TC readership (and I know of now such issues), so I still contend that this should be removed from the paper. That said, the authors are clearly intent on making this well-known point, and I won't bring it up again. I'll simply end by saying that if the authors are intent on making this point, they should at least say something about the conversion between Eulerian and Lagrangian coordinates.

*Reply: We do not mind repeating this, perhaps well-known point, because to us, geophysicists mainly working with small deformation of rigid objects this is an unfamiliar concept, although it is much better understood in the cryospheric field. We commented that the material derivative can serve as a link between Eulerian and Lagrangian descriptions of continuum deformation.*

- A few sentences in this manuscript are identical to those found in Samsonov et al. 2021 (Remote Sensing of the Environment). This isn't a major issue as these are minor sentences that give context, but a bit of editing will avoid the appearance of copy-paste between two published works. This comment should not be taken by the editor or anyone else as an ethical issue, merely a logistical detail as the published manuscript was probably written around the same time as this one and it's easy for these things to happen.

*Reply: We believe that after this revision this issue is resolved.*

---

## Author Response (AR3)

*Dear Dr. MacGregor*

*Thank you very much for the opportunity to revise our manuscript. We successfully addressed all the comments. Below, please find our response to your comments in italic font.*

*Best regards,*
*Sergey Samsonov, Kristy Tiampo, and Ryan Cassotto.*

Editor Decision: Publish subject to minor revisions (review by editor) (26 Jul 2021) by Joseph MacGregor
Comments to the Author:
Dear Dr. Samsonov et al.,

Thanks for your thorough revisions to your revised MS following the three referee reports. Your MS generated more intense feedback from the referees than I had initially anticipated, but all in cases they have stayed engaged with the peer review process and your MS has improved substantially. So, even though there is still some disagreement in approach between yourselves and the referees, it has been fairly debated and your MS will undoubtedly motivate further investigations into velocity time series from large volumes of SAR imagery. I will accept your MS to The Cryosphere once you address the following comments below.

*Reply: Thank you for you time and effort.*

NB1: Note that line numbers are in reference to your track changes version (tc-2021-257-ATC2.pdf).

NB2: WRT line 396, I *will not* accept this MS if the full data/code repository is not available for inspection beforehand. Include the final *public* URL in your next revision.

*Reply: We added  url as requested.*

Abstract: Given that the primary advance of the MS is IMO the application of a new-ish method to a large volume of data regarding glacier flow, the abstract ought to give some clue as to what that method's key advance actually *is*. Also, given the ferocity of the debate amongst the reviewers on this point, the abstract ought to address its uncertainty relative to earlier or contemporaneous methods. As it stands, the abstract is devoid of these highly relevant high-level concerns, which strike me as more important than satellite revisit times.

*Reply: We made a few changes to the abstract. We believe it now better describes the key advances.*

13: "Glacier dynamics" cannot be reduced solely to surface velocity. Strictly speaking their "dynamics" concern the evolving force balance. The downstream effects listed are an odd mix and not well supported when only considering surface velocity. Rethink this whole first sentence.

*Reply: Corrected.*

13: "intensity" is simply the wrong word here. "magnitude and direction" is in much more common

usage and physically clearer. A referee brought this up previously and you responded that it was "corrected"… but it wasn't.

*Reply: Corrected.*

38: "specifically" makes more sense here than "especially"

*Reply: Corrected.*
42: The GNSS acronym was introduced earlier, and GPS is left undefined. Use GNSS instead.

*Reply: Corrected.*

44: combined using the MSBAS technique.

*Reply: Corrected.*

59: remove "But"

*Reply: Corrected.*

61: Riel et al. (2021), who adopted some of the methods of Riel et al. (2014, 2018) and applied them

*Reply: Corrected.*

72: Figures should be referenced in the order they are presented. Either Figure 4 should move up to #1 or not be referenced here. I suggest the later approach.

*Reply: Corrected.*

77: using the speckle

*Reply: Corrected.*

91: incidence angle. (IMO "azimuth" does not need to be qualified with "angle")

*Reply: Corrected.*

141: Passive voice. "We used a value of 0.1 for lambda." Was this value selected using L-curve?

*Reply: Corrected.*

151: hundreds and often thousands of columns

*Reply: Corrected.*

156: "called from C++ code" does not seem relevant here

*Reply: Corrected.*

162: Passive voice

*Reply: Corrected.*

164: in the respective legends

*Reply: Corrected.*

165: 10% of what? I assume the harmonic's amplitude but this origin should be made explicit here not just legend.

*Reply: Corrected. Let's say the ice flows with the velocity 1 m/day. This means that when sampled every 12 days (Sentinel-1 revisit period) the individual speckle offset map will detect 12 m of displacement. Then we take 10% (or 1.2 m) of that as a noise. The annual amplitude is not a good reference for an error because it does not account for the data sampling rate (e.g. 12 days).*

168: Now I'm confused. Did you add 10% of amplitude noise or 0/10/30? Reconcile with 165.
*Reply: There are three grouped tests, labeled in the text as First, Second, Third. The first test was done with 0/10% noise and two latter tests were done with 0/10/30%. The first test used only one value of noise to preserve paper space, it produced three times more figures, one for each component.*

184: Is "cross-feed" the best term here? Not sure

*Reply: Corrected. Deleted this part of the sentence.*

186: Overall these tests indicate that

*Reply: Corrected.*

193: RGI Consortium

*Reply: Corrected.*

196: "pale in comparison" is strong language, especially given that parity is now effectively reached via ITS_LIVE and other international efforts. Reconsider this phrasing.

*Reply: Corrected.*

225: Does "lower" here mean "coarser" (larger value) or "finer" (smaller value)? Unclear.

*Reply: Corrected. Coarser.*

228: A pers. comm. citation from the GAMMA developers is appropriate here.

*Reply: Corrected.*

229: using the TerraSAR-x

*Reply: Corrected.*

231: Filter width

*Reply: Corrected.*

236: This statement about data volume is odd and should either be further specified somehow or removed.

*Reply: Corrected.*

239-240: I don't understand what "approximated to the actual glacier flow pattern" means. I assume OGGM uses a DEM for the flowlines, whereas you have actual surface velocities available. Clarify this statement.

*Reply: Deleted this sentence. The idea was to say that the location of flow lines derived using coarse DEM is approximate. I think it is clear without saying. All numerical results are approximate.*

243: intensity -> backscatter?

*Reply: Corrected.*

251: direction or sign?

*Reply: We believe that the direction of flow can be seen in the flow velocity vector map.*

303: Unless the authors have checked, it is unlikely that there are no GNSS stations on any of the studied glaciers during at least some part of the 5-year period. If not checked, I recommend moderating this statement to something like "we did not compare against GNSS)

*Reply: We checked, there is no freely available GNSS data for these glaciers.*

305: "~4 m" not "four meters"

*Reply: Corrected.*

318: "they still show some agreement" -> "

*Reply: Corrected.*

334-347: As a referee pointed out, offending the solid earth community by implying they don't understand Lagrangian vs. Eulerian reference frames is a strange and unnecessary stance. In recalling my own education, scientists in those disciplines learned these concepts right alongside myself. This paragraph should be shortened substantially by assuming that the likely reader is a glaciologist who is familiar with the two reference frames and by focusing on the argument the authors seek to make on the links between GNSS and SAR flow displacement time series for *glaciers*. A distinction for that link over deglaciated tectonically deforming surfaces should be made only as a final aside.

*Reply: Corrected. We removed this part of the discussion.*

348-365: As the referees pointed out more than once, the ability to recover credible vertical displacements concurrent with horizontal displacements from large data volumes with MSBAS is a potentially valuable advance. However, the very large magnitudes reported (e.g., >200 m/yr) *require* further explanation, some of which was provided in the response to reviewers (e.g., ArcticDEM not available at coincident times, sparser ICESat-2 data). Without that further explanation, there is a risk of casual misinterpretation of the reported vertical velocities as rates of elevation change rather than evolving submergence/emergence velocities indicative of broader (and faster) changes in dynamic configuration than previously understood (and also much greater than short-term SMB fluctuations). This paragraph appears to be the right place for that clarification, as the discussion concerning Malaspina Glacier mostly works.

*Reply: Corrected.*

351: "is more concerning" reframe this qualification as it is already well established that most Alaskan glaciers are losing mass rapidly (e.g., Larsen et al., 2015).

*Reply: Corrected.*

367: Move Meier and Post (1969) citation to the group at the end of the sentence.

*Reply: Corrected.*

384, 395: Please be clearer here as to what you mean by "data". The data you use are SLC Sentinel-1 images, but you're probably not archiving those with the MS as they're already at ASF. I believe the velocity dataset (over space AND time) is what is meant.

*Reply: Corrected.*

Figures throughout MS: It is odd to use the standard abbreviation for meters (m) but not that for year (yr) when showing speeds. Recommend changing unit abbreviation from m/year to m/yr.

*Reply: Corrected.*

Figure 4: Add a red arrow in inset map of North America so that small red box is more easily spotted.

*Reply: Corrected.*

Thanks again for choosing The Cryosphere for your work,

Joe MacGregor
NASA/GSFC